J Physiol 603.19 (2025) pp 5565–5612

5565

# Ectopic sodium channel expression decreases excitability of *Drosophila* Kenyon cells

Katie Greenin-Whitehead[1,2] ⓘ, Eyal Rozenfeld[3,4,5] ⓘ, Anthony Moreno-Sanchez[6] ⓘ, Melissa W. Tan[1,2] ⓘ, Kurtulus Kullu[6,7] ⓘ, Jessica Ausborn[6] ⓘ, Moshe Parnas[3,4] ⓘ and Andrew C. Lin[1,2] ⓘ

[1] *School of Biosciences, University of Sheffield, Sheffield, UK*
[2] *Neuroscience Institute, University of Sheffield, Sheffield, UK*
[3] *Department of Physiology and Pharmacology, Gray Faculty of Medical and Health, Sciences, Tel Aviv University, Tel Aviv, Israel*
[4] *Sagol School of Neuroscience, Tel Aviv University, Tel Aviv, Israel*
[5] *Neuroscience Institute, New York University Langone Medical Center, New York, NY, USA*
[6] *Department of Neurobiology and Anatomy, Drexel University College of Medicine, Philadelphia, PA, USA*
[7] *Department of Computer Science, University of Central Florida, Orlando, FL, USA*

The peer review history is available in the Supporting information section of this article (https://doi.org/10.1113/JP288790#support-information-section).

**Abstract figure legend**: We tested the effects of expressing the bacterial voltage-gated sodium channel NaChBac in the Kenyon cells of the *Drosophila* memory centre, the mushroom body. NaChBac expression decreases expression of endogenous voltage-gated sodium channels (left), reduces action potential firing (upper) and prevents learning (lower). However, odour-evoked calcium influx in Kenyon cell axons and dendrites (but not somata) is normal after 4-day adult-only NaChBac expression (but not 2-day adult-only or constitutive expression).

**Katie Greenin-Whitehead** is a Research Associate in the Lin Lab at the University of Sheffield. She completed her PhD in the same lab, where her research focused on homeostatic regulation and olfactory coding in *Drosophila melanogaster*. She is particularly interested in how Kenyon cells adapt to exogenous stimuli.

E. Rozenfeld, A. Moreno-Sanchez, and M. W. Tan contributed equally to this work. J. Ausborn and M. Parnas also contributed equally to this work.

The Journal of Physiology

**Abstract** Neurons stabilize their physiological properties in part by homeostatic compensation between different ion channel conductances. However, little is known about this process in the central brain *in vivo*. We studied this problem in Kenyon cells, the third-order olfactory neurons in the fruit fly *Drosophila* that store olfactory associative memories. We investigated whether Kenyon cells regulate their excitability homeostatically by testing how their activity is affected by ectopic expression of the bacterial voltage-gated sodium channel NaChBac, a manipulation previously reported to increase neuronal excitability in other systems. Surprisingly, NaChBac expression decreases Kenyon cell excitability. Whether expressed constitutively (throughout development) or only in the adult, NaChBac expression in Kenyon cells suppresses Kenyon cell spiking, reduces odour-evoked calcium influx, and prevents olfactory aversive conditioning. However, odour-evoked calcium influx in Kenyon cell axons and dendrites (but not somata) is normal after 4-day adult-only NaChBac expression (but not 2-day adult-only or constitutive expression), suggesting limited homeostatic regulation of calcium influx that is prevented by developmental NaChBac expression. NaChBac expression also decreases expression of endogenous voltage-gated sodium channels (Para) in the spike initiation zone, suggesting homeostatic regulation of sodium influx. Indeed, a compartmental model best fits the data when the exogenous NaChBac conductance is accompanied by a decrease in endogenous sodium conductance. These results suggest that manipulating neuronal activity with ion channels can have unexpected effects depending on compensatory plasticity.

(Received 22 February 2025; accepted after revision 16 July 2025; first published online 16 August 2025)

**Corresponding author** A. C. Lin: School of Biosciences, University of Sheffield, Sheffield, UK. Email: andrew.lin@sheffield.ac.uk

## Key points

- Neurons stabilize their physiological properties through compensation between key ion channels, but little is known about this process in the central brain *in vivo*.
- Here we test homeostatic compensation *in vivo* in *Drosophila* Kenyon cells, the neurons that store olfactory associative memories, by ectopically expressing NaChBac, a bacterial voltage-gated sodium channel commonly used to increase neuronal excitability.
- Surprisingly, NaChBac expression in Kenyon cells *decreases* their excitability: it reduces expression of endogenous sodium channels, prevents spiking, reduces odour-evoked calcium influx, and impairs learning. The electrophysiological phenotype is reproduced in a compartmental model.
- However, odour-evoked dendritic/axonal calcium influx (but not spiking or learning) returns to normal if NaChBac is expressed for 4 days only in adults (not during development), suggesting limited homeostatic regulation of voltage-gated calcium influx.
- These results show that manipulating ion channels can have unexpected effects depending on homeostatic compensation.

## Introduction

Neurons compensate for activity perturbations through homeostatic plasticity, a phenomenon critical for stabilizing average activity levels and maintaining useful network activity patterns (Abdelrahman et al., 2021; Marder & Goaillard, 2006; Wen & Turrigiano, 2024a). In particular, compensation between different ion channel conductances is a fundamental mechanism for ensuring stable neuronal physiology in the face of variability, but can also lead to pathological outcomes (Li et al., 2019; O'Leary, 2018; Waxman et al., 2004; Yang & Prescott, 2023). However, we still know little about how this process works in the central brain *in vivo*. Most studies on homeostatic plasticity of intrinsic excitability have been done on reduced preparations like dissociated neurons or brain slices, yet neuronal physiology *in vivo* in awake animals differs from *ex vivo* preparations in important ways (Cazakoff et al., 2014; Destexhe et al., 2003; Kirov et al., 1999). Moreover, *in vivo* manipulations are required to link homeostatic compensation to sensory processing and behaviour.

While homeostatic plasticity in the central brain *in vivo* is beginning to be studied, most studies have focused on

synaptic rather than intrinsic compensation mechanisms (Barnes et al., 2015; Keck et al., 2013; Kuhlman et al., 2013). In addition, many studies of intrinsic homeostatic plasticity focus on ion channel compensation for perturbations in synaptic or sensory input (Baines et al., 2001; Driscoll et al., 2013; Gainey et al., 2018; Hengen et al., 2013; Maffei & Turrigiano, 2008; Wen & Turrigiano, 2024a, 2024b); or they find correlations between conductance/expression levels of different ion channels consistent with compensation, without directly manipulating ion conductances (Schulz et al., 2006, 2007; Temporal et al., 2014). There are relatively fewer examples where manipulating ion channel conductances has been shown to causally affect other ion channels, and these are in dissociated neurons (Nerbonne et al., 2008; Peng & Wu, 2007; Swensen & Bean, 2005), *ex vivo* preparations (MacLean et al., 2003; Wart & Matthews, 2006), or neurons outside the central brain (Bergquist et al., 2010; Kim et al., 2017; Kulik et al., 2019; Li et al., 2019; Parrish et al., 2014). Thus, it remains unclear how ion channels compensate for each other in the central brain *in vivo*.

We recently established the mushroom body in the *Drosophila* olfactory system as a model of homeostatic plasticity *in vivo* (Apostolopoulou & Lin, 2020). The mushroom body's principal excitatory neurons, the Kenyon cells, respond sparsely to odours in order to store odour-specific associative memories (Parnas et al., 2024). When Kenyon cells are subjected to prolonged excess inhibition from the GABAergic neuron APL, they compensate by increasing their activity through a combination of reduced inhibition and increased excitation (Apostolopoulou & Lin, 2020). However, it remains unknown how Kenyon cells use plasticity of intrinsic excitability to compensate for perturbations in ion channel conductances.

Because overexpression or incomplete knockdown of endogenous channels could potentially be nullified by autoregulatory feedback without triggering compensation of other conductances (Minchington et al., 2020), we asked how Kenyon cells compensate for ectopic expression of a heterologous channel, the prokaryotic voltage-gated sodium channel NaChBac. Originally found in the bacteria *Bacillus halodurans*, NaChBac opens at lower voltages and inactivates at higher voltages with $10-100\times$ slower inactivation kinetics than eukaryotic voltage-gated sodium channels because it lacks the 'ball and chain' inactivation plug (Pavlov et al., 2005; Ren et al., 2001). When expressed in neurons, NaChBac commonly produces long ($>100$ ms) plateau-like depolarizations (Giachello et al., 2022; Kelsch et al., 2009; Sheeba, Sharma et al., 2008; Sim et al., 2013; Xue et al., 2014; Zhang & Gaudry, 2016) thanks to its slow inactivation kinetics. Therefore, NaChBac is commonly used to constitutively overactivate neurons, and it often produces effects consistent with increased intrinsic excitability. For example, expressing NaChBac in mammalian neurons increases their spontaneous activity and intrinsic excitability, and induces a compensatory increase in inhibitory inputs (Kim et al., 2020; Sim et al., 2013; Xue et al., 2014).

NaChBac is also commonly used to overexcite neurons in *Drosophila*, where it can increase axonal conduction speed (Kadas et al., 2019) and increase spontaneous activity (Park et al., 2016). Expressing NaChBac in neurons often has behavioural effects consistent with overactivating them (Donlea et al., 2011; Hergarden et al., 2012; Root et al., 2011) and opposite to silencing them (Crocker et al., 2010; Deng et al., 2019; Nitabach et al., 2006; Zimmerman et al., 2017). However, sometimes NaChBac expression does the opposite, with behavioural effects consistent with silencing neurons (Luan et al., 2006) and opposite to activating them (Alphen et al., 2021; Flood et al., 2014). Indeed, NaChBac expression may simultaneously cause long plateau depolarizations and hyperpolarize the resting membrane potential (Sheeba, Fogle et al., 2008; Sheeba, Sharma et al., 2008).

In light of these conflicting results, we hypothesized that NaChBac might have unusual effects on neuronal physiology. Indeed, NaChBac expression in Kenyon cells blocked endogenous action potentials and strongly reduced odour-evoked calcium influx and olfactory learning. However, odour-evoked calcium influx in Kenyon cell axons and dendrites (but not cell bodies) was normal with $>4$ day adult-only expression (but not 2-day adult-only or constitutive expression) despite the lack of action potentials, suggesting homeostatic compensation of axonal/dendritic calcium influx with adult-only NaChBac expression. This apparent compensation did not restore learning, as NaChBac expression in Kenyon cells reduced olfactory learning in all conditions. NaChBac expression also reduced the expression of endogenous sodium channels, suggesting homeostatic regulation of sodium influx. Compartmental modelling predicted that the observed effects of NaChBac can be best explained by a concomitant loss of endogenous sodium conductances, and that NaChBac increases excitability at low expression levels but blocks spiking at high levels.

## Methods

### Flies

Flies (*Drosophila melanogaster*) were raised at 18°C or 25°C, in standard plastic/glass fly vials/bottles, with *ad libitum* access to standard cornmeal agar. For experiments involving auxin treatment, 10 ml of 1 M 1-Naphthaleneacetic acid potassium salt (Glentham Life Sciences Ltd.) dissolved in water was added per litre of liquid media to give a final concentration of 10 mM. Flies were anaesthetized by carbon dioxide via

a Flypad (Flystuff) for genetic crosses, or on ice before imaging/electrophysiology experiments. Either carbon dioxide or ice was used before dissecting the brain out for structural imaging. Before behavioural experiments, flies of the correct genotype were selected by carbon dioxide anaesthesia at least 1 day before the experiment; flies were loaded into the behavioural apparatus (see below) by aspiration without anaesthesia. Details of genetic strains used are given in Appendix A, Tables A1 and A2.

## 2-photon calcium imaging

Calcium imaging was performed as described (Apostolopoulou & Lin, 2020). Cuticle and trachea in a window overlying the mushroom body were removed, and the exposed brain was superfused (perfusion pump Watson-Marlow 120S DM2, ~2.7 ml/min) with carbogenated (95% $O_2$, 5% $CO_2$) solution containing 103 mM NaCl, 3 mM KCl, 5 mM trehalose, 10 mM glucose, 26 mM $NaHCO_3$, 1 mM $NaH_2PO_4$, 1.5 mM $CaCl_2$, 4 mM $MgCl_2$, 5 mM TES, pH 7.3. The experimenter was not blind to fly genotype. Odours ($10^{-2}$ isoamyl acetate (Sigma, 112674), $10^{-2}$ butyl acetate (Sigma, 402842), $10^{-2}$ $\delta$-decalactone (Sigma, W236101); $10^{-1}$ ethyl butyrate (Fisher, 10429220), $10^{-1}$ 4-methylcyclohexanol (Sigma, 153095), $10^{-1}$ 3-octanol (Sigma, 218405), $5 \times 10^{-2}$ apple cider vinegar (Tesco)); all odours diluted in mineral oil (except for the vinegar, which was diluted in water) were delivered by switching mass-flow controlled carrier and stimulus streams (Sensirion) via software-controlled solenoid valves (The Lee Company). The flow rate at the fly was ~0.5 l/min.

Brains were imaged by two-photon microscopy (Ng et al., 2002; Wang et al., 2003). Fluorescence was excited by 75–80 fs pulses of 910 nm light at 80 MHz from a Ti:Sapphire laser (Spectra-Physics eHP DS), attenuated by a Pockels cell (Conoptics, Model 350-80LA) and coupled to a galvo-resonant scanner on a Movable Objective Microscope (Sutter Instruments). Excitation light was focused by a 20×, 1.0 NA objective (Olympus XLUMPLFLN20XW), and emitted photons were passed through a 750 nm short-pass filter (to exclude excitation light) and bandpass filters (green: 525/50; red: 605/70), and detected by GaAsP photomultiplier tubes (Hamamatsu Photonics, H10770PA-40SEL), whose currents were amplified (Thorlabs, TIA60) and transferred to the imaging computer running ScanImage 5 (Vidrio Technologies). Volume imaging was performed using a piezo objective stage (nPFocus400, nPoint) using ScanImage's FastZ control in sawtooth mode.

Movies were motion-corrected in X-Y using the moco ImageJ plugin (Dubbs et al., 2016), with pre-processing to collapse volume movies in Z and to smooth the image with a Gaussian filter (standard deviation = 4 pixels;

the displacements generated from the smoothed movie were then applied to the original, unsmoothed movie) (Bielopolski et al., 2019). $\Delta F/F$ traces were calculated using manually drawn regions of interest for the background and brain structure of interest, and smoothed with a 0.2 s boxcar filter, in MATLAB (Mathworks) or Igor Pro 9 (WaveMetrics). Where traces with different frame times needed to be averaged, traces were linearly interpolated to the same frame rate. Flies were excluded if the neurons of interest did not respond to odour, the GCaMP6f signal was too low/noisy, or the brain moved too much to correct for motion artefacts.

Activity maps were generated as in Apostolopoulou and Lin (2020). Briefly, movies were smoothed with a 5-pixel-square Gaussian filter (standard deviation 2). Baseline fluorescence was taken as the average fluorescence during the pre-stimulus period. Frames with sudden, large axial movements were discarded by correlating each frame to the baseline image and discarding it if the correlation fell below a threshold value, which was manually selected for each brain by noting the constant high correlation value when the brain was stationary and sudden drops in correlation when the brain moved. $\Delta F/F$ was calculated for each pixel as the difference between mean fluorescence during the stimulus period *versus* the baseline fluorescence ($\Delta F$), divided by the baseline fluorescence. In this study we did not apply a threshold to zero out low-responding pixels. We excluded non-responsive flies and flies whose motion could not be corrected.

Inter-odour correlations were calculated by first aligning the activity maps of each odour response by maximizing the inter-odour correlations of baseline fluorescence, and then converting image matrices of the activity maps of each odour response into linear vectors and calculating the Pearson correlation coefficients between each 'odour vector'. A threshold for baseline fluorescence was applied as a mask to the activity map to exclude pixels with no baseline GCaMP6f signal. Areas with non-GCaMP6f fluorescence (e.g. cuticle) or non-Kenyon-cell-soma areas (e.g. calyx) were manually excluded. For calculating the overall response across all cell bodies, the same region of interest was used as for correlation/sparseness calculations. Population sparseness was calculated for activity maps using the following equation (Vinje & Gallant, 2000; Willmore & Tolhurst, 2001):

$$S_P = \frac{1}{1 - \frac{1}{N}} \left( 1 - \frac{\left( \sum_{i=1}^{N} \frac{r_i}{N} \right)^2}{\sum_{i=1}^{N} \frac{r_i^2}{N}} \right) \tag{1}$$

where $N$ is the number of pixels and $r_i$ is the response of each pixel. Analysis code is available at https://github.com/aclinlab/calcium-imaging.

## Patch-clamp electrophysiology

For *in vivo* whole-cell recordings, the preparation and perfusion were the same as for calcium imaging. Flies 2–4 days old were used as previously described (Rozenfeld et al., 2023). Briefly, the brain was visualized on a Scientifica SliceScope with a 40× water immersion objective. Patch pipettes of 9–15 MΩ resistance were used. The intracellular solution contained (in mM): potassium aspartate 140, HEPES 10, KCl 1, MgATP 4, $Na_3GTP$ 0.5, EGTA 1; pH adjusted to 7.3 and osmolarity 265 mOsm. The recordings were performed at 25°C using an Axon Instruments MultiClamp 700B amplifier and pCLAMP 10.7 in current- or voltage-clamp mode. Data were low-pass filtered at 1 kHz and sampled at 50 kHz. Upon break-in to the cell, a small constant current was applied to maintain a membrane potential of −60 mV. The membrane capacitance was estimated by fitting an exponential decay function to the hyperpolarization induced by a negative current step; the capacitance was taken as the time constant divided by the input resistance, which was in turn taken as the plateau of the fitted exponential decay function (steady-state voltage step) divided by the current step. Cells with input resistance less than 0.8 GΩ were excluded (see Appendix A). Excluded data points are shown in Fig. A3. Series resistance (see Appendix) was automatically compensated by pCLAMP. Uncompensated series resistance was estimated by fitting the exponential decay time constant of the capacitive transient and dividing by the membrane capacitance (Molleman, 2003).

Action potentials were extracted using the findpeaks MATLAB function followed by manual inspection to verify correct identification. Since the measured Kenyon cell action potentials' amplitudes were small and varied in size throughout time, findpeaks was only able to detect large, uniform action potentials. In cases where spike detection failed, spikes were detected using visual inspection. A spike was included based on a sharp depolarization and a pronounced narrow peak followed by repolarization. For analysing spike waveforms and spike thresholds, spikes were only used if they were from cells that fired a total of at least 10 spikes over the range of +10 to +100 pA current injection.

To analyse the spike shape, voltage traces 20 ms before and after the voltage peak of each spike were extracted and averaged for each current step. Voltage traces were normalized to the voltage at spike onset, which was identified as the peak in the second derivative of the voltage within the 5 ms before the peak voltage (with 2 ms moving average smoothing at each differentiation step). The time of spike onset was calculated from the average waveform across all spikes at each current step. The spike height was the difference between the voltage at the peak *versus* at spike onset. The minimum voltage to spike was taken as the minimum absolute voltage at spike onset (peak second derivative) across all spikes identified for that cell. The speed of the action potential upstroke was taken as the peak first derivative of voltage, $dV/dt$ (after smoothing $V(t)$ by a 0.6 ms moving average), in the 5 ms preceding the peak voltage.

To measure NaChBac currents in voltage-clamp, the current trace was first smoothed by a 50 ms moving median filter to remove fast inward currents arising from unclamped spikes. The steady-state current (median current in the last 100 ms of the voltage step) was subtracted and the peak inward current in the first 100 ms was taken. Note that the current in the last 100 ms includes the leak current and any long-lasting voltage-gated currents, including NaChBac current in cases where it decays so slowly that there remains some NaChBac current in the last 100 ms; in these cases, this method likely underestimates the NaChBac current. We present current density (normalizing NaChBac currents to capacitance) because, when considering only cells with a clear NaChBac inward current, the current was moderately correlated with the capacitance ($r = -0.59$) and $y$-intercept of the best-fit line relating current to capacitance did not deviate significantly from the origin. Absolute NaChBac currents are given in the legends of Figs 1,3 and plotted in Fig. A1.

To measure fast inward currents in voltage-clamp, the final 400 ms of each current trace (which largely excludes the NaChBac current) was smoothed by a moving average of 0.8–2.2 ms (the optimal smoothing window for detecting real events and rejecting noise was determined empirically for each cell). The first derivative $dI/dt$ was taken and smoothed by the same moving average window. We used the MATLAB findpeaks function to find events where (1) the derivative $dI/dt$ showed a trough $<-2.5$ pA/ms followed by a peak $>2.5$ pA/ms within 3 ms (each with |prominence| > 5 pA/ms, to remove double peaks) and (2) the current showed a trough with prominence > 2 pA between the peak and trough of $dI/dt$. The average current waveform at each voltage step was taken by averaging 3 ms before and 6 ms after the trough of the current (without smoothing) across all events.

To measure the average depolarization of NaChBac potentials in Fig. 3H, the depolarizing current step with the biggest NaChBac potential was selected for each cell, and the plateau voltage (the mean voltage over the 200 ms period with the lowest mean voltage during the 1 s current step) was subtracted. The average depolarization was taken as the average voltage from when the normalized voltage first went above zero, to the end of the current step. This method captures both the amplitude and length of NaChBac potentials. To characterize the kinetics of NaChBac current in Fig. 3I,J, for each cell the voltage-clamp step with the biggest inward current density

was selected (excluding cells where the average current density in the first 100 ms was less than 2 p$A$/p$F$), and each trace was normalized to its peak. The decay of the current from the peak was fitted to a single exponential decay function, $I = -\exp(-t/\tau)$.

To measure voltage-gated steady-state currents, the steady-state current at each voltage step was taken as the median current in the last 100 ms of the voltage step, minus the baseline current before the voltage step. Cells were excluded from this analysis if in any current trace, the baseline before and after the voltage step differed by more than 15 pA. The leak current was estimated by extrapolating from a linear fit to the steady-state currents between −100 and −70 mV (the slopes of these linear fits well-matched the reciprocals of the input resistances calculated from current-clamp recordings; $R^2 = 0.584$). Voltage-gated steady-state currents were estimated by subtracting the leak current from the steady-state currents. Some NaChBac-expressing cells had a NaChBac current that had not yet fully decayed by the end of the 1 s voltage step (note the decay time constants in Fig. 3J). Therefore, the estimated residual NaChBac current in the last 100 ms was subtracted as follows. We assumed that the measured current $I$ is made of $I_{obs} = I_{leak} + I_K + I_{NaChBac}$, where $I_{leak}$ (the leak current) and $I_K$ (the voltage-gated outward current) are constant over the voltage step while $I_{NaChBac}$ has a peak ($I_{Np}$) early on and possibly a residual inward current ($I_{Nr}$) at the end. Note that because the peak NaChBac current was calculated by subtracting the current in the last 100 ms (i.e. $I_{obs}$), this observed peak NaChBac current $I_{Np,obs}$ actually represents $I_{Np} - I_{Nr}$. Because the voltage-gated outward current in control cells was largely absent between −60 and −40 mV (i.e. $I_K = 0$), we took the trace with the biggest NaChBac current in that range, and used $I_{NaChBac} = I_{obs} - I_{leak}$ to estimate $I_{Nr}$. To estimate $I_{Nr}$ at other voltages, we assumed that the kinetics of the NaChBac current are not dramatically altered by voltage (Ren et al., 2001). We calculated the ratio $I_{Nr}/I_{Np,obs}$ (the ratio of residual to peak NaChBac current), and applied this across the other voltage steps: for example, $I_{Nr,-30mV} = I_{Np,obs,-30mV} \times (I_{Nr,-60mV}/I_{Np,obs,-60mV})$. For the purposes of calculating $I_{Nr}$, only inward values of $I_{Np,obs}$ were used. Finally, we estimated $I_K$ as $I_{obs} - I_{leak} - I_{Nr}$. We report the voltage-gated outward current as absolute current rather than current density because the outward current was not strongly correlated with membrane capacitance ($r = 0.136$, $P = 0.410$).

The analysis code for electrophysiology data is available at https://github.com/aclinlab/greenin-whitehead-et-al.

## Olfactory learning assays

Learned odour discrimination was analysed in clear polycarbonate chambers (length 50 mm, width 5 mm, height 1.3 mm) incorporating printed circuit boards (PCBs) with 1 mm electrodes and 1 mm electrode gaps as floors and ceilings. Solid-state relays (Fairchild HSR312L) connected the PCBs to a 60 V source. For electric shock reinforcement, the relays were activated for 1.25 s every 5 s during a 1 min odour presentation. Flow-controlled (1.8 l/min; CMOSens PerformanceLine, Sensirion), filtered and humidified carrier air was mixed with flow-controlled odour streams (0.2 l/min) drawn through glass bottles filled with water or $10^{-2}$ dilutions of apple cider vinegar (Tesco) in water. The air/odour streams were split among 20 chambers, yielding a flow rate of 0.1 l/min per half-chamber. A stack of 20 chambers was backlit by 830 nm LEDs (TSHG5510, Vishay) and imaged by a AV MAKO G-234B POE camera (Allied Vision Technologies) equipped with a Kowa LM16HC lens. The apparatus was operated in a temperature-controlled incubator (PHC MIR-154-PE) maintained at 25°C. Behavioural experiments were performed during the day (10 AM–5 PM). A virtual instrument written in LabVIEW 2017 (National Instruments) extracted fly position data from video images and controlled the delivery of odours and electric shocks. Experimental data were analysed offline in MATLAB R2017b (MathWorks). Flies' preferences were quantified based on their behaviour each time they entered and then exited the choice zone (the central 20% of the chamber): exits toward the vinegar (air) were scored as decisions for (against) vinegar. Learning was quantified as the difference between the percentage of decisions for vinegar before and after training, with positive scores representing fewer decisions for vinegar after training. Flies making fewer than two entries into the choice zone during odour presentation were excluded from analysis.

## Structural imaging

Brains were fixed in 4% (wt/vol) paraformaldehyde solution prepared by 1:5 dilution of 20% paraformaldehyde (Electron Microscopy Sciences, Cat. No. 15713-S) with PBS-T (tablet made phosphate-buffered saline, Sigma P4417-100TAB, with added 0.3% Triton-X100). Following fixation, the brains were washed with PBS-T, initially with two quick washes and subsequently with three 20 min washes. The tissue was then blocked using a solution of 5% goat serum (Sigma, Cat. No. G6767) in PBS-T. Primary antibody incubation was carried out at 4°C for 2–3 nights with the following antibodies and dilutions: chicken anti-GFP (Abcam, Cat. No. ab13970, 1:2000) and rabbit anti-dsRed (TaKaRa, Cat. No. 632496, 1:250). After primary antibody incubation, the brains were washed in PBS-T (two quick washes, followed by three 20 min washes). Secondary antibody incubation was performed at 4°C for 2–3 nights using

the following antibodies and dilutions: goat anti-chicken Alexa 488 (ThermoFisher, Cat. No. A-32931, 1:1000), goat anti-rabbit Alexa 546 (ThermoFisher, Cat. No. A-11071, 1:1000), and DAPI (Merck, Cat. No. D9542, 4 ng/ml). The brains were washed in again in PBS-T (two quick washes, followed by three 20 min washes). Lastly, the samples were mounted in Vectashield mounting medium (Vector Laboratories, Cat. No. H-1000). Imaging was performed using a Nikon A1 confocal microscope located in the Wolfson Light Microscopy Facility at the University of Sheffield. Cell counts were recorded in ImageJ.

### Para-FlpTag skeletonization

Mushroom bodies expressing Para-FlpTag underwent a skeletonization process, adapted from established protocols (Amin et al., 2020). The mushroom body structure was delineated by manually tracing its outline using the dsRed signal on every third *z*-slice of the confocal image stack. A 3D mask encompassing the mushroom body volume was then generated by linearly interpolating between these traced outlines across all *z*-slices. From this volume mask, a simplified 'backbone skeleton' was derived, representing the central trajectory of the major mushroom body branches (vertical lobe, horizontal lobe, and peduncle/calyx). This backbone was constructed by manually specifying key points along these branches (e.g. the tips of the vertical and horizontal lobes, the branch junction, and the points of the curvature), subsequently connecting these points to form an undirected graph.

The four distinct branches (calyx, peduncle, horizontal lobe and vertical lobe) were identified based on the following criteria: the peduncle and the two lobes were distinguished by the junction point (defined as the sole node in the graph with three connecting edges). The division between the calyx and the peduncle was determined by creating a vector perpendicular to the visually estimated plane separating these two structures. The backbone skeleton was then segmented into calyx and peduncle at the points where this plane intersected the skeleton.

To enable comparisons of backbone structures across different flies, a 'standard' mushroom body backbone was created. This standard backbone was generated by calculating the average length of each branch (calyx, peduncle, horizontal lobe, and the vertical lobe) across the entire population of flies. Each individual fly's backbone was then normalized to this standard by positioning nodes along the backbone at intervals of $x \times 10$ μm, where $x$ represents the ratio of the individual branch length to the corresponding standard branch length. The 3D volume of each mushroom body was subsequently partitioned into Voronoi cells, with each cell's centroid corresponding to one of the evenly spaced nodes on the normalized backbone. Within each *z*-plane, a background signal, measured from a region of the brain outside the mushroom body but within the same *z*-plane, was subtracted from the signal within the mushroom body. For each node along the backbone, the background-subtracted GFP and dsRed signals were averaged across all voxels contained within its corresponding Voronoi cell. Finally, the average GFP signal was divided by the average dsRed signal to calculate the normalized Para-FlpTag signal for that node.

### Computational modelling and simulations

**Kenyon cell model generation.** A $\gamma$ Kenyon cell (cell ID: 720575940606954507) morphology was reconstructed using data from the Flywire connectome (FAFB v783), which provides electron microscopy (EM)-based neuronal reconstructions at nanometre resolution. Morphological data was obtained from the EM surface reconstruction and then converted into a skeletonized format suitable for compartmental modelling utilizing a previously established methodology (Moreno-Sanchez et al., 2024). Following skeletonization, manual proofreading was conducted to correct errors inherent to automated skeletonization processes, particularly in the soma's representation and excessive branching. The soma was reconstructed using the average radius of the original EM surface reconstruction. Erroneous branches and zero-radius nodes were removed, and the final skeleton model was overlaid onto the original mesh to verify accuracy. Surface area comparisons between the reconstructed models and the EM surface reconstruction further confirmed accuracy of skeletonization. The finalized skeleton model was prepared for simulation in NEURON (Hines & Carnevale, 2001). The skeleton morphology was imported into neuTube (Feng et al., 2015), where individual nodes were assigned compartmental labels (soma = 1, axon = 2, dendrites = 3, tether = 11, pre-SIZ = 12, SIZ = 13). Each section was categorized according to its anatomical features to support accurate compartmentalized biophysical modelling. To align with the NEURON simulation environment, morphological coordinates were converted to micrometers for computational modelling. This methodology ensured that the Kenyon cell model maintained morphological accuracy while being computationally efficient for simulations and ion channel distributions.

**Model initialization.** The $\gamma$ Kenyon cell model was implemented in the NEURON simulation environment using the skeletonized morphology. Passive properties were assigned uniformly across all compartments, and the length constant ($\lambda$) was calculated for each section based on its geometry and passive properties. To achieve proper

spatial resolution, sections longer than 0.1 λ were subdivided, ensuring no individual segment exceeded this length. The membrane potential was initialized for each individual simulation corresponding to its experimental counterpart, while the electrode placement and recording section (Soma) were the same for each simulation.

**Optimization of passive properties.** The passive properties, including capacitance ($C_m$), leak conductance ($g_{leak}$), axial resistivity ($R_a$) and reversal potential ($E_{rev}$), were optimized by fitting the model responses to hyperpolarizing current injections. A baseline subtraction was applied to the experimental data using an average of the first 200 ms of each recording to account for variability in the resting membrane potential during current step protocols. This approach allowed for simultaneous fitting of multiple current injections. Experimental parameter ranges from published models were used to constrain biologically plausible values for axial resistivity ($R_a$ = 30–400 Ω*cm) and capacitance ($C_m$ = 0.6–2.6 uF/cm$^2$) (Borst & Haag, 1996; Gouwens & Wilson, 2009).

An electrode was modelled as a cylindrical section with a length of 10 μm, a diameter of 1 μm, and passive properties set to approximate experimental conditions for: axial resistivity ($R_a$ = 205.513 Ω*cm), leak conductance ($g_{leak}$ = 3.978e–4 S/cm$^2$), and capacitance ($C_m$ = 6.4 uF/cm$^2$), corresponding to an electrode resistance of 13 MΩ, seal resistance of 8 GΩ and capacitance of 2 pF.

The biophysical parameters for the wildtype (WT) Kenyon cell models, $g_{leak}$ = 1.10e-4–1.115e-4 S/cm$^2$, $E_{rev}$ = −(71.25–80.25) mV, $R_a$ = 100–128 Ω*cm, $C_m$ = 2 uF/cm$^2$ and NaChBac-expressing Kenyon cell models $g_{leak}$ = 3.8755e-5–2.3215e-3 S/cm$^2$, $E_{rev}$ = −(62.5–88) mV, $R_a$ = 70–125 Ω*cm, $C_m$ = 1–2 uF/cm$^2$, were determined through manual adjustments within physiological constraints (Borst & Haag, 1996; Gouwens & Wilson, 2009), minimizing the squared difference between individual experimental current traces and the simulated trace to optimize model passive parameter sets.

**Ion channel parameters.** All ion channels were modelled using Hodgkin–Huxley type equations

$$I = g_{max} \cdot m^p \cdot h \cdot (V_m - E) \tag{2}$$

where $g_{max}$ represents the maximal conductance, $m$ and $h$ are the activation and inactivation variables, respectively, $p$ is the power of the activation gate, $V_m$ is the membrane potential, and $E$ is the reversal potential of the ion conducted through the channel. The activation ($m$) and inactivation ($h$) variables follow the differential equation

$$\frac{dx}{dt} = \frac{(x_\infty (V_m) - x)}{\tau_x (V_m)} \tag{3}$$

where $x$ is replaced by $m$ or $h$ accordingly. The voltage-dependent steady-state activation and inactivation gate functions are described using a Boltzmann function:

$$x_\infty (V_m) = \frac{1}{1 + exp \left( \frac{\left( V_m - V_{\frac{1}{2}x} \right)}{k_x} \right)} \tag{4}$$

where $V_{\frac{1}{2}x}$ is the activation half-voltage and $k_x$ is the slope factor (both in mV).

Ion channel parameters for the transient and persistent sodium channel and delayed rectifying potassium channels were used from previously published ion channel models described by the equations above (Günay et al., 2015). Parameters for the activation and inactivation steady-state kinetics, time constants and current for the NaChBac channel model were determined from previous whole-cell patch-clamp recordings from HEK293 cells transfected with NaChBac (Strege et al., 2023). The established ion channel model was compared using a simple single compartment model recreating similar voltage-clamp protocols compared with the voltage-clamp recordings of transfected HEK293 cells (Strege et al 2023). The voltage-dependence of the inactivation and activation time constants were fit using a sigmoidal function. The specific parameters for $x_\infty (V_m)$ and the time constant $\tau_x (V_m)$ for NaChBac are listed in Table 1.

The reversal potentials for the transient and persistent sodium channels as well as NaChBac were set to 60 mV, while the reversal potential for the delayed rectifier potassium channel was set to –80 mV.

Active properties for individual ion channels were manually adjusted to qualitatively recreate their experimental counterparts. Table 2 shows the compartment-specific maximal conductances ($g_{max}$) for each model, including the transient sodium channel (NaT), persistent sodium channel (NaP), the delayed rectifying potassium channel (K), and NaChBac in the soma, spike initiation zone (SIZ) and everywhere throughout the model. All values are in mS/cm$^2$. Stars indicate models used in Fig. 9C,H and A8 (WT), and in Fig. 9E,I (NaChBac).

**Simulations.** All simulations were run using Python 3.10.12 and the NEURON (8.2.0+) simulation environment (Hines & Carnevale, 2001). Simulations were executed on the Windows Subsystem for Linux, using Ubuntu 22.04 LTS, to leverage CoreNEURON (Kumbhar et al., 2019).

Simulations aimed to examine the firing responses of the γ Kenyon cell model subjected to current injection protocols. The simulated current-clamp protocol followed the experimental protocol, except that for each individual current injection simulation, the model was

**Table 1. Voltage-dependence parameters for NaChBac channels**

| | Parameters | Values |
|---|---|---|
| Activation | $p$ | 3 |
| | $V_{\frac{1}{2}m}$ | $-47.21$ mV |
| | $k_m$ | 8.17 mV |
| | $\tau_m$ (ms) | $125,258.35 - \frac{125,253.86}{(1+exp\,(-0.06*(V_m+187.78)))}$ |
| Inactivation | $V_{\frac{1}{2}h}$ | $-56.68$ mV |
| | $k_h$ | $-6.08$ mV |
| | $\tau_h$ (ms) | $165.14 + \frac{36.12}{(1+exp\,((-24.60-V_m)/-5.158))}$ |

**Table 2. Maximal conductances in mS/cm$^2$**

| Type | NaT SIZ | NaT Axon | NaP SIZ | K SIZ | K Axon | NaChBac Soma | NaChBac SIZ | NaChBac all |
|---|---|---|---|---|---|---|---|---|
| WT | 88 | 66 | 0.414 | 800 | 60 | 0 | 0 | 0 |
| WT* | 40 | 30 | 0.18 | 200 | 15 | 0 | 0 | 0 |
| WT | 80 | 60 | 0.36 | 530 | 39.75 | 0 | 0 | 0 |
| NaChBac* | 25 | 18.75 | 0.18 | 200 | 15 | 15 | 0.3125 | 0.05 |
| NaChBac | 5.714 | 4.286 | 0.018 | 400 | 30 | 0.55 | 0.00833 | 0.00833 |
| NaChBac | 8 | 6 | 0.09 | 500 | 37.5 | 2 | 0.15 | 0.05 |

reinitialized at the starting membrane potential. Current injections into the soma of the model neuron were implemented using the IClamp process. The amplitude, duration, and delay of the current injections were set to reproduce the experimental data. Multiple current amplitudes were tested, and the corresponding membrane potential responses were recorded. These traces were used to assess neuronal firing characteristics such as spike count and firing frequency.

All simulation code is available at https://github.com/AusbornLab/KC_NaChBac_Model.

**Spike detection and analysis.** Spike detection was performed by identifying local peaks in the membrane potential trace that exceeded a predefined threshold (spike amplitude threshold was set to 1 mV). The number of spikes and the firing frequency were calculated for each current condition, which were then compared across different simulations. A relative threshold approach was used to avoid false positives from early transient responses in the NaChBac condition. Manual inspection of the detected spikes was then done to verify accuracy in detection.

## Statistics

Statistical analysis was performed using SciPy (Virtanen et al., 2020) for the modelling or MATLAB (MathWorks) or Prism 10 (GraphPad) for all other experiments. The Mann–Whitney or Kruskal–Wallis tests were performed when the normality assumption was violated (assessed by the Shapiro–Wilk test). Sample sizes were not pre-determined. Error bars in graphs are as described in the legends. Details of statistical tests are given in Table A3.

## Results

### NaChBac expression decreases Kenyon cell excitability and odour responses

Given past reports that NaChBac expression increases neuronal excitability, we asked whether the same was true in Kenyon cells. We started with patch-clamp electrophysiology. Unlike endogenous sodium channels, which are localized to Kenyon cell axons (Ravenscroft et al., 2020), NaChBac is expressed throughout Kenyon cells' dendrites, axons and somata, as shown by expression of NaChBac-GFP in all Kenyon cells (Fig. 1A). The somatic expression of NaChBac allowed us to characterize NaChBac's properties and effects at the soma by patch-clamp recordings (Fig. 1B,C).

There are three major subtypes of Kenyon cells ($\alpha'\beta'$, $\alpha\beta$ and $\gamma$), which have slightly different electrical properties (Groschner et al., 2018; Inada et al., 2017; Turner et al., 2008). To prevent inter-subtype heterogeneity from increasing variability in our data, we initially focused on $\gamma$ Kenyon cells, which are crucial for olfactory associative learning (Guven-Ozkan & Davis,

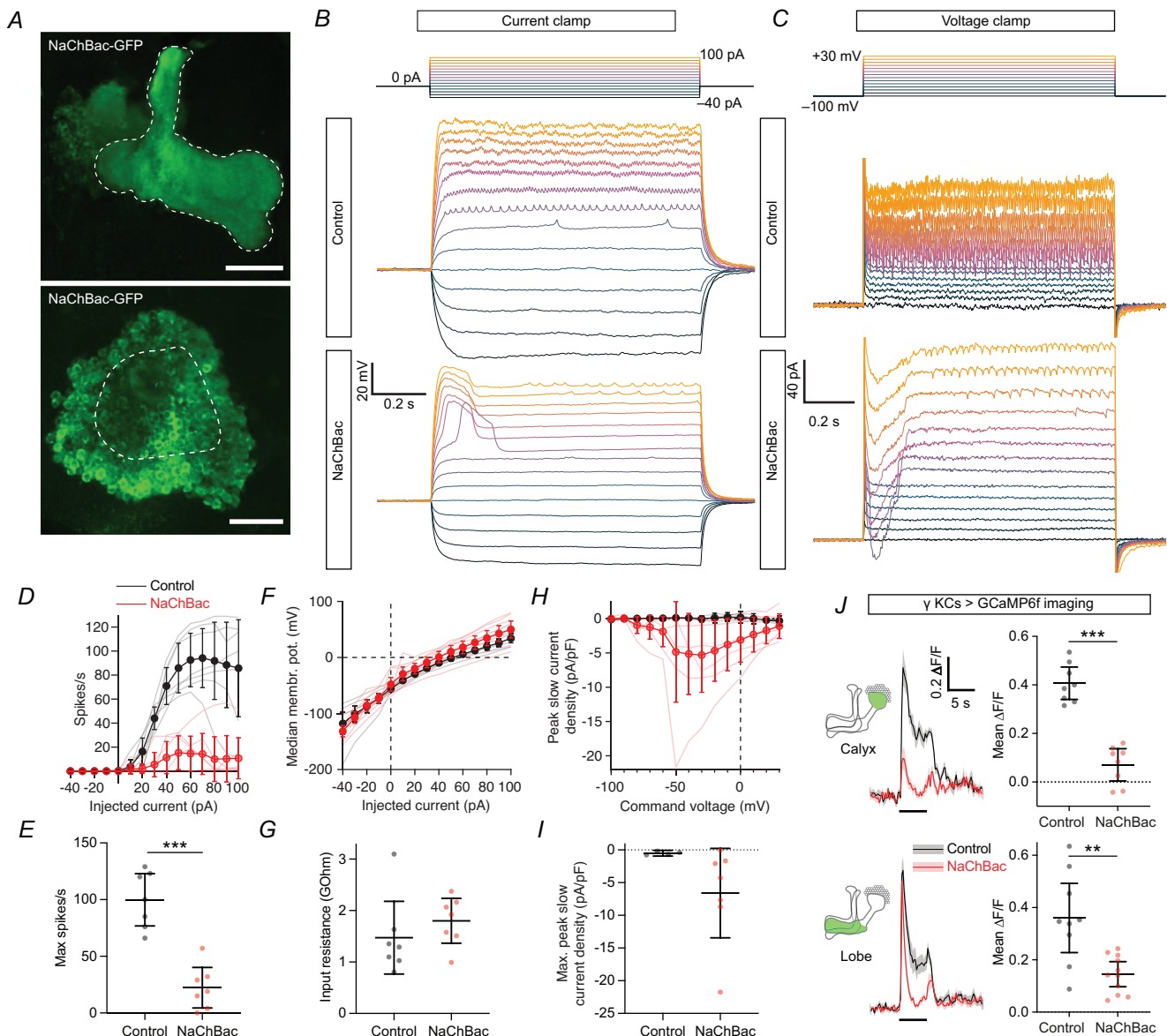

**Figure 1. NaChBac expression decreases Kenyon cell excitability and odour responses**

*A*, maximum intensity Z-projections of confocal stacks of NaChBac-GFP expression in Kenyon cells driven by R13F02-GAL4, through the whole mushroom body (upper) or just the calyx and cell bodies (lower). Scale bar: 50 μm. Dashed lines outline the lobes (upper) and calyx (lower). *B*, current-clamp traces (baseline-subtracted) of $\gamma$ Kenyon cells with 1 s current injection from −40 to +100 pA with 10 pA intervals, with or without NaChBac expression. The example cells shown here and in panel *C* have the following estimated parameters: membrane capacitance, 17.2 pF (control), 18.3 pF (NaChBac); input resistance, 1.36 GΩ (control), 0.99 GΩ (NaChBac). *C*, voltage-clamp traces (baseline-subtracted, but not leak-subtracted) of $\gamma$ Kenyon cells with 1 s voltage steps starting at −100 mV going to −100 to +30 mV with 10 mV intervals, with or without NaChBac expression. The capacitive transients have been truncated. *D*, spike rate at each level of current injected. Control, $n = 7$ (black); NaChBac, $n = 7$ (red). *E*, peak spike rate for each cell. *F*, median membrane potential during the current pulse at each level of current injected (excluding the first and last 50 ms). *G*, input resistance for each cell. *H*, peak slow inward current density in the first 100 ms of each voltage step. The steady-state current was subtracted, and the traces were smoothed by a 50 ms moving median to remove the fast inward currents. Control, $n = 4$; NaChBac, $n = 7$. *I*, maximum peak inward current density across voltages from (*H*). Absolute peak current amplitudes were (mean ± s.d.): control, −7.0 ± 5.0 pA; NaChBac, −124.0 ± 162.6 pA. Membrane capacitances were (mean ± s.d.), control, 14.7 ± 4.4 pF; NaChBac, 16.2 ± 3.6 pF. *J*, GCaMP6f responses of $\gamma$ Kenyon cells to 5 s pulses of isoamyl acetate (black bar), quantified at right as the mean $\Delta F/F$ during the odour. Top panels: Calyx responses. Control, n given as number of hemispheres (number of brains) = 8(8); NaChBac, $n = 8(8)$. Bottom panels: Lobe responses. Control, $n = 9(8)$; NaChBac, $n = 11(8)$. Traces are means ± SEM. In (*B–J*), $\gamma$ Kenyon cells were labelled with

GFP or GCaMP6f using the mb247-GAL4 driver, with expression of lexAop-GAL80 to restrict expression to the γ lobe driven by R44E04-LexA. All graphs show mean ± 95% confidence interval. **$P < 0.01$, ***$P < 0.001$, Mann–Whitney test. See Table A3 for details of statistical tests.

2014). We expressed GFP and NaChBac in γ Kenyon cells using mb247-GAL4 (labels αβ and γ Kenyon cells) and R44E04-LexA driving lexAop-GAL80 (to exclude αβ Kenyon cells) (Bielopolski et al., 2019). We carried out whole-cell patch-clamp recordings and injected current steps and measured the resulting voltage changes and action potentials.

Surprisingly, NaChBac expression drastically suppressed action potential firing in γ Kenyon cells (Fig. 1B,D,E). The decreased excitability was specific to action potential firing, as the input resistance was normal in NaChBac-expressing cells (i.e. current injections produced similar magnitude voltage deflections in both normal and NaChBac-expressing cells) (Fig. 1F,G). 5/7 NaChBac-expressing cells showed prolonged (>100 ms) but transient depolarizations consistent with previously reported effects of NaChBac (Giachello et al., 2022; Sheeba, Fogle et al., 2008; Sheeba, Gu et al., 2008; Sim et al., 2013; Xue et al., 2014; Zhang & Gaudry, 2016). Depending on the cell, these depolarizations, which we call 'NaChBac potentials', could occur with a depolarizing current step and/or with the end of a hyperpolarizing current step. None of the control cells (0/7) showed slow depolarizations. When recorded in voltage-clamp mode, 5/7 NaChBac-expressing cells showed large depolarization-induced inward currents that lasted for >100 ms (Fig. 1C), consistent with previously described kinetics of NaChBac (Ren et al., 2001). None of the control cells (0/4 voltage-clamp recordings) showed slow inward currents. The peak current occurred around −60 to −30 mV (Fig. 1H,I), similar to the reported peak around −50 mV in *Drosophila* LNv neurons (Sheeba, Gu et al., 2008). The variability in recorded currents and depolarization may partially reflect variability in input resistance (Fig. A1) and/or NaChBac expression levels (Fig. A2). Because NaChBac current amplitude was correlated with the measured input resistance, we excluded cells with input resistance below 0.8 GΩ, as we tended to observe small/no NaChBac currents in these cells. However, we obtained the same results when including all recordings; in particular, the difference in spiking between control and NaChBac-expressing cells was consistent across the whole range of input resistances (Figs A1 and A3).

Somatic recordings may not necessarily capture all electrical activity in Kenyon cells due to space-clamp problems (Bar-Yehuda & Korngreen, 2008; Gouwens & Wilson, 2009). Moreover, recording odour responses by electrophysiology is challenging as Kenyon cells' odour responses are sparse (Honegger et al., 2011; Turner et al., 2008). Therefore, we measured odour responses using calcium imaging with GCaMP6f, which allows direct measurement from Kenyon cell dendrites and axons. Consistent with our electrophysiological results, odour responses were much smaller in NaChBac-expressing Kenyon cells than control Kenyon cells, in both the calyx and lobes (Fig. 1J). These results suggest that NaChBac expression in γ Kenyon cells decreases their activity.

We extended these results by expressing NaChBac in all Kenyon cell subtypes using the pan-Kenyon cell driver R13F02-GAL4 (Jenett et al., 2012), which is expressed in approximately all Kenyon cells (both R13F02-GAL4 and OK107-GAL4 labelled ∼2000 Kenyon cells per hemisphere, Fig. A4). We recorded responses in the calyx and the five axonal lobes (α, α′, β, β′, γ; αβ Kenyon cell axons split into two branches to form the α and β lobes; likewise α′β′ Kenyon cell axons form the α′ and β′ lobes). Again, NaChBac expression decreased odour responses in all areas imaged (Fig. 2A). A mixed-effects model revealed a significant main effect of genotype (control *vs.* NaChBac). Although not all individual lobes showed a significant difference in *post hoc* tests, there was no significant interaction between genotype and area imaged (Table A3). These results suggest that NaChBac non-specifically decreases activity across all Kenyon cell types.

### NaChBac expression decreases odour-evoked calcium influx in Kenyon cells only when expressed during development or for a short time in adults

We next asked whether NaChBac's effect on Kenyon cell activity was developmental. To test this, we used two approaches to suppress NaChBac expression during development: tubP-GAL80[ts] (McGuire et al., 2003) and the auxin-inducible gene expression system (AGES) (McClure et al., 2022). We added GAL80[ts] to the R13F02-GAL4 > UAS-NaChBac genotype, and we raised flies at 18°C to block NaChBac expression and heated flies to 31°C to disrupt the GAL80[ts] for 2 or 4 days after eclosion (Fig. 2B). For AGES, we added the transgene AtTIR1-T2A-AID-GAL80-AID, which produces two proteins: TIR1, an auxin receptor, and AID-GAL80, which is GAL80 fused to an auxin-inducible degron (AID). We put flies on food with 10 mM auxin for 2 or 4 days to induce degradation of GAL80 and hence allow expression of NaChBac (Fig. 2C). We confirmed that GAL80[ts] and AGES expression effectively suppressed GAL4 activity during development by quantifying GCaMP6f fluorescence driven by R13F02-GAL4 (Fig. A5).

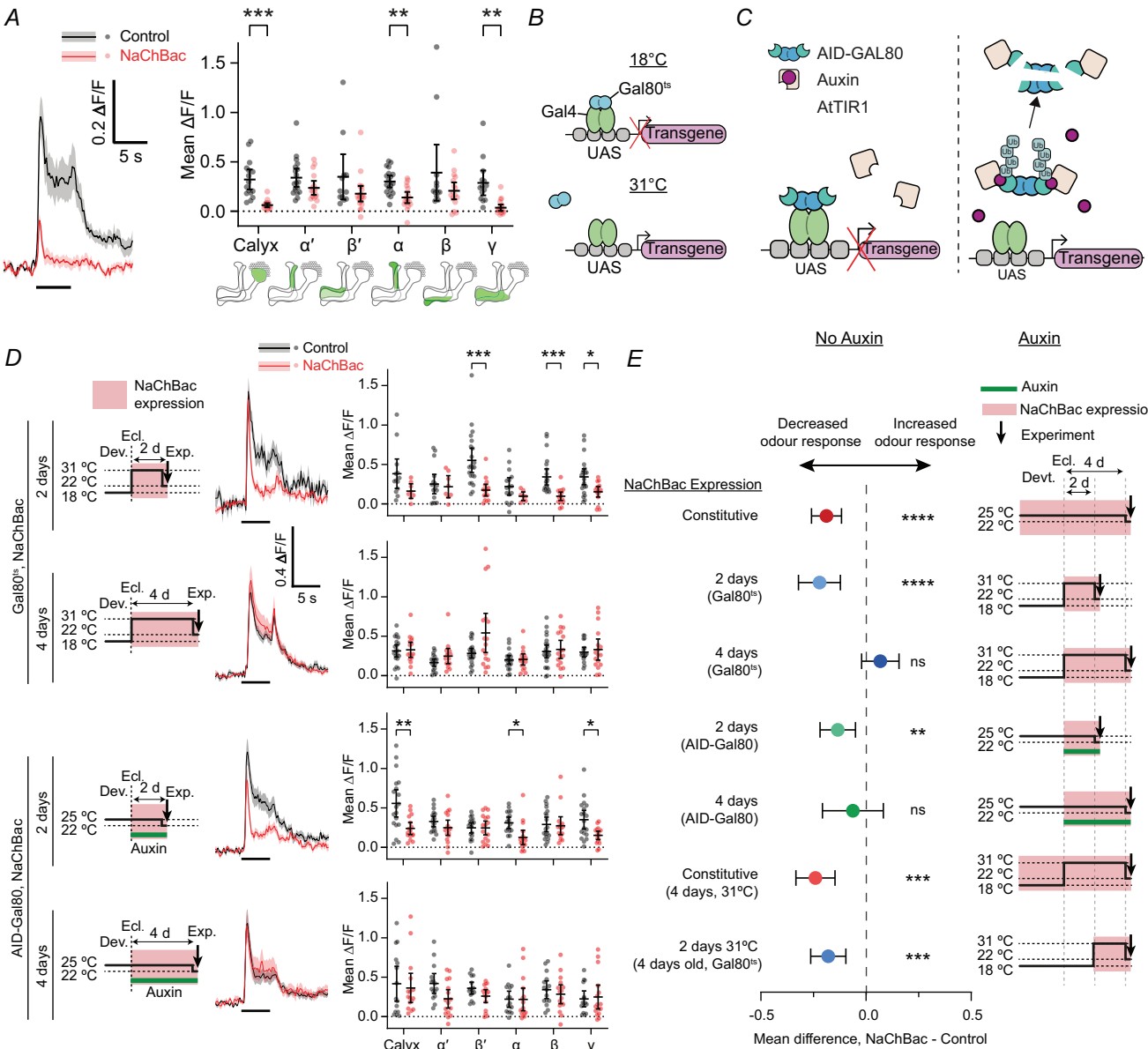

**Figure 2. Odour-evoked calcium influx is reduced by constitutive or 2-d, but not 4-d, NaChBac expression**
*A*, constitutive expression of NaChBac reduces odour-evoked calcium influx in Kenyon cells; all Kenyon cells labelled with GCaMP6f driven by R13F02-GAL4. Left: $\gamma$ lobe response to 5 s isoamyl acetate (black bar). Right: odour responses for all areas quantified as mean $\Delta F/F$ during the odour presentation. Diagrams depict the locations analysed for the calyx and each axonal lobe. Control, $n = 16(11)$, 19(11), 12(7), 19(11), 13(7), 14(7); NaChBac, $n = 19(10)$, 17(10), 21(13), 18(10), 17(13), 20(13). *B*, schematic of temperature-sensitive regulation by GAL80[ts]. At 18°C, GAL80[ts] blocks GAL4 activity, but at 31°C, GAL80[ts] is inactivated, releasing GAL4 to drive transcription of the target transgene. *C*, schematic of the auxin-inducible gene expression system (AGES). Without auxin present (left), AID-tagged GAL80 blocks GAL4 activity. With auxin present (right), AID-tagged GAL80 is targeted for degradation through ubiquitination (Ub), releasing GAL4 to drive transcription of the target transgene. *D*, calcium responses in all Kenyon cells with GCaMP6f expression driven by R13F02-GAL4. Cells conditionally expressed NaChBac for either 2 or 4 days using GAL80[ts] (top two rows) or auxin-inducible degradation of AID-GAL80 (bottom two rows). Left panels: Schematics of the experimental timeline, indicating housing temperature (for GAL80ts) or auxin presence (for AID-GAL80), and the time of imaging (arrow). Centre panels: $\gamma$ lobe responses to 5 s isoamyl acetate (black bar) (see Fig. A6 for all lobes). Right panels: odour responses for all areas quantified as mean $\Delta F/F$ during the odour presentation. 2 d GAL80[ts], Control $n = 13(10)$, 16(10), 22(13), 16(10), 22(13), 22(13), NaChBac $n = 8(5)$, 8(5), 16(12), 8(5), 16(12), 16(12). 4 d GAL80[ts], Control $n = 21(12)$, 20(12), 22(13), 20(12), 23(13), 23(13), NaChBac $N = 16(8)$, 15(8), 17(9), 16(8), 17(9), 17(9). 2 d AID-GAL80, Control $n = 19(11)$, 19(11), 19(11), 18(11), 19(11), 19(11), NaChBac $n = 16(12)$, 17(12), 17(12), 17(12), 17(12), 17(12). 4 d AID-GAL80, Control $n = 15(9)$, 15(9), 15(9), 15(9), 15(9), 15(9), NaChBac $n = 16(9)$, 16(9), 16(9), 16(9), 16(9), 16(9). *E*, summary of

the main effect of genotype (control *vs.* NaChBac) from mixed-effects model analysis of panel *D* and Fig. A6. The graph shows the mean difference in odour response between control and NaChBac flies across all lobes (negative means NaChBac responses were smaller), with 95% confidence interval, in the different timings and methods of NaChBac expression described by the labels at left and depicted in the schematics at right. Graphs show means $\pm$ 95% confidence intervals. *$P < 0.05$, **$P < 0.01$, ***$P < 0.001$, mixed-effects analysis with Geisser-Greenhouse correction and Šidák's multiple comparison test. Odour evoked calcium traces are means $\pm$ SEM. See Table A3 for details of statistical tests.

Two days (d) adult-only expression of NaChBac had a similar effect as constitutive expression: Kenyon cell odour responses were decreased (Fig. 2D). Whether using GAL80^ts or AGES, there was a significant main effect of genotype on odour responses, although the effect was not individually significant in every lobe (Table A3). However, surprisingly, adult-only NaChBac expression for 4 days had no significant effect on Kenyon cell odour responses and in some cases even slightly increased them (Fig. 2D). Overall, mixed-effects models showed no significant main effect of genotype for 4 d expression, whether using GAL80^ts or AGES. We also tested 8 days NaChBac expression using GAL80^ts and saw no significant effect (Fig. A6). The difference between 2 *versus* 4 d adult-only expression was not due to age, because when we recorded from 4 day-old flies where NaChBac was expressed for only 2 days before imaging (controlled by GAL80^ts), these flies also showed decreased Kenyon cell odour responses (Figs 2E and A6). We can exclude the possibility that the different effect from 4 d expression arose from the temperature protocol *per se* rather than the different timing of NaChBac expression for two reasons. First, we observed the same effect with AGES, which does not involve any temperature shift. Second, when we expressed NaChBac constitutively, raised the flies at 18°C and heated them to 31°C for 4 d after eclosion, this manipulation also decreased Kenyon cell odour responses (Figs 2E and A6).

These results suggest that although adult-only NaChBac expression for a short time decreases odour-evoked calcium influx in Kenyon cell dendrites and axons, this effect disappears when NaChBac is expressed for a longer time, likely through homeostatic compensation. Why doesn't compensation also cancel the effect of constitutive NaChBac expression? It may be that developmental NaChBac expression, perhaps during a critical period (Giachello & Baines, 2015; Lowe et al., 2024), 'locks in' the effect such that it cannot be compensated away, or it fixes a lower 'set-point' for Kenyon cell calcium influx (see Discussion).

### Adult-only NaChBac expression decreases Kenyon cell excitability

One potential explanation for normal odour-evoked calcium influx after 4 d adult-only NaChBac expression could be that Kenyon cells' intrinsic excitability recovered to normal levels. To test this possibility, we returned to patch-clamp electrophysiology. Because the excitability phenotype from NaChBac in Fig. 1 was so drastic, and we did not observe consistent differences in how odour-evoked calcium influx in different Kenyon cell subtypes is affected by NaChBac (Fig. 2), we used R13F02-GAL4 to drive NaChBac in all Kenyon cells, reasoning that any differences between Kenyon cell subtypes would be overwhelmed by the effect of the NaChBac. Using R13F02 to drive NaChBac also kept these experiments consistent with the adult-only NaChBac expression with calcium imaging in Fig. 2. We induced adult-only expression of NaChBac using AGES. In contrast to odour-evoked calcium influx in Kenyon cell dendrites/axons, both 2- and 4-d NaChBac expression strongly decreased action potentials induced by current injection (Fig. 3A,B). As with constitutive expression, 2-d and 4-d NaChBac expression did not affect Kenyon cells' input resistance (Fig. 3C,D), and both conditions showed strong depolarization-induced inward currents as with constitutive NaChBac expression (Fig. 3E,F, compare to Fig. 1H,I; 8/8 cells showed inward currents at 2 d and 7/7 at 4 d). Kenyon cells expressing NaChBac for 2 d or 4 d both showed NaChBac potentials similar to those seen in constitutive Kenyon cells (9/9 2-d cells and 9/9 4-d cells showed NaChBac potentials). Similarly, there was no significant difference in the amount of current needed to trigger NaChBac potentials (Fig. 3G), or in their overall size (Fig. 3H), nor in the kinetics of NaChBac currents (Fig. 3I,J).

Although the NaChBac current and NaChBac's effect on spike rate appeared slightly smaller at 4 d than at 2 d, these differences were not statistically significant. Because cells with stronger NaChBac current density somewhat tended to have higher input resistance ($r = 0.34$) and lower peak spike rate ($r = -0.21$), we asked whether a more subtle difference between 2 d and 4 d might appear when taking into account the covariance of these three parameters. No differences between 2 d and 4 d were apparent when plotting these variables against each other or when normalizing NaChBac current to input resistance (Fig. A7). In addition, principal components analysis using these three parameters (NaChBac current, input resistance, spike rate) did not reveal significant differences between 2 d and 4 d along any of the principal components (Fig. A7). While we do not exclude the possibility that a larger sample size might reveal a significant but small

difference between 2-d *versus* 4-d NaChBac expression, our results show that both manipulations strongly reduce action potentials.

## NaChBac expression makes Kenyon cell odour responses more sparse

If Kenyon cells expressing NaChBac fire almost no normal spikes, why do they show any odour-evoked calcium influx in their axons at all? Presumably, the odour-evoked calcium influx comes from voltage-gated calcium channels (VGCCs) opened by NaChBac potentials, which would be triggered when odour-evoked synaptic depolarization passes NaChBac's opening threshold. Note that while axonal calcium influx is likely entirely from VGCCs, as Kenyon cell axons do not respond to locally applied nicotinic agonists (Barnstedt et al., 2016), dendritic odour-evoked calcium influx

is likely a mixture of direct influx through nicotinic receptors and depolarization-triggered influx through VGCCs (Amin et al., 2020; Gruntman & Turner, 2013; Li et al., 2013; Oertner et al., 2001). Why is axonal/dendritic odour-evoked calcium influx reduced in NaChBac-expressing Kenyon cells? This might be because of reduced voltage-gated calcium conductance and/or because compared with normal spikes, NaChBac potentials are less easily triggered by odour input (whether because of the intrinsic nature of NaChBac, or because there is less synaptic excitation or more inhibition onto NaChBac-expressing Kenyon cells). If the reduction is purely from reduced calcium conductance, the same fraction of cells should respond to odours in control and NaChBac-expressing Kenyon cells (but with lower amplitude response per cell). Conversely, if NaChBac is less likely to respond, then fewer NaChBac-expressing cells should respond.

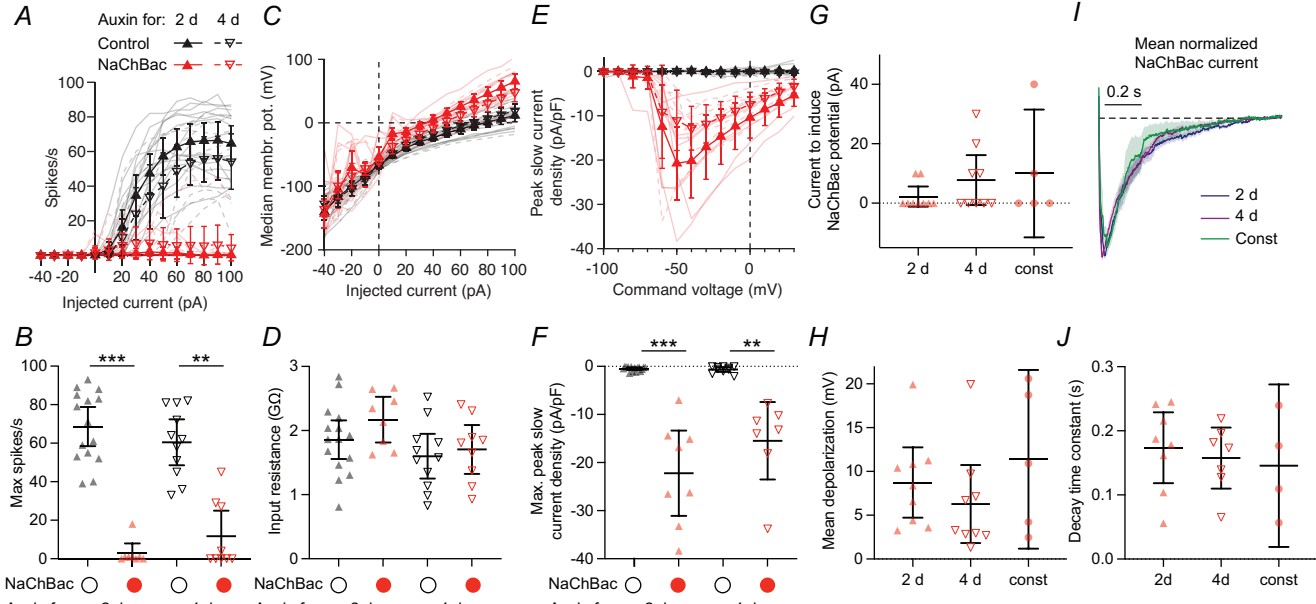

**Figure 3. Adult-only NaChBac expression decreases Kenyon cell excitability**
*A*, spike rate at each level of current injected, in control Kenyon cells (black) or Kenyon cells expressing NaChBac (red) for 2 d (upward-pointing triangles) or 4 d (downward-pointing triangles). *n* = 15, 9, 11, 9. *B*, peak spike rate for each cell. *C*, median membrane potential during the current pulse at each level of current injected (excluding the first and last 50 ms). *n* = 15, 9, 11, 9. *D*, input resistance for each cell. *n* = 15, 8, 11, 9. *E*, peak inward current density at each voltage step (subtracted steady-state current). *F*, maximum peak inward current density across voltages from (*E*). Absolute peak current amplitudes were (means ± s.d.): control 2 d, −7.3 ± 6.6 pA; NaChBac 2 d, −294.6 ± 180.0 pA; control 4 d, −7.3 ± 9.0 pA; NaChBac 4 d, −221.0 ± 123.3 pA; membrane capacitances were (means ± s.d.):, control 2 d, 19.3 ± 9.7 pF; NaChBac 2 d, 13.6 ± 5.4 pF; control 4 d, 13.0 ± 5.1 pF; NaChBac 4 d, 13.9 ± 3.4 pF. *G*, current injection required to induce a NaChBac potential. If the required current is 0, that means that NaChBac potentials occurred at the end of a hyperpolarizing current injection. Cells that did not show NaChBac potentials are not included. *n* = 9, 9, 5. *H*, average depolarization for the biggest NaChBac potential for each cell, over the full 1 s current injection (see Methods). Cells that did not show NaChBac potentials are not included. *n* = 9, 9, 5. *I*, mean normalized NaChBac current in Kenyon cells expressing NaChBac for 2 d (blue), 4 d (purple), or constitutively (green) (see Methods). Traces are means ± SEM. *J*, decay time constants for NaChBac currents (fitted to a single exponential decay function). *n* = 8, 7, 4. Graphs show means ± 95% confidence interval. **\**P* < 0.01, **\*\**P* < 0.001, Kruskal–Wallis ANOVA with Dunn's multiple comparisons test. See Table A3 for details of statistical tests.

Kenyon cell dendrites and axons are packed together too tightly to distinguish individual Kenyon cells' odour responses by calcium imaging. Therefore, we recorded odour-evoked calcium influx in Kenyon cell somata. We calculated the population sparseness of these responses without applying any threshold to the responses to prevent weak responses from being truncated, so that changes in sparseness would reflect the fraction of cells responding rather than a uniform change in response amplitude. The population sparseness of the responses was increased by constitutive and 2-d NaChBac expression (i.e. fewer cells responded) (Fig. 4A–C). This result indicates that each cell is less likely to fire a NaChBac potential to odour input than a control cell is to fire normal spikes, though it does not rule out reduced calcium influx in NaChBac-expressing cells.

We next addressed the contrast between the electrophysiology and calcium imaging results: whereas both 2-d and 4-d NaChBac-expressing Kenyon cells show a similar drastic reduction in normal spikes compared with control cells, odour-evoked calcium influx in dendrites/axons is reduced only with 2-d, not 4-d, NaChBac expression. As above, this difference between 2 d and 4 d might be because each NaChBac potential triggers more axonal/dendritic calcium influx at 4 d than 2 d. In this case, the fraction of responsive Kenyon cells would be the same, i.e. reduced compared with controls in both cases. Conversely, it might be that NaChBac potentials are more easily triggered by odour at 4 d than 2 d (e.g. because of increased synaptic excitation at 4 d). In this case, more cells would respond at 4 d than 2 d, or in other words, the increase in sparseness caused by NaChBac at 2 d would not occur at 4 d. However, 4-d NaChBac expression caused the same increase in population sparseness as constitutive/2-d expression, whether induced by GAL80[ts] or AGES (Fig. 4A–C). These results suggest that the different effects of NaChBac on odour-evoked calcium influx at 2 *versus* 4 d can be best explained by a compensatory increase in voltage-gated calcium influx at 4 d.

The sparseness result alone is not definitive. It could be that compensation after 4 d actually did increase excitatory synaptic input (not calcium influx), but contrary to our prediction above, this did not decrease sparseness because cells vary in NaChBac expression levels and only a small minority of cells express enough NaChBac to be triggered by odour-evoked synaptic input. However, two findings argue against this scenario. First, this scenario predicts that population responses to different odours would be more correlated in NaChBac flies than control flies, as odour responses would be restricted to a small minority of cells. Indeed, we previously showed computationally that variability in excitability between Kenyon cells decreases the

angular separation of population odour responses (Abdelrahman et al., 2021), similar to increasing the inter-odour correlation. However, contrary to this prediction, NaChBac expression decreased inter-odour correlations, whether with constitutive expression or for 2 or 4 days (Fig. 4D,E).

Second, this scenario predicts that while the same fraction of Kenyon cells responds to odours after 4 d than after 2 d NaChBac expression, they would respond more strongly. In this case, NaChBac should decrease the overall amplitude of Kenyon cell somatic odour responses (averaged across all cells, including non-responsive cells) at 2 d, but not at 4 d. Instead, overall somatic responses were decreased in all NaChBac conditions: constitutive, 2 d and 4 d (Fig. 4F,G).

Together, these results suggest that the recovery of odour-evoked calcium influx in axons/dendrites at 4 d (*vs.* 2 d) results from increased voltage-gated calcium influx, not more NaChBac potentials (see Discussion). The fact that odour-evoked calcium influx recovers at 4 d in axons/dendrites but not somata suggests that this increase in calcium influx is restricted to axons and dendrites, not the cell body.

## NaChBac expression in Kenyon cells prevents olfactory learning

Given that 4 days NaChBac expression decreases Kenyon cell excitability and Kenyon cell somatic responses, yet leaves odour-evoked calcium influx in Kenyon cell axons/dendrites normal, we asked how olfactory learning is affected by NaChBac expression. Olfactory associative memory requires Kenyon cell synaptic output (Schwaerzel et al., 2002), and decreasing Kenyon cell excitability can impair olfactory memory performance (DasGupta et al., 2014; Groschner et al., 2018). However, Kenyon cell activity and memory performance are not straightforwardly linked: manipulations that increase Kenyon cell activity can affect memory performance in both directions, improving it (Liu et al., 2009; Wong et al., 2023) or worsening it (Bielopolski et al., 2019; Lin et al., 2014; Manoim et al., 2022). Given that NaChBac decreases Kenyon cell spiking so severely, we hypothesized that it would worsen memory performance. To examine only the efficiency of the learning process, we used a task that does not involve odour discrimination. In particular, we tested learning in an aversive conditioning assay where 1:100 apple cider vinegar was paired with 60 V electric shock, so that flies would learn to avoid this innately attractive odour (see protocol diagrams in Fig. 5).

Indeed, flies expressing NaChBac in Kenyon cells driven by R13F02-GAL4 performed significantly worse on this task, whether the NaChBac was expressed

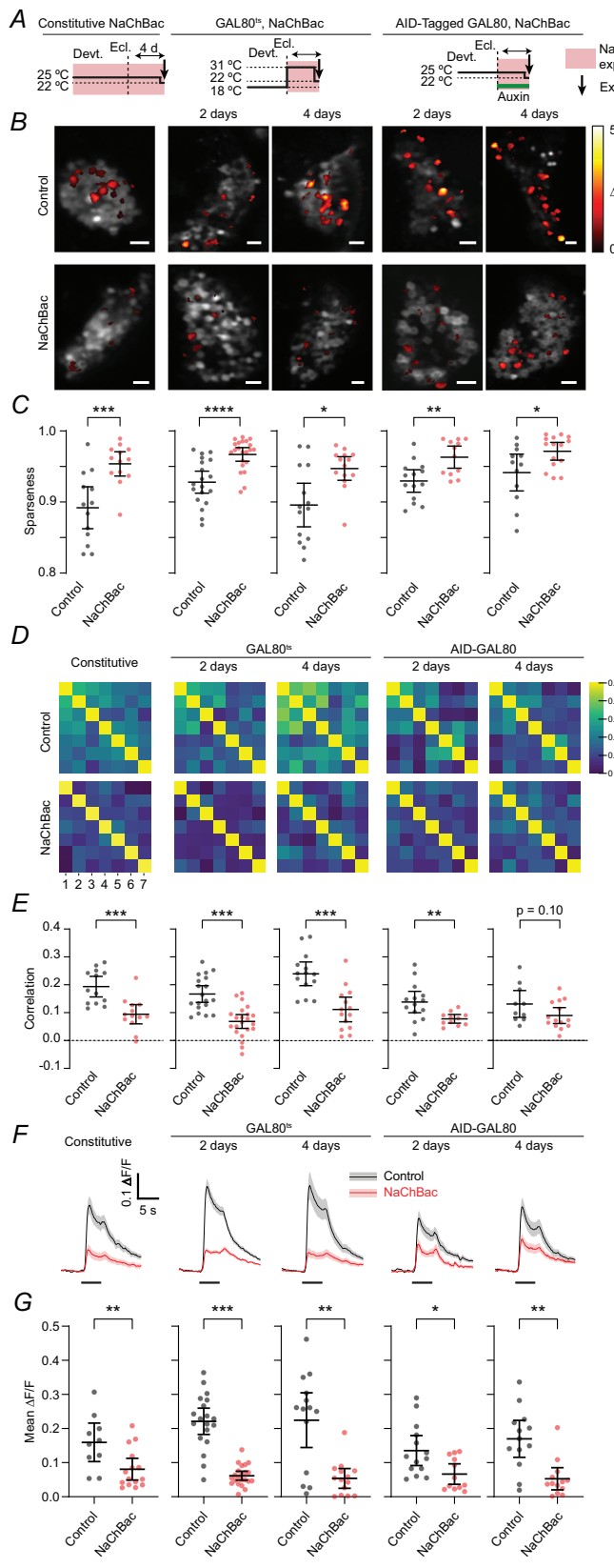

**Figure 4. NaChBac expression increases sparseness of Kenyon cell odour responses**

*A*, schematics show the housing temperatures or auxin exposure applied in each condition. NaChBac and GCaMP6f were expressed under the control of R13F02-GAL4: (left to right) constitutively, for 2 or 4 d (by GAL80[ts]) or for 2 or 4 d (by AGES). The same left-to-right order of conditions is used in every panel. *B*, activity maps of Kenyon cell soma responses to isoamyl acetate, with and without NaChBac expression. Greyscale, baseline fluorescence; false colouring, Δ*F/F*. Scale bar: 10 μm. *C*, mean population sparseness of Kenyon cell soma odour responses for the seven odours listed in panel *D*, with (red) and without (black) NaChBac expression, under the experimental conditions shown in panels *A* and *B*. *D*, matrices show pairwise correlations between Kenyon cell soma odour responses, to: (1) isoamyl acetate, (2) 3-octanol, (3) butyl acetate, (4) ethyl butyrate, (5) apple cider vinegar, (6) δ-decalactone, (7) methylcyclohexanol. *E*, mean inter-odour correlations from (*D*), with (red) and without (black) NaChBac expression. *F*, traces show Kenyon cell soma odour responses averaged across all cells and all odours listed in (*D*). Black bars indicate odour presentation. Traces are means ± SEM. *G*, Responses in (*F*) quantified as the mean Δ*F/F* during the odour. Graphs show means ± 95% confidence interval. *$P < 0.05$, **$P < 0.01$, ***$P < 0.001$ and ****$P < 0.0001$, Mann–Whitney's test. From left to right, $n = 13(9)$, $13(10)$, $19(13)$, $22(12)$, $14(9)$, $14(10)$, $14(8)$, $12(8)$, $10(7)$, $15(9)$. See Table A3 for details of statistical tests.

constitutively or for 2 or 4 days in adults only (Fig. 5). (We used AGES rather than GAL80[ts] for adult-only expression because flies raised at 18°C and heated to 31°C for 2–4 days did not walk enough in our behavioural apparatus to measure learning.) NaChBac-expressing flies were still innately attracted to vinegar before training (Fig. 5C,F), so the defect is specific to associative learning rather than odour perception *per se*. Interestingly, the learning defect was less severe with adult-only expression: whereas flies with constitutive expression showed no significant learning (Fig. 5B), the learning scores in flies with adult-only expression were significantly higher than zero, though lower than the controls (Fig. 5E). It is striking that Kenyon cells that hardly fire normal action potentials (Fig. 3) can still support some learning (see Discussion).

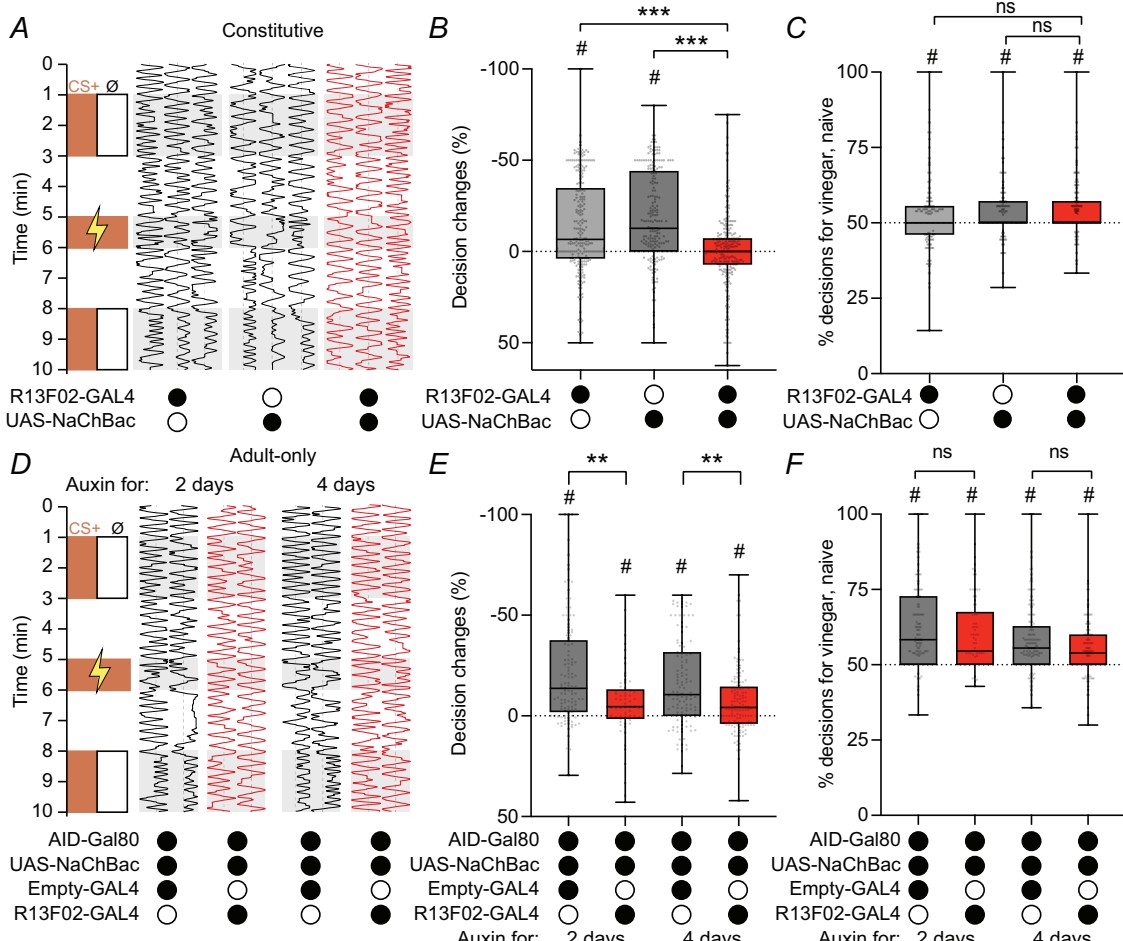

**Figure 5. NaChBac expression prevents olfactory learning**
*A*, example traces of individual fly trajectories before and after training. Flies were presented with a choice between air (Ø) or vinegar (1:100; 'CS+', orange) on each side of the chamber for 2 min, to assess their naïve preference. Two minutes later, vinegar was paired with electric shock for 1 min; 2 min later, air *versus* vinegar was presented again for 2 min. Fly position within the chamber (horizontal dimension) is plotted against time (vertical dimension). The dotted line shows the centre of the chamber. *B*, the change in the percentage of decisions made in favour of the CS+ odour. *n*, given as number of flies [number of experiments]: R13F02-GAL4: 124[9], UAS-NaChBac: 109[7], R13F02-GAL4, UAS-NaChBac: 140[10]. *C*, percentage of decisions in favour of the CS+ (vinegar) before training, with constitutive expression of NaChBac in Kenyon cells driven by R13F02-GAL4. *D*, as in (*A*), but with 2-d or 4-d expression of NaChBac. AID-GAL80: GAL80 tagged with auxin-inducible degron (AID). *E*, as in (*B*), but with 2-d or 4-d expression of NaChBac. *N* = 119[8], 61[5], 120[10], 133[10]. *F*, as in (*C*), but with 2-d or 4-d expression of NaChBac. **$P < 0.01$, ***$P < 0.001$, Kruskal–Wallis ANOVA with Dunn's multiple comparisons test. Innate decision scores do not differ significantly between groups (Kruskal–Wallis test with Dunn's multiple comparison test). # significantly different from 0 (B,E) or 50 (C,F), $P < 0.05$, Wilcoxon's signed-rank test with Holm–Bonferroni correction. Box plots show median and interquartile range and whiskers show the min–max range. See Table A3 for details of statistical tests.

## NaChBac expression decreases endogenous Para expression

We next asked whether compensation for NaChBac expression might contribute to NaChBac's suppression of Kenyon cell activity. *Drosophila* motor neurons compensate for increased (decreased) synaptic excitation by down- (up-) regulating the fly's sole voltage-gated $Na^+$ channel, *paralytic* (*para*) (Mee et al., 2004). Therefore, we asked whether Kenyon cells might react to the abnormal NaChBac $Na^+$ conductance (Figs 1 and 3) by down-regulating Para.

We measured Para expression using the FlpTag allele Para-FlpTag (Fendl et al., 2020), in which GFP is inserted between the first two exons of the *para* mRNA only in cells expressing the recombinase Flp (Fig. 6A). Insertion of GFP in this locus does not substantially affect Para function at 25°C and labels all splice iso-forms of Para (Ravenscroft et al., 2020). We expressed dsRed using the direct fusion mb247-dsRed as an anatomical marker, lexAop-FLP under the control of mb247-LexA, and UAS-NaChBac under the control of R13F02-GAL4. Consistent with previous findings (Ravenscroft et al., 2020), we found no Para expression in the calyx, with Para expressed most strongly in the posterior peduncle (the likely SIZ) and somewhat less intensely in the lobes (Fig. 6B,C). (In contrast, NaChBac-GFP is expressed throughout the calyx, lobes and somata: Fig. 1A). Compared with control flies, flies constitutively expressing NaChBac in Kenyon cells showed decreased Para-GFP fluorescence, especially in the posterior peduncle (Fig. 6C–E). We observed the same effect with or without normalizing GFP to dsRed fluorescence (Fig. 6E,F). The Para-GFP/dsRed ratio in the peduncle was also reduced when NaChBac was acutely expressed for 2 d or 4 d (Fig. 6G,H). These results indicate that NaChBac expression makes Kenyon cells decrease their Para expression.

Reduced Para expression is consistent with the observed reduction in action potential frequency (Figs 1, 3, 7A), but also predicts higher spiking thresholds and smaller spikes. As a proxy for the spiking threshold, we examined the minimum somatic voltage at which an action potential could be detected (that is, the minimum voltage at action potential onset, defined by the peak of the second derivative of the voltage). The minimum voltage at spike onset should be considered as an estimate, not a precise measure, of the spiking threshold, for two reasons. First, different voltages at spike onset were induced by different current steps (at intervals of 10 pA) rather than a ramp protocol; this was because ramping up current injection would trigger a NaChBac potential before normal action potentials. Second, imperfect space clamp means that the voltage at the soma and the SIZ are not necessarily the same. Still, it is reasonable to assume that higher current steps driving higher somatic voltage also drive higher voltage at the SIZ. This estimate of the spiking threshold was higher in NaChBac-expressing cells (Fig. 7B, also visible in the examples in Fig. 1B). (We combined the 2-d and 4-d data because the 2-d and 4-d control data were similar, and only a single 2-d NaChBac-expressing cell spiked.)

We next tested whether NaChBac-expressing cells have shorter or shallower spikes. Both the spike height (difference between voltage at peak *vs.* spike onset) and the upstroke speed (peak d*V*/d*t*) were lower in NaChBac-expressing cells (2-way ANOVA, main effect of NaChBac expression, $P < 0.01$) (Fig. 7C–F). Although only acute expression showed significant differences in *post hoc* tests, there was no significant interaction between NaChBac expression *vs.* acute/constitutive (see Table A3). Because the NaChBac-expressing cells tended to spike only at higher current steps (which drove higher somatic voltage) (Fig. 7B), we sought to control for somatic voltage at spike onset. In control cells, spike height and upstroke speed declined at higher voltages, likely not only because the sodium driving force is naturally lower and more Para channels are inactivated at high voltages, but also because at higher current steps, control cells spiked at high enough rates that the sodium channel refractory period likely also diminishes spike amplitudes. However, cells with acute NaChBac expression showed shorter and shallower spikes than control cells at all voltages (Fig. 7G,H, mixed-effects model, main effect of genotype, $P < 0.01$, see Table A3).

We next examined the repetitive fast inward currents that occur at command voltages higher than around $-30$ mV (Fig. 7I, extracted from Fig. 1C). These fast currents have a similar time course to action potentials: a fast inward current (full width at half-maximum: 1.9 $\pm$ 0.37 ms, mean $\pm$ SD) followed by a slower outward current (Fig. 7J). The currents tended to get more frequent (Fig. 1C) but smaller (Fig. 7K) as the command voltage increased (linear regression of current amplitude *versus* command voltage, slope different from zero for control cells, $P < 0.05$, see Table A3). However, the slope of the IV curve was generally too shallow to reach zero at the reversal potential of sodium; this shallow slope would be expected from inadequate voltage-clamp of the SIZ, which is separated from the soma by a long, thin neurite lacking Para expression (Fig. 6) (Ravenscroft et al., 2020). Thus, these currents likely represent action potentials which are triggered by the current injected to maintain the soma at the command voltage, and which cannot be prevented by the voltage-clamp; indeed, if our voltage-clamp were perfect, we would not get repetitive inward currents, as Para channels would be inactivated after the first current (O'Dowd & Aldrich, 1988).

The fast currents occurred in every control cell (4/4 constitutive; 15/15 2 d acute; 11/11 4 d acute). We

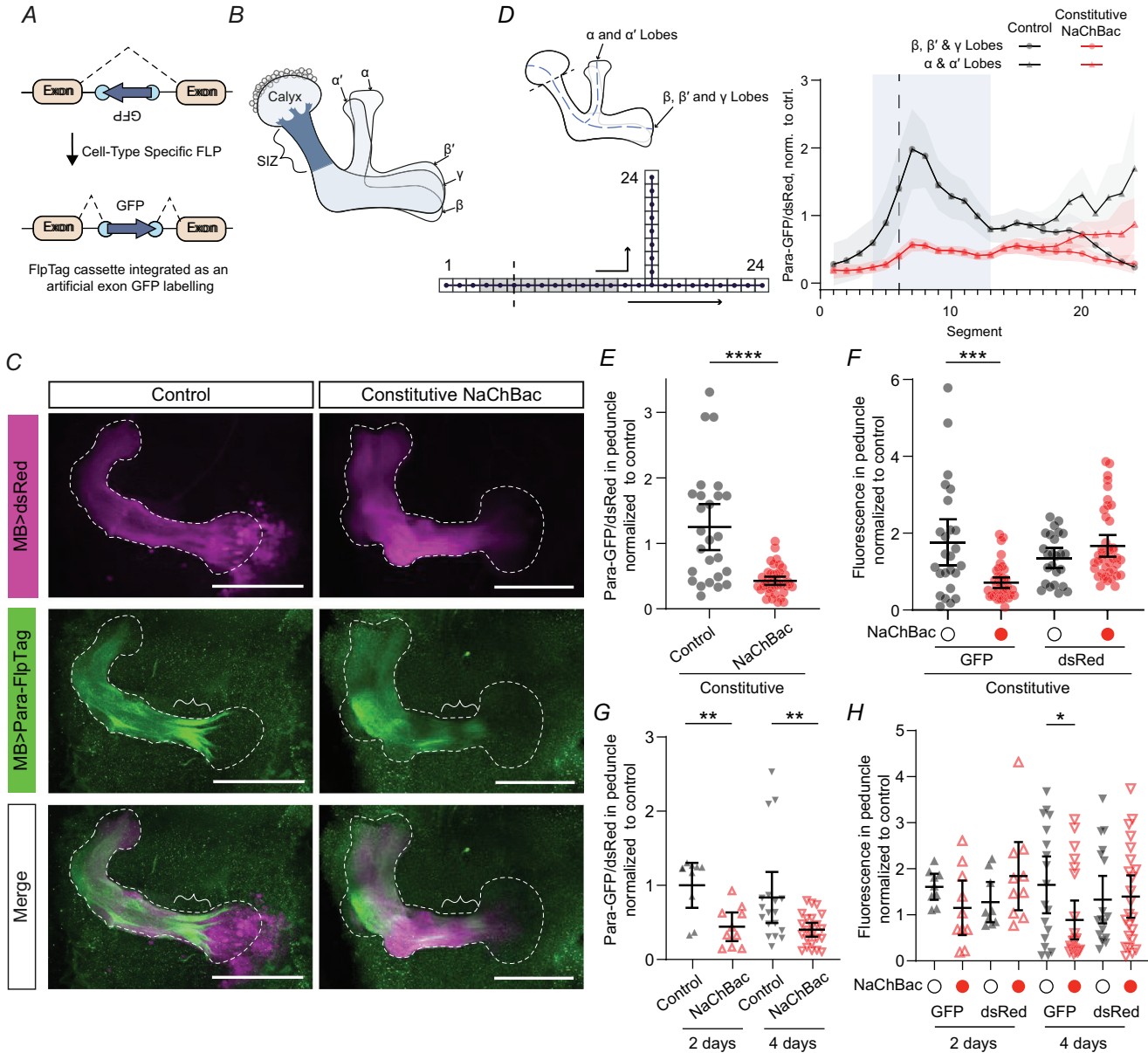

**Figure 6. NaChBac expression reduces Para expression**

*A*, schematic of FlpTag cassette integration into a target gene. The FlpTag cassette is initially inserted in the reverse orientation relative to the target gene. Following cell type-specific Flp expression, the cassette is flipped, so that GFP is spliced into the mRNA between two exons. *B*, diagram of the mushroom body highlighting the spike initiation zone (SIZ), where Para is most highly concentrated. *C*, example maximum intensity Z-projections of control and Kenyon cell > NaChBac mushroom bodies. Constitutive NaChBac expression was driven by R13F02-GAL4; Para-FlpTag (green) was flipped by mb247-LexA driving FLP; the direct fusion mb247-dsRed was included as an anatomical marker (magenta). Dashed white line outlines the mushroom body (excluding the cell bodies). Scale bar: 50 µm. The curly bracket indicates the spike initiation zone (SIZ). *D*, schematic illustrating the mask and skeleton (blue dashed line) used for analysing the mushroom body. The black dashed line marks the boundary between the calyx and the peduncle. Lower panel: diagram showing how the mushroom body structure was divided into evenly spaced segments for quantifying the localization of Para. Right panel: GFP/dsRed ratio in each segment of the mushroom body, normalized to the average ratio across all segments in controls. The shaded area highlights the regions of the calyx and peduncle where Para is concentrated. Dashed line indicates the cut off between the calyx and peduncle. Control $n = 25(17)$, NaChBac $n = 42(28)$. *E*, quantification of the ratio of GFP/dsRed signal averaged across the 10 segments in the shaded area of panel (*D*) (normalized to the average across all segments in the control, mean ± 95% confidence interval). ****$P < 0.0001$, Mann–Whitney test. *F*, mean fluorescence (±95% confidence interval) of GFP and dsRed signal within the shaded area of panel (*D*), normalized to the average across all segments in the control. ***$P < 0.001$, Kruskal–Wallis test with Dunn's multiple comparison

test. *G*, quantification of the ratio of GFP/dsRed signal averaged across the 10 segments of the peduncle for 2 days and 4 days acute NaChBac expression (normalized to control, mean ± 95% confidence interval). **$P < 0.01$, Mann–Whitney test. 2 day expression *n*, control = 9(5), NaChBac = 10(6). 4 day expression *n*, control = 18(10), NaChBac = 23(14). *H*, mean normalized fluorescence (±95% confidence interval) of GFP and dsRed signal within the 10 segments of the peduncle for 2 days and 4 days acute NaChBac expression. *$P < 0.05$, Kruskal–Wallis test with Dunn's multiple comparison test. See Table A3 for details of statistical tests.

detected them more rarely in NaChBac-expressing cells (4/7 constitutive; 1/8 2 d acute, 3/7 4 d acute), consistent with our finding that NaChBac-expressing cells spike much less than control cells (Figs 1 and 3), and they were generally only visible toward the end of the 1 s voltage step after the end of the NaChBac current. Despite the small number of cells showing these fast currents, we observed a significant decrease in their amplitude in cells with adult-only NaChBac expression (Fig. 7L), consistent with decreased Para expression. Note that this quantification does not include the cells where we did not detect fast currents, so it actually understates the effect of NaChBac. Further supporting the interpretation that these fast currents represent action potentials, the fast current amplitude was correlated with the height and upward slope of spikes recorded in current clamp (Fig. 7M,N), and with the total number of spikes recorded in current clamp (Fig. 7O; note the overlapping dots at (0,0): cells without fast currents almost always fired fewer than 10 spikes total across current steps +10 to +100 pA: 7/7 2 d acute, 4/4 4 d acute, 2/3 constitutive). The one NaChBac outlier with unusually large fast currents also had unusually tall, steep spikes and fired an unusually high number of spikes, and showed no NaChBac current (empty red circle, Fig. 7M–O).

Are the spikes and fast currents smaller because of reduced Para expression or because Para channels are inactivated by depolarization block from NaChBac currents? Depolarization block from NaChBac likely contributes to the *absence* of spikes and fast currents, because spikes and fast currents mostly occurred after the end of the NaChBac potential or the NaChBac current, respectively (e.g. Fig. 1C). In cells where the NaChBac current was so large and long-lasting that a 'tail' of residual NaChBac current >30 pA remained in the last 100 ms of the current step, we did not observe fast inward currents (Fig. 7P). However, many cells with a smaller (or non-existent) residual NaChBac current also showed no fast currents, and among those that did show fast currents, there was no correlation between the size of the residual NaChBac current and the size of the fast currents (Fig. 7P). Notably, even cells with little residual NaChBac current had smaller fast currents than controls (Fig. 7P). Finally, spikes in NaChBac-expressing cells were smaller than in control cells regardless of the somatic voltage at spike onset (Fig. 7G,H). These results must be interpreted with caution given that currents and voltages recorded at the soma likely differ from what is experienced by Para

channels in the axon. However, while we do not exclude the possibility that the lower amplitude of spikes and fast currents is caused by continuing depolarization block after the end of the NaChBac potential, we also do not observe any evidence supporting this interpretation.

## NaChBac expression does not alter steady-state voltage-gated outward currents

We next asked whether the decreased excitability of NaChBac-expressing cells might also be explained in part by increased voltage-gated outward currents. Kenyon cells express abundant Shaw and Shab (Amin et al., 2020; Aso et al., 2019), which encode slowly inactivating voltage-gated potassium channels (Smith et al., 2019; Tsunoda & Salkoff, 1995; Wei et al., 1990). Increasing (decreasing) Shaw or Shab conductance generally decreases (increases) neuronal excitability (Hodge et al., 2005; Peng & Wu, 2007), while loss of Shal is compensated in part by increased Shab in fly larvae (Kulik et al., 2019). The presence of a steady-state voltage-gated outward current (likely from Shaw and/or Shab) can be seen in the example voltage-clamp recordings in Fig. 1C: at low command voltages, the steady-state current (after the end of the NaChBac current) increases linearly with the command voltage (representing the leak current), but at higher command voltages, the steady-state current increases faster. This steady-state voltage-gated outward current opens in control cells when the command voltage reaches around −40 mV (Fig. 8A), consistent with reported Shab conductances (Wei et al., 1990). We quantified this current by subtracting the estimated leak current and residual NaChBac current from the steady-state current (see Methods) and found no significant difference between control and NaChBac-expressing cells (Fig. 8B,C). While we cannot directly measure fast-inactivating outward currents, as these currents are obscured by the NaChBac current (Fig. 8A), these results suggest that NaChBac expression at least does not alter slowly inactivating voltage-gated outward currents.

## Compartmental modelling reproduces physiological results and predicts variable effects of NaChBac

Given that NaChBac expression decreases Para expression (Fig. 6), we investigated the relative contributions of

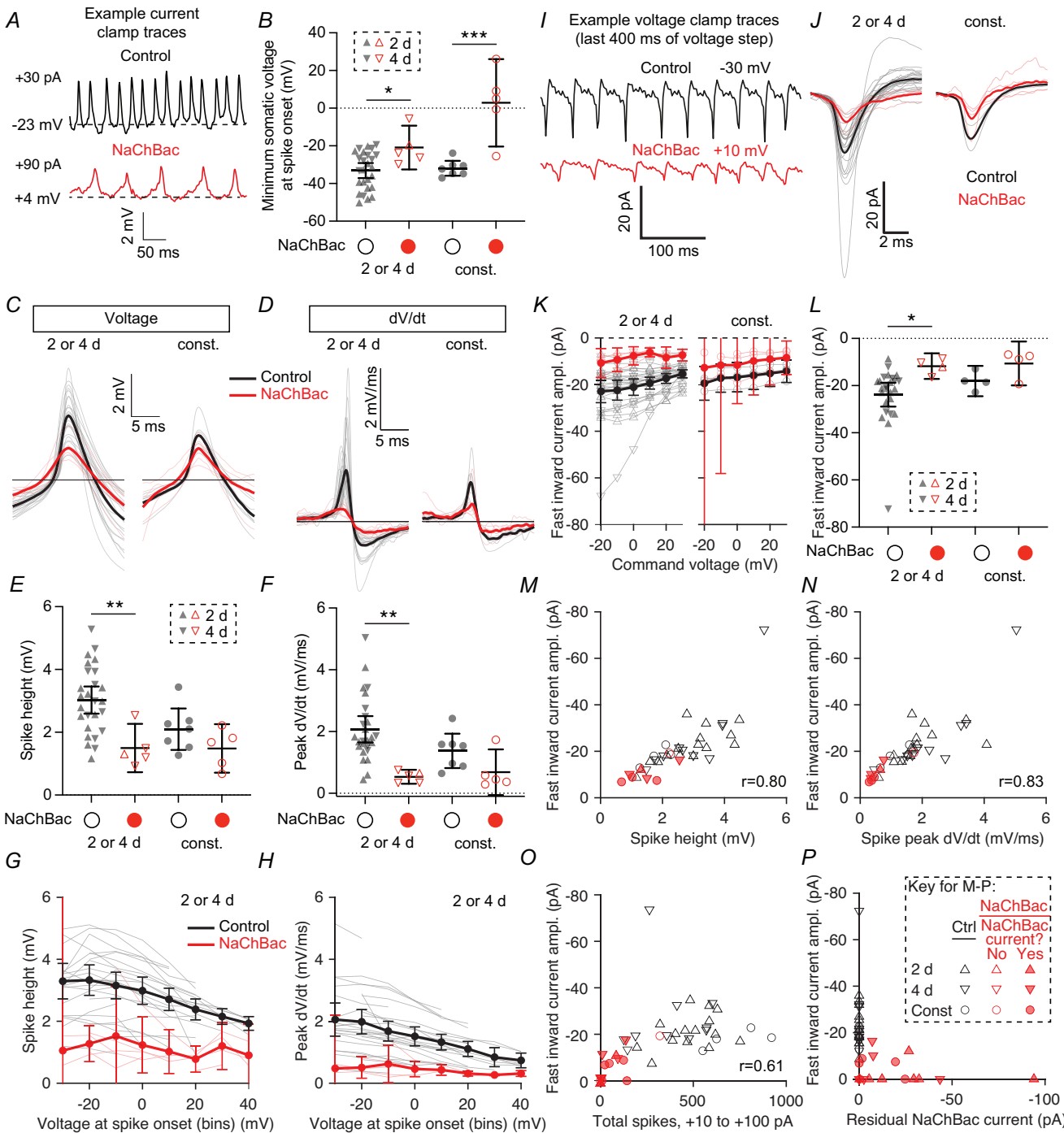

**Figure 7. Action potentials and fast inward currents are smaller in NaChBac-expressing cells**

*A*, action potentials in control *versus* NaChBac-expressing cells (taken from the current-clamp traces in Fig. 1B at +30 and +90 pA, respectively; dashed lines show −23 mV and +4 mV, respectively). *B*, somatic voltage at the time of spike onset (peak second derivative of voltage in the 5 ms before the peak voltage), for the spike with the lowest somatic voltage at spike onset for each cell. The data for 2-d and 4-d NaChBac expression are combined (2 d, upward-pointing triangles; 4 d, downward-pointing triangles). *P = 0.043, ***P < 0.001, 2-way ANOVA with Sidak's multiple comparisons test. *n* (here and panels *E–H*) = 26, 5, 7, 5. 2-d and 4-d data are with AID-GAL80 from Fig. 3; constitutive data are from Fig. 1. *C*, spike waveforms for control (black) and NaChBac-expressing (red) cells. For each cell, the 20 ms around the spike peak were averaged for each current step, and the trace from the current step with the most prominent peak (greatest difference between peak and mean voltage) is shown, to exclude other spikes from the trace window. Horizontal line shows the voltage at time of spike onset. *D*, as with (*C*), but with the derivative of the voltage, showing the slope of the spike waveform. Horizontal line shows d*V*/d*t* = 0. *E*, spike height from (*C*), taken as the difference between the peak voltage and the voltage at the time of spike

onset (horizontal line in (*C*)). \*\**P* = 0.00235, 2-way ANOVA with Sidak's multiple comparisons test. Main effect of genotype, *P* = 0.00555; no significant interaction between genotype and acute *versus* constitutive expression (*P* = 0.149). *F*, Peak d*V*/d*t* from (*D*). \*\**P* = 0.00407, 2-way ANOVA with Sidak's multiple comparisons test. Main effect of genotype, *P* = 0.00508; no significant interaction between genotype and acute *versus* constitutive expression (*P* = 0.213). *G*, as with (*E*), for 2 or 4 days NaChBac expression, but using spike waveforms averaged from spikes whose voltage at spike onset was at different 10 mV bins (e.g. spike height at 0 mV is the height of the waveform averaged across all spikes whose voltage at spike onset was between −5 and +5 mV). *H*, as with (*G*), but for peak d*V*/d*t*. *I*, fast inward currents in control *versus* NaChBac-expressing cells, taken from Fig. 1C at -30 mV and +10 mV, respectively. *J*, waveforms of fast inward currents for control (black) and NaChBac-expressing (red) cells. The waveform from the command voltage with the biggest current for each cell is shown. *K*, amplitudes of fast inward currents from (*J*) at different command voltages. *L*, amplitudes of fast inward currents from (*J*) at the command voltage with the highest current amplitude for each cell. *n* = 25, 4, 4, 4. \**P* = 0.0129, Kruskal–Wallis test with Dunn's multiple comparison test. *M–O*, fast inward current amplitude from (*L*) is strongly correlated with spike height (*M*), peak d*V*/d*t* (*N*), and total number of spikes across current steps +10 to +100 pA (*O*). *P*, fast inward current amplitude is not correlated with the residual NaChBac current in the last 100 ms of the voltage step. In *O* and *P*, cells where we detected no fast inward currents are plotted at 0 on the *y*-axis. *n* = 37 (M,N); 51 (O,P). *M–O*: Filled symbols, NaChBac current; empty symbols, no NaChBac current. Thick lines show averages across cells; thin lines and transparent markers show individual cells. Black: control; Red: NaChBac. Upward-pointing triangles, 2 d auxin; downward-pointing triangles, 4 d auxin; circles, constitutive expression. Graphs show mean ± 95% confidence intervals (where the error bars are very large and extend beyond the axis limits, it is because there are only two cells at that data point). See Table A3 for details of statistical tests.

ectopic NaChBac expression and reduced Para levels to the observed changes in neuronal activity exploring their interactions and how they could lead to decreased spiking in NaChBac-expressing flies. We constructed a detailed compartmental model of a $\gamma$ Kenyon cell from the Flywire connectome (Dorkenwald et al., 2024; Schlegel et al., 2024) (cell ID: 720575940606954507) (Fig. 9A). We fitted the model's passive parameters to experimental hyperpolarizing current injections for individual wildtype Kenyon cells and Kenyon cells constitutively expressing NaChBac. We fitted each individual cell with its own distinct parameter set (rather than fitting one model to averaged data from multiple recordings), because recordings from different cells varied significantly, and averaging across cells would have created an unrealistic model, not representative of any individual neuron (Goaillard & Marder, 2021; Golowasch et al., 2002; Marder, 2023; Marder & Taylor, 2011). We incorporated active firing properties by implementing transient and persistent sodium channels and a delayed rectifier potassium channel, based on previously established ion channel model parameters in *Drosophila* (Günay et al., 2015). We modelled the location of the SIZ as the region just distal to the boundary between the calyx and peduncle of the mushroom body, following our experimental data (Fig. 6) and previous findings (Ravenscroft et al., 2020; Trunova et al., 2011). The resulting wildtype models reproduced the current-clamp recordings, showing action potentials and firing frequencies qualitatively similar to those observed in actual Kenyon cells (Fig. 9B,C,L; compare to Figs 1 and 3).

To model the impact of NaChBac expression in Kenyon cells, we developed an ion channel model using the Hodgkin–Huxley formalism. We based the activation and inactivation curves as well as activation and inactivation

time constants on whole-cell patch-clamp recordings from transfected HEK293 cells expressing NaChBac (Strege et al., 2023). We asked whether introducing NaChBac into the model would reproduce the NaChBac potentials and reduction in normal spikes seen in our data (Figs 1B–E and 9D). We found that we could best recreate the experimental current-clamp data with low NaChBac conductance throughout the model neuron, higher NaChBac conductance in the soma and SIZ, and a decrease in the transient sodium conductance relative to wildtype models (Fig. 9E). Including NaChBac throughout the cell was consistent with the localization of NaChBac-GFP throughout Kenyon cell somata, dendrites and axons (Fig. 1A). The high somatic NaChBac conductance was needed to reproduce the NaChBac potentials, that is, the large, prolonged but transient depolarizations at the start of the current pulse (Fig. 9F). Introducing NaChBac into the model without reducing the transient sodium conductance allowed neurons to sustain high-frequency spiking and sometimes resulted in spikes after the end of the current injection (Fig. 9G), which we never observed in our data.

Therefore, to best fit the experimental electrophysiological data, we adjusted both the compartment-specific NaChBac conductance and transient sodium conductance across each individual model. The best-fitting transient sodium conductances in the SIZ in our NaChBac-expressing models were significantly lower than those in our wildtype control models (Fig. 9K). This reduction improved the match between the model and experimental recordings, including capturing the decrease in the number of action potentials observed in Kenyon cells expressing NaChBac during current injections (Fig. 9L, compare with Fig. 1D). Without this adjustment, the models did not fully reproduce the

physiological response, predicting that changes in transient sodium conductance might contribute to the altered excitability of these neurons.

However, action potentials were largely suppressed at the peak of the NaChBac-mediated depolarization, even without reducing the transient sodium conductance (Fig. 9G). To investigate the underlying mechanism, we examined the gating kinetics of each ionic conductance in the model. Simulations were performed using a 20 pA current injection, corresponding to rheobase in the representative wildtype model cell (Fig. 9H–J, same simulations as in Fig. 9C,E,G). In models with NaChBac conductance levels sufficient to reproduce the observed NaChBac potentials, the NaChBac-mediated depolarization was strong enough to substantially inactivate the transient sodium current (Fig. 9H–J, orange line close to 0). Although small oscillations in the transient sodium channel activation gate persisted, these resulted in only small membrane potential fluctuations

($\sim$0.1$-$0.5 mV), remaining below the spike detection threshold. This transient sodium channel inactivation effectively suppressed action potentials even when the transient sodium conductance was not reduced (Fig. 9G,J). In contrast, in the wildtype model, transient sodium channels remained available to support spiking for all tested suprathreshold current steps. These results predict that depolarization block of transient sodium channels contributes to the experimentally observed suppression of action potentials during NaChBac potentials.

While in Kenyon cells ectopic NaChBac expression decreases Para expression (Fig. 6) and overall excitability (Figs 1 and 3), other studies have reported or assumed that NaChBac increases excitability (Park et al., 2016; Sim et al., 2013). To systematically investigate the effect of NaChBac expression on excitability, we introduced varying levels of NaChBac conductance into the wildtype model from Fig. 9C. Because the subcellular distribution

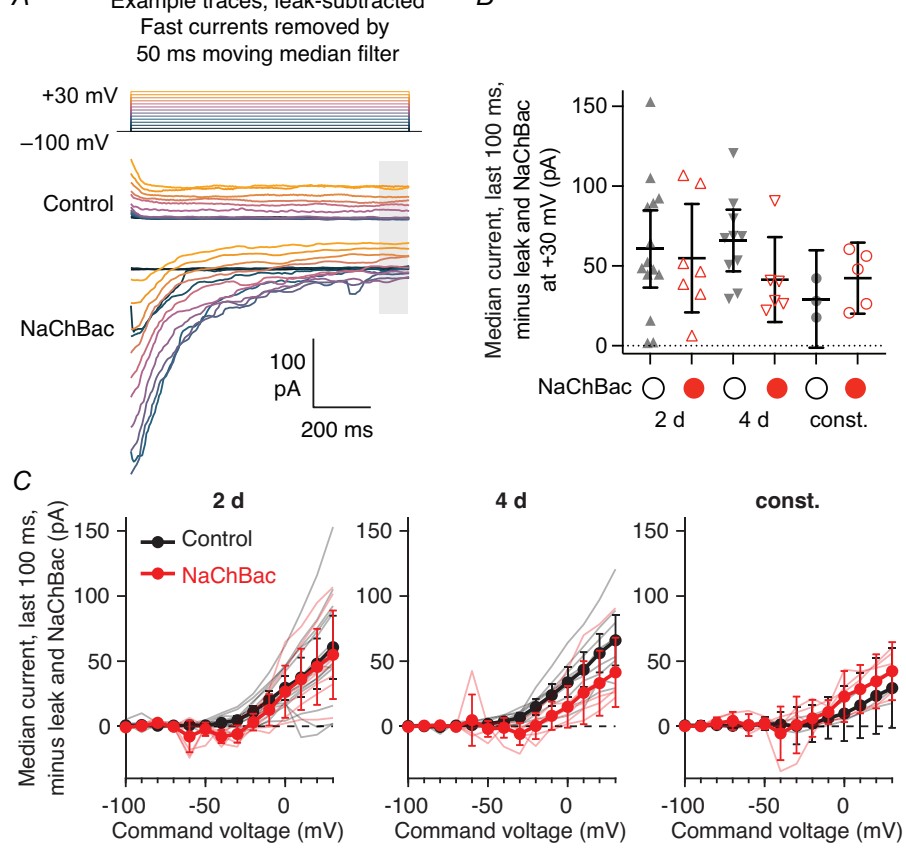

**Figure 8. NaChBac expression does not alter voltage-gated steady-state outward current amplitudes**
*A*, example voltage-clamp traces from control and NaChBac-expressing cells with AID-GAL80 and 2 d auxin exposure. The baseline and leak currents were subtracted and the traces were smoothed by a 50 ms moving median filter to remove the fast inward currents shown in Fig. 7. Both the control and NaChBac cell show a steady-state outward current at command voltages above −30 mV. *B*, median current in the last 100 ms of the voltage step (shaded area in panel *A*), with estimated leak and residual NaChBac currents subtracted, at +30 mV. Mean ± 95% confidence interval; *P* = 0.212, Kruskal–Wallis test. *C*, as in B, but at all voltage steps. Circles/thick lines show the mean ± 95% confidence interval; thin lines show individual cells. See Table A3 for details of statistical tests.

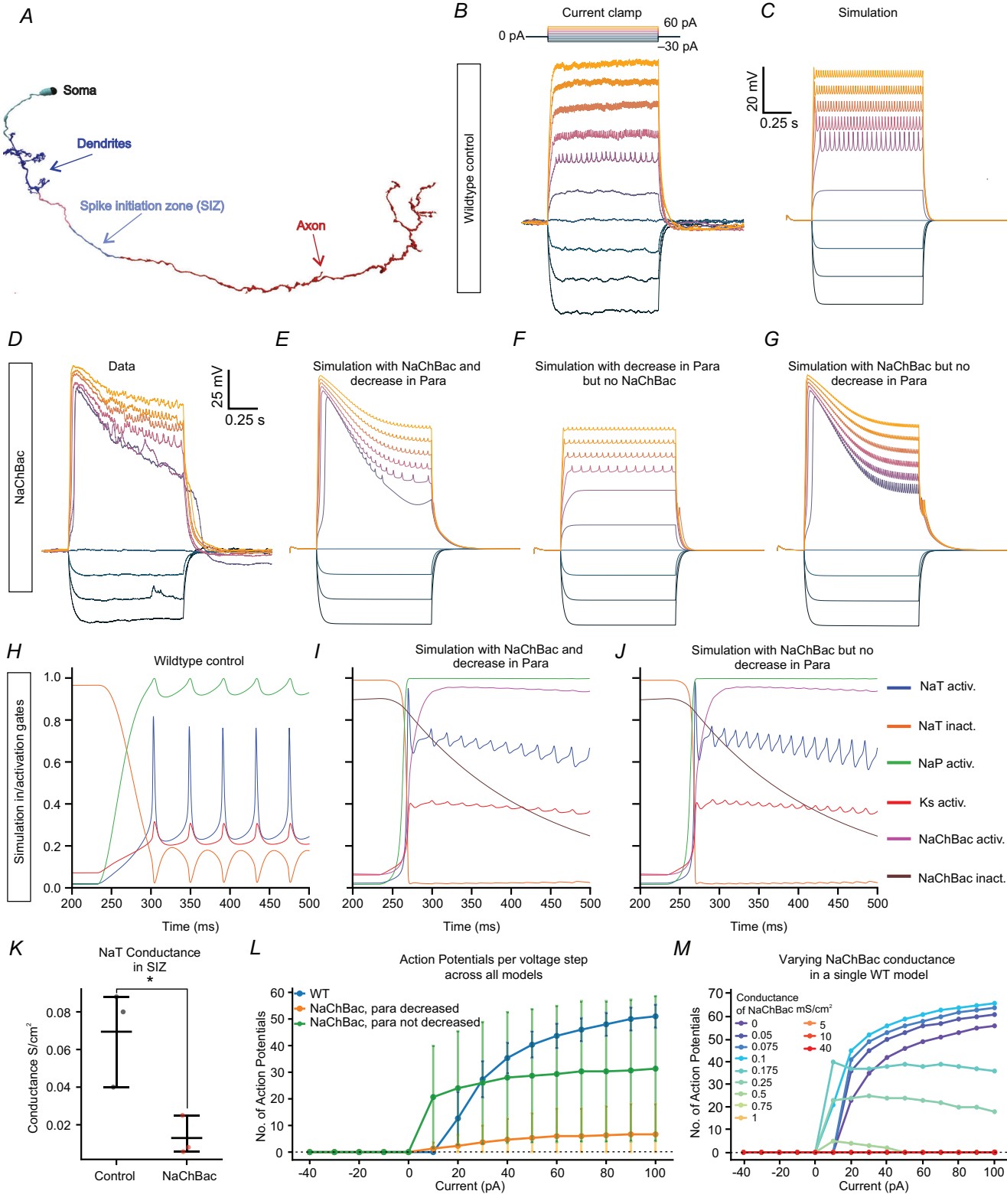

**Figure 9. Compartmental modelling reveals the dual effects of NaChBac on Kenyon cell excitability**

*A*, schematic of the compartmental model of a *γ* Kenyon cell reconstructed from the Flywire connectome. The model includes compartmentalized sections, soma, dendrites, spike initiation zone (SIZ) and axon. *B*, current-clamp recording of wildtype (WT) Kenyon cell to varying current injections. *C*, simulated current-clamp of best-fit model for the cell in (*B*). *D*, current-clamp recording of NaChBac-expressing Kenyon cell. *E*, simulation using optimal parameter set recreating NaChBac Kenyon cells response. *F*, simulation using optimal parameter set from (*E*)

but with no NaChBac. *G*, simulation using average transient sodium conductance of (WT) models and optimal NaChBac conductances from (*E*). *H–J*, NaChBac expression induces a depolarization block. Simulated activation and inactivation gates during 20 pA current injection for the transient sodium channel (NaT), persistent sodium channel (NaP), potassium channel (Ks) and bacterial sodium channel (NaChBac) are shown for (*H*) the wildtype control model (same parameters as in *C*), (*I*) the NaChBac model with reduced Para conductance (same as in (*E*)) and (*J*) the NaChBac model without a reduction in Para conductance (same as in (*G*)). *K*, comparison of transient sodium (Para) conductance in best-fit WT and NaChBac-expressing models. Lines represent means. *$P < 0.05$, unpaired *t*-test. *L*, simulated number of action potentials in response to varying somatic current injections comparing WT and NaChBac-expressing cells with and without reduced Para expression. Error bars indicate standard deviation. *M*, simulated number of action potentials in response to varying current injections with varying conductance of NaChBac within a single WT model (same model parameters as in (*B*)). Initial resting membrane potential for WT cell and simulation (*B–C*) is −60 mV and −70 mV for NaChBac-expressing cell (*D*) and −69 mV for the corresponding models (*D–G*).

of NaChBac expression for other cell types is usually unknown, we modelled a uniform NaChBac conductance throughout the entire morphology. These simulations revealed that firing frequency actually slightly increased at low NaChBac conductance levels when compared with wildtype, while firing frequencies decreased once conductance levels exceeded ∼0.175 mS/cm$^2$ (Fig. 9M). This bimodal effect may reconcile why we see a decrease in excitability, while other studies reported an increase. These modelling results also suggest that high NaChBac expression levels alone can reduce firing frequencies, independently of Para downregulation, as NaChBac-mediated depolarization induces voltage-dependent inactivation of the transient sodium channel. In contrast, at lower NaChBac conductance levels, the NaChBac-mediated depolarization is less pronounced, resulting in minimal inactivation of transient sodium channels and allowing action potentials to persist or even increase in frequency (Figs 9M and A8B–F). However, in our case, introducing only a NaChBac conductance in the models (without a compensatory reduction of the transient sodium conductance) did not replicate the physiological responses observed in NaChBac-expressing Kenyon cells (Figs 9G and A8). Collectively, these results underscore the complex interactions among ionic conductances in determining neuronal excitability and provide a more refined mechanistic understanding of NaChBac's role in modulating neuronal firing.

## Discussion

We have shown that ectopic expression of NaChBac in Kenyon cells reduces their expression of endogenous sodium channels and reduces their intrinsic excitability, thereby reducing odour-evoked calcium influx and olfactory associative learning. Odour-evoked calcium influx (but not excitability or learning) recovers after 4-day adult-only NaChBac expression (but not with constitutive expression), suggesting a developmentally limited compensatory increase in voltage-gated calcium influx.

### Decreased excitability explained by loss of Para

We showed experimentally that NaChBac expression decreases Para expression (Fig. 6) and fast inward currents likely carried by Para (Fig. 7). We found in a compartmental model that some of the loss of spiking, particularly during a NaChBac potential, can be explained by a depolarization block of Para channels. However, the electrophysiological phenotype of NaChBac-expressing Kenyon cells is best fit in the model when we also decrease the endogenous voltage-gated sodium conductance (Fig. 9). These results are consistent with previous findings in dissociated neurons where pharmacologically blocking Na$^+$ channels with tetrodotoxin causes a compensatory increase in sodium conductance revealed when the drug is removed (Desai et al., 1999). Our findings show that sodium channels also decrease for an ectopic sodium conductance. As we measured Para protein levels, it is unclear whether this is due to decreased transcription or translation (Mee et al., 2004), or increased degradation. This Para downregulation suggests that Kenyon cells may homeostatically control not the firing rate of action potentials, but other physiological variables like total depolarization or sodium influx. Under this interpretation, because Kenyon cells lack an endogenous mechanism to remove a foreign prokaryotic sodium channel or downregulate expression of GAL4-driven NaChBac, they reduce sodium influx by the only mechanism available to them: downregulating Para. Such a compensatory mechanism makes sense in a natural setting, where spike rate and sodium influx are correlated; only by introducing unnatural NaChBac potentials can we dissociate spike rate and sodium influx to reveal that Kenyon cells respond to excess sodium influx by reducing Para expression even to the point of nearly eliminating spikes.

We do not exclude the alternative interpretation that decreased Para reflects a passive side effect of NaChBac expression, rather than homeostatic compensation driven by the cell sensing excess depolarization or sodium influx. A generic side effect (e.g. NaChBac outcompetes other membrane proteins for trafficking or chaperone machinery) seems unlikely, since steady-state

voltage-gated outward currents were unaffected by NaChBac expression (Fig. 8), and these currents are presumably carried by Shaw/Shab potassium channels that likely rely on the same trafficking or chaperone machinery as Para and are expressed at similar levels (Amin et al., 2020; Aso et al., 2019). However, it could be that Para expression is especially sensitive to, for example, cellular stress triggered by excess depolarization.

Similar to our finding that expression of a sodium channel decreases excitability, counterintuitive effects due to compensation have also been observed in neurological diseases (Shabani et al., 2024). In Dravet syndrome, caused by loss of function of the sodium channel Scn1a/Nav1.1, epileptic seizures were classically attributed to decreased excitability of inhibitory neurons, but Nav1.1 loss of function can also increase excitability of excitatory neurons (Liu et al., 2013), in part through a compensatory increase in Scn8a/Nav1.6 and decrease of SK2 calcium-activated potassium channels (Ritter-Makinson et al., 2019). Meanwhile, whereas myelinated axons express Nav1.6 at nodes of Ranvier, loss of myelin often leads to compensatory expression of Nav1.2 in demyelinated sections; while this can restore action potential propagation, it can also lead to ectopic firing (Smith, 2007). Indeed, theoretical studies have shown that compensation in ion conductances can often cause pathological outcomes (O'Leary, 2018). Together, results underscore the conservation of homeostatic regulation of ion channels across species from arthropods to humans and suggest that the *Drosophila* mushroom body may be a fruitful model for studying compensation for channelopathies *in vivo*.

### Adult-specific plasticity in calcium influx

Although NaChBac reduced Kenyon cell excitability whether it was expressed constitutively or only in adults, it affected odour-evoked calcium influx differently depending on timing: odour responses were decreased when NaChBac was expressed constitutively or for 2 d in adults, but not when it was expressed for four or more days in adults. The difference between 2 and 4 d suggests homeostatic compensation in voltage-gated calcium influx between 2–4 d expression. The apparent lack of compensation with constitutive expression likely reflects developmental 'lock-in': activity-dependent plasticity is often subject to critical periods, where plasticity only occurs in certain stages of development (Desai et al., 2002; Lipska et al., 2002; Lowe et al., 2024; Wen & Turrigiano, 2021) and indeed the homeostatic set-point itself may be specified by average activity during a critical period (Giachello & Baines, 2015, 2017). Thus, NaChBac expression during development could fix the set-point for Kenyon cell calcium influx at an inappropriately low level, such that homeostatic compensation would not be engaged in the adult. In contrast, with adult-only expression, the set-point is correct and homeostatic compensation can restore odour-evoked calcium influx given enough time. Defects specific to developmental expression of NaChBac (e.g. in synaptic connectivity) may also explain why the learning defect appears more severe with constitutive NaChBac expression than adult-only expression (Fig. 5). As with Para expression, we do not exclude the alternative interpretation that the increase in odour-evoked axonal/dendritic calcium influx between 2 and 4 d reflects a passive side effect of NaChBac expression, rather than active compensation targeting a set-point of calcium influx.

The difference between 2 and 4 d likely reflects a change in axonal/dendritic voltage-gated calcium influx because there was no difference in odour responses in Kenyon cell somata. If the change occurred through increased intrinsic excitability, we would have observed increased spiking from current injections in Fig. 3, but we did not. If the change occurred through increased excitatory synaptic input, we would have observed increased activity also in Kenyon cell somata, but we did not (Fig. 4). The contrast between a change in calcium response in dendrites/axons but no change in cell bodies (Figs 2 and 4) is consistent with previous findings that sparseness/correlation of soma odour responses can be dissociated from odour-evoked axonal/dendritic calcium influx (Bielopolski et al., 2019).

A compensatory increase in voltage-gated calcium influx in Kenyon cell axons and dendrites would be consistent with previous findings of homeostatic regulation of presynaptic voltage-gated calcium influx (Frank et al., 2020). (Note that Kenyon cell dendrites also contain presynaptic sites (Christiansen et al., 2011)). One candidate mechanism is increased expression of Cacophony VGCCs. Cacophony's localization to the synaptic neuropil as opposed to cell bodies (Gratz et al., 2019) could explain the lack of compensation in somatic odour responses. Compensation could also occur not through expression but rather through post-translational modification/transport of VGCCs (Cunningham & Littleton, 2023). It will be interesting in future studies to test these possible mechanisms.

Why do flies with 4-d NaChBac expression in Kenyon cells not learn normally, even though their odour-evoked calcium influx is normal? Presynaptic calcium may not be sufficient for olfactory learning: synaptic plasticity at Kenyon cell output synapses requires not just cAMP (produced by the calcium-dependent adenylyl cyclase Rutabaga), but also depolarization (Yamada et al., 2024), and it is unclear whether the non-cAMP effects of depolarization occur via calcium influx. Since the NaChBac-expressing Kenyon cells fire NaChBac potentials and few normal spikes, presynaptic depolarization is likely abnormal in unpredictable ways.

Moreover, information normally carried by Kenyon cell spike timing (Gupta & Stopfer, 2014) would be lost. Perhaps what is more remarkable is that Kenyon cells with adult-only NaChBac expression can support learning at all (Fig. 5) despite firing few normal spikes, suggesting that even abnormal NaChBac potentials can carry useful information.

### Divergent possible effects of NaChBac expression

Some studies find behavioural effects of neuronal NaChBac expression in *Drosophila* that suggest that NaChBac activates neurons (Crocker et al., 2010; Deng et al., 2019; Donlea et al., 2011; Hergarden et al., 2012; Nitabach et al., 2006; Root et al., 2011; Zimmerman et al., 2017) while others suggest NaChBac silences neurons (Alphen et al., 2021; Flood et al., 2014; Luan et al., 2006). Physiological studies in mammalian neurons have typically found that NaChBac increases intrinsic excitability (Kelsch et al., 2009; Kim et al., 2020; Sim et al., 2013; Xue et al., 2014). Our study suggests two possible reasons for these divergent results. First, some cell types may downregulate Para like Kenyon cells (Fig. 6), while others might not. Second, our compartmental model predicts that NaChBac can both enhance excitability at low expression levels and suppress it at high levels (Fig. 9J). Different methods for NaChBac expression (different viral vectors or GAL4 drivers) likely drive different expression levels, potentially leading to divergent physiological (and therefore behavioural) effects. Different neuronal morphologies would likely also contribute to different effects of NaChBac expression. These findings highlight the importance of verifying the physiological effects of genetic tools used to manipulate neural signalling when studying neural circuits and behaviour.

### Appendix A

To investigate the sources of heterogeneity in measured NaChBac currents, we looked for correlations between NaChBac current amplitude and other electrophysiological parameters. The most striking correlation was that cells with low input resistance tended to have small/no NaChBac currents ($r = -0.61$) (Fig. A1A,B). Indeed, most cells with $R_{in} < 0.8$ GΩ had no NaChBac current. It may be that recordings with a low input resistance simply had an unhealthy cell or an imperfect seal between the pipette and the membrane. On this basis, we excluded cells with $R_{in} < 0.8$ GΩ from our main analyses. Excluded cells are shown in Figs A1 and A3 marked by 'x's. Importantly, NaChBac-expressing cells spike less than control cells across the whole range of input resistances measured, even in the excluded cells (Fig. A1C).

We also asked whether NaChBac current amplitude was related to series resistance; although the amplifier software compensated for series resistance, the capacitive transients at the start of the voltage step indicated that a few cells had abnormally high uncompensated series resistance, possibly indicating incomplete break-in. Most cells with high series resistance were excluded on the basis of low input resistance, but there were three non-excluded cells in our dataset with high series resistance (80, 83, 320 MΩ) (Fig. A1D). However, unlike input resistance, series resistance (estimated from the capacitive transient; see Methods) was not significantly correlated with NaChBac current amplitude: several cells with low series resistance had no NaChBac current, while the three non-excluded cells with abnormally high series resistance still showed clear NaChBac currents (Fig. A1D).

To test if series resistance would be predicted to affect the measured voltage-sensitivity of NaChBac currents, we first noted that the voltage-clamp error introduced by series resistance is estimated as $V_{error} = I^*R_s$. Cells with high series resistance tended to have smaller absolute currents (Fig. A1D), a tendency that might naturally limit the voltage-clamp error. Indeed, there was no obvious effect of series resistance on the command voltage at which the peak measured NaChBac current occurred (Fig. A1E). The estimated $R_s$ error from the NaChBac current ($V_{error} = I_{NaChBac}^*R_s$) was $<5$ mV for most cells, and even for cells with estimated $R_s$ error in the order of 10 mV, their voltage of peak NaChBac current did not differ from the other cells (Fig. A1F). The predicted $R_s$ error in a voltage step can be estimated as $V_{error} = I_{leak}^*R_s = (\Delta V/R_{in})^*R_s$, that is, the error is a fraction ($R_s/R_{in}$) of the intended voltage step. The ratio $R_s/R_{in}$ was relatively low even for the cells with high series resistance (0.04, 0.08, 0.21). For the cell with the highest $R_s/R_{in}$, the $R_s$ error would be 21% (Fig. A1G), so in stepping down from $-60$ to $-100$ mV, the cell would have reached $-92$ instead of $-100$ mV, which should still be sufficient to open the NaChBac channel. Thus, voltage errors from series resistance were minimal and unlikely to qualitatively change our voltage-clamp results.

Series resistance might affect our current-clamp data by (1) leading to overestimated input resistance or (2) filtering fast events like action potentials. On (1), estimated series resistance was not positively correlated with measured input resistance (as would be expected if measured input resistance were dominated by series resistance), and the measured input resistance in cells lacking a matching voltage-clamp recording (thus unknown series resistance) was no higher than in cells with a matching voltage-clamp recording (Fig. A1G). On (2), clear spikes could be detected even in cells with high series resistance, so it is unlikely that filtering caused by high series resistance explains the lack of spikes in NaChBac-expressing cells. Indeed, we observed fewer spikes in NaChBac-expressing cells than in control

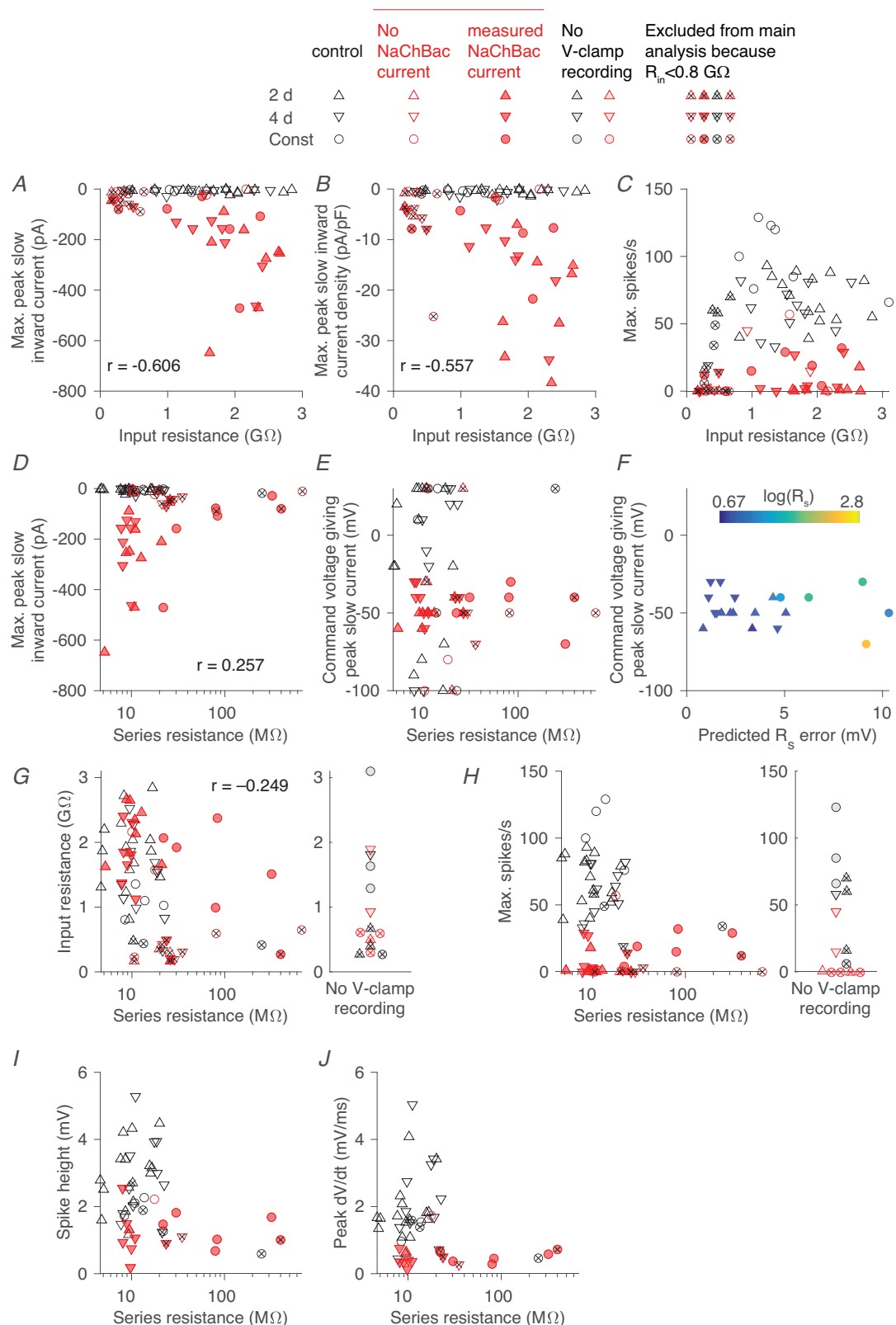

**Figure A1. Correlations between electrophysiological parameters**

*A*, maximum peak slow inward current plotted against input resistance, including cells excluded from the main figures due to low input resistance (marked by '*x*'s). Black, control; red, NaChBac. Circles, constitutive expression; upward-pointing triangles, 2-d expression, downward-pointing triangles, 4-d expression. Empty symbols: No NaChBac current; filled symbols, NaChBac current; lightly filled symbols, no voltage-clamp recording. *B*, maximum peak slow inward current density (from Fig. 1I, F) plotted against input resistance. *C*, maximum spike rate plotted against input resistance. Note that control cells spike more than NaChBac cells at every input resistance level. *D*, maximum peak slow inward current plotted against uncompensated series resistance estimated from the capacitive transient. Cells with high series resistance tend to have small NaChBac currents. *E*, command voltage giving the largest slow inward current, plotted against uncompensated series resistance. Cells without NaChBac currents are distributed randomly along the *y*-axis, but cells with NaChBac currents consistently have their peak slow inward current at command voltages around −70 to −30 mV, regardless of the uncompensated series resistance. *F*, command voltage giving the largest slow inward current, plotted against the predicted voltage error from series resistance, calculated according to $V_{error} = I_{NaChBac}*R_s$. *G*, input resistance plotted against uncompensated series resistance to test whether high series resistance leads to overestimating the input resistance. *H*, maximum spike rate plotted against series resistance. NaChBac recordings have few spikes regardless of the series resistance. *I–J*, spike height (*I*) and peak d*V*/d*t* (*J*) plotted against series resistance. Spikes in NaChBac cells have reduced amplitude and upstroke slope even when considering only cells with low series resistance. *r* values are Pearson's correlation coefficients for the NaChBac cells only.

cells whether considering cells with low, high, or unknown series resistance (Fig. A1H). Moreover, the few action potentials that NaChBac-expressing cells fired had reduced amplitude and upstroke slope even when considering only cells with low series resistance (Fig. A1I,J). In summary, because series resistance did not appear to affect our main conclusions, we did not exclude cells on the basis of high series resistance.

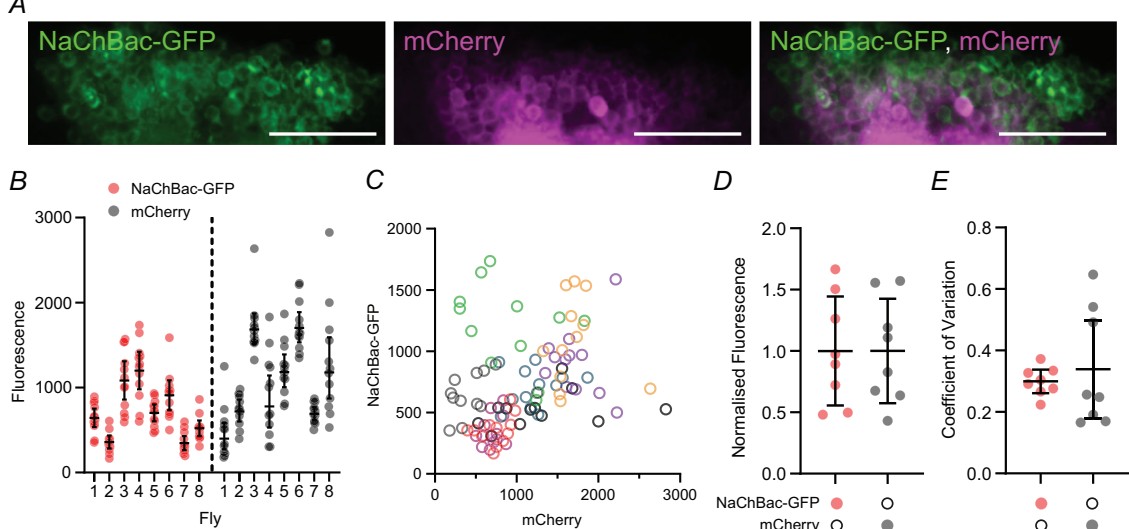

**Figure A2. Heterogeneity in NaChBac-GFP fluorescence across Kenyon cells**

*A*, example maximum intensity Z-projections of R13F02-GAL4 driving NaChBac-GFP (green) and mCherry (magenta) expression. Scale bar: 50 μm. *B*, fluorescence intensities of NaChBac-GFP and mCherry were quantified in a random selection of Kenyon cells (*n* = 12 cells from eight brains per group). Graph shows mean ± 95% confidence interval. *C*, NaChBac-GFP fluorescence intensity plotted against mCherry in the same cells to assess relative expression levels. Each point represents a single Kenyon cell, NaChBac-GFP (*n* = 12[8]) and mCherry (*n* = 12[8]); each colour represents a different brain. *D*, normalized average fluorescence per fly from panel (*B*) for NaChBac-GFP and mCherry. Graph shows mean ± standard deviation. *E*, quantification of the coefficient of variation within flies from panel (*B*) for NaChBac-GFP and mCherry fluorescence intensities. Graph shows mean ± standard deviation.

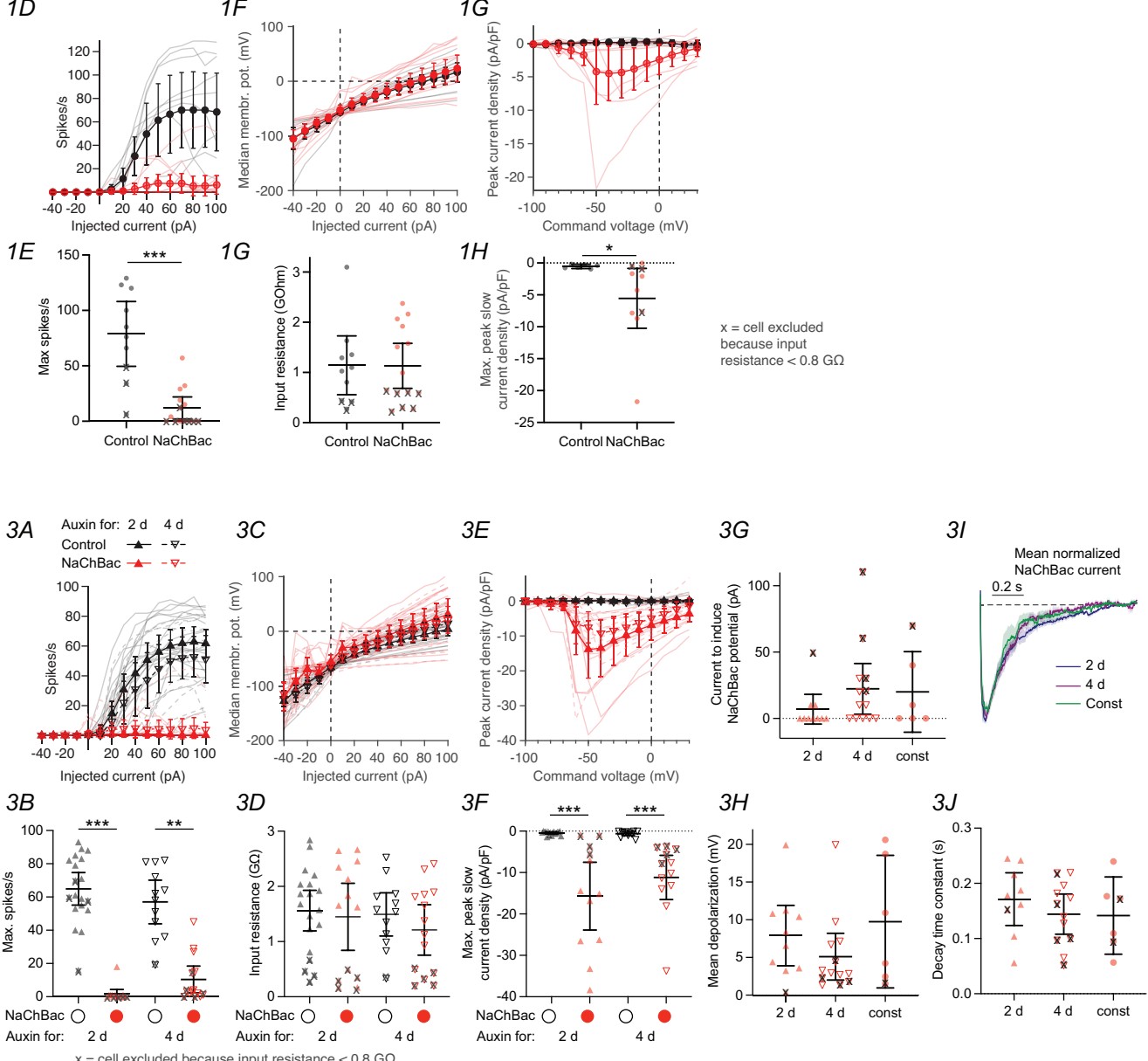

**Figure A3. Excluded data points from Figs 1 and 3**
Cells excluded from Figs 1 and 3 due to input resistance less than 0.8 GΩ are shown here marked by 'x's, to illustrate that our results are unaffected by including the excluded cells. Panel letters follow those of Figs 1 and 3; see main figure legends. See Table A3 for details of statistical tests and Dataset S1 for raw data.

*A*

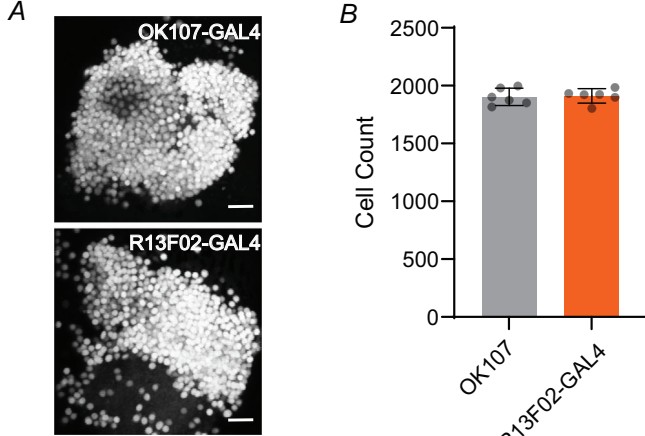

*B*

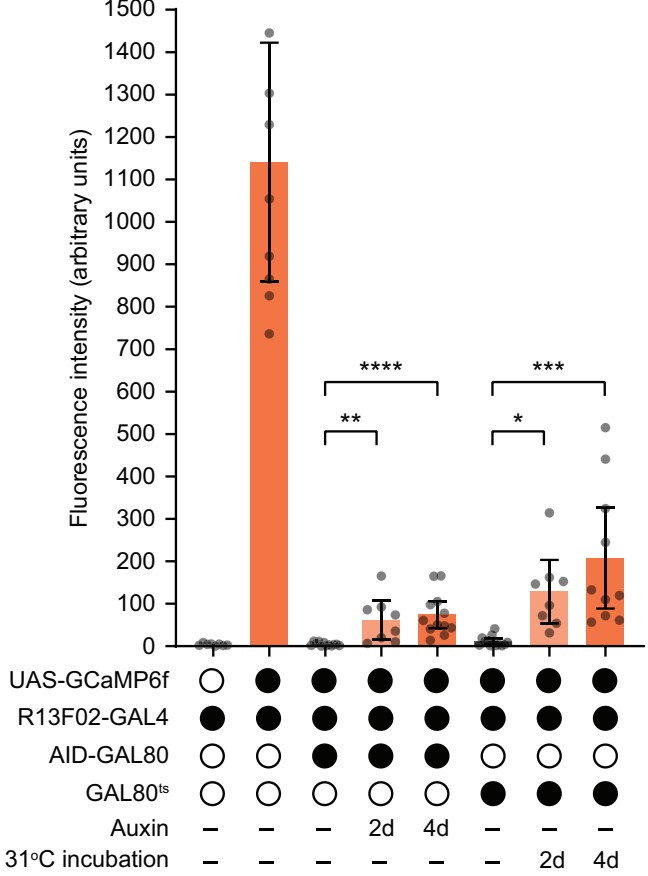

**Figure A4. R13F02-GAL4 labels approximately all Kenyon cells**
*A*, example images showing maximum intensity Z-projections of
OK107 > Stinger-GFP and R13F02-GAL4 > Stinger-GFP. Scale bar =
15 μm. *B*, Kenyon cell count for (*A*), comparing OK107 (*N* = 6)
*versus* R13F02-GAL4 (*N* = 6) (means ± 95% confidence intervals).

**Figure A5. GAL4 efficacy with temporally limited suppression
by GAL80$^{ts}$ or AID-GAL80**
Fluorescence levels in the mushroom body were compared in
R13F02-GAL4 > UAS-GCaMP6f flies under the different conditions
shown by the open and closed circles. *N* = 8, 9, 12, 8, 12, 13, 8, 10.
Graph shows means ± 95% confidence intervals. *$P < 0.05$,
**$P < 0.01$ ***$P < 0.001$, ****$P < 0.0001$, Kruskal–Wallis, Dunn's
multiple comparisons test.

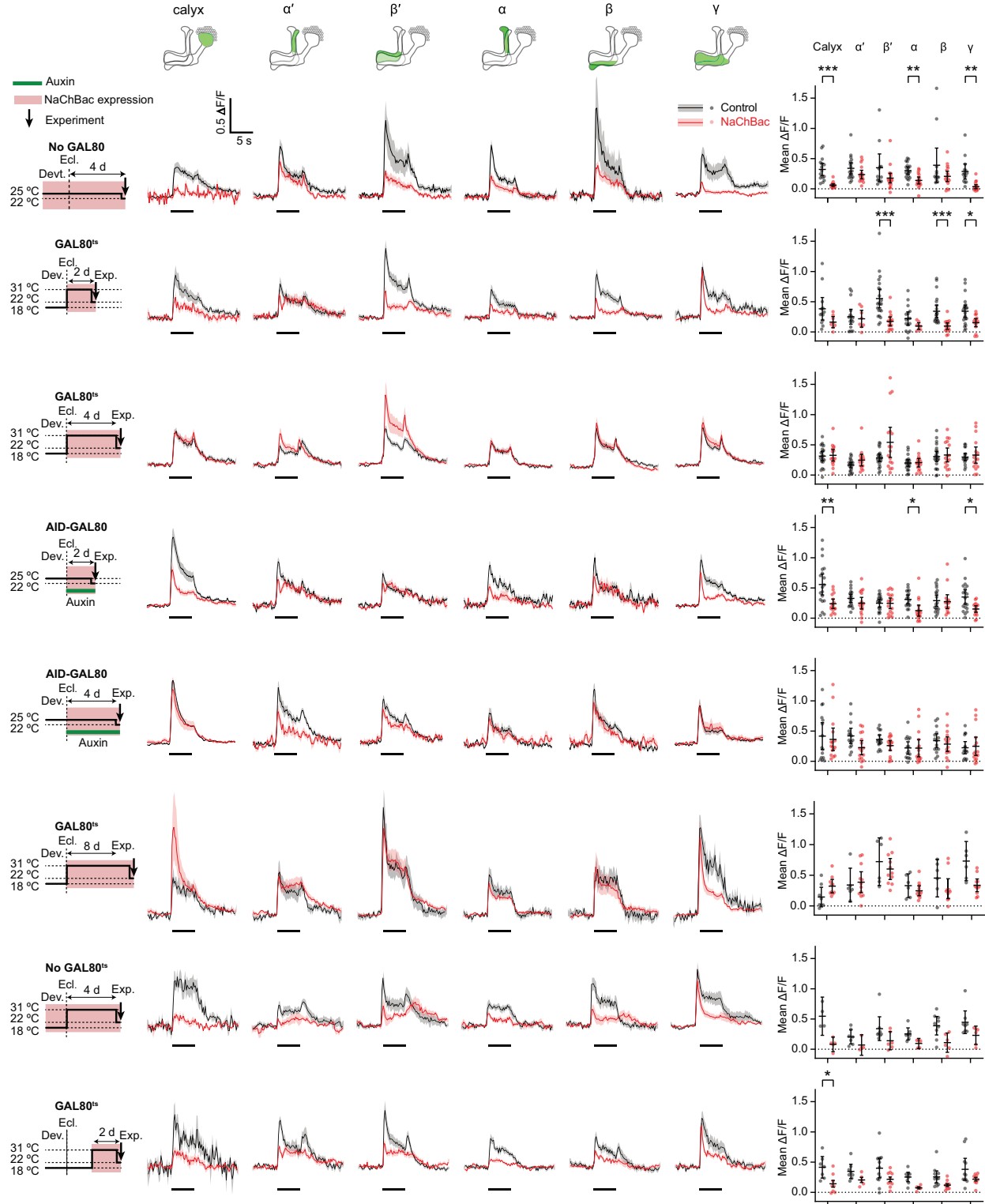

**Figure A6. Odour-evoked calcium influx reduced by constitutive or 2-d, but not 4-d, NaChBac expression: additional traces and graphs supporting Fig. 2**

Odour response traces to isoamyl acetate were recorded from the calyx and each lobe of the mushroom body under various experimental conditions, all using the R13F02-GAL4 driver. Left panels indicate experimental protocol,

where the shaded area shows NaChBac expression: constitutively at 25°C; 18°C during development then 2 days at 31°C with GAL80[ts]; 18°C then 4 days at 31°C with GAL80[ts]; 2 days auxin with AID-GAL80; 4 days auxin with AID-GAL80; 18°C then 8 days at 31°C GAL80[ts], control $n$ = 8(5) 8(5) 6(5) 6(5) 6(5) 6(5) NaChBac $n$ = 13(8) 16(8) 12(6) 12(6) 12(6) 12(6); 18°C then 4 days at 31°C without GAL80[ts], control $n$ = 4(3) 7(4) 8(4) 7(4) 10(5) 8(4), NaChBac $n$ = 5(3) 4(2) 6(3) 5(3) 6(3) 6(3); 18°C until 2 days after eclosion and then 2 days at 31°C with GAL80[ts], control $n$ = 8(6) 9(7) 12(7) 9(5) 14(7) 13(7), NaChBac $n$ = 10(5) 6(5) 12(7) 6(6) 12(6) 11(7). Odour stimulus duration is indicated by the black bar (5 s). Graphs show means ± 95% confidence intervals. *$P < 0.05$, **$P < 0.01$ and ***$P < 0.001$, Mixed-effects analysis, Geisser–Greenhouse correction and Sidak's multiple comparison test. Traces are means ± SEM.

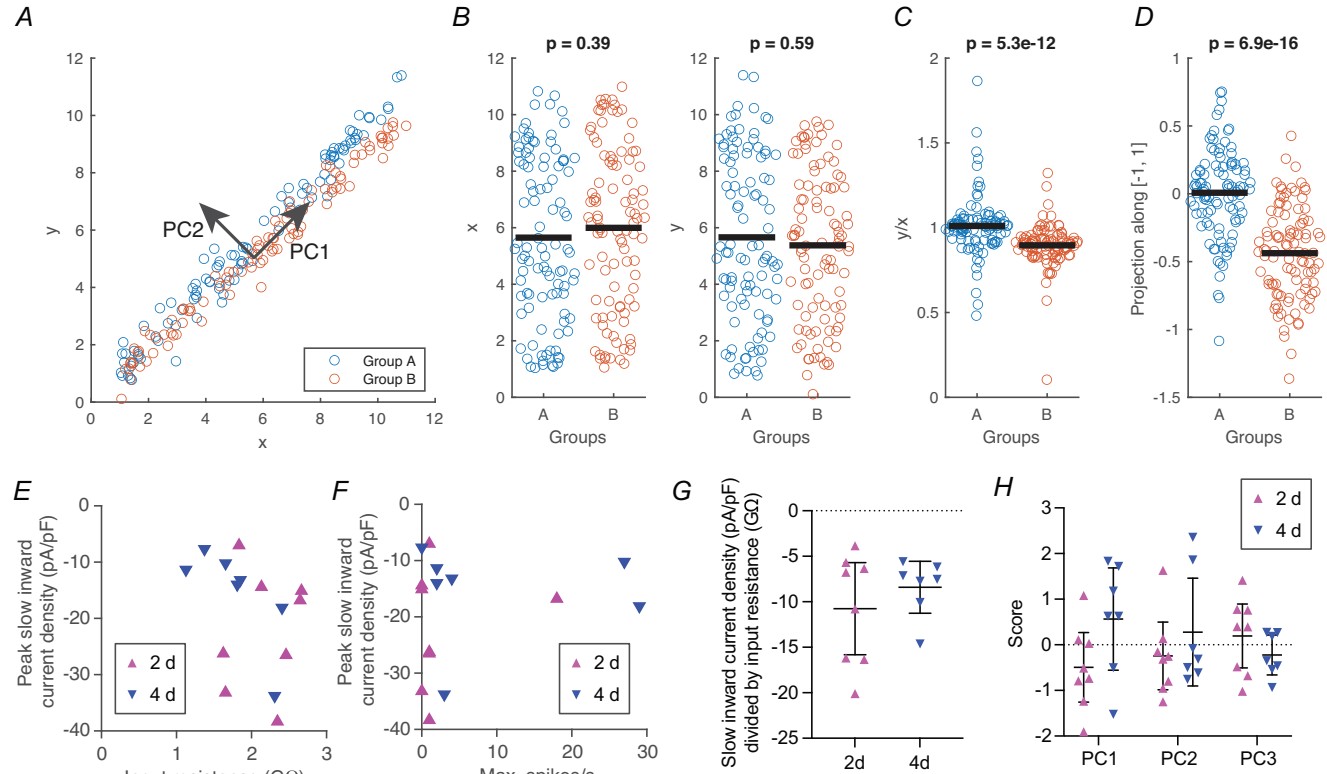

**Figure A7. Multi-parameter comparison of 2 *versus* 4 day NaChBac expression**

*A–D*, rationale for multi-parameter analysis: if data points exist in a parameter space where some parameters are highly correlated, two groups may not appear different when examining each parameter individually, but may be clearly different when plotting parameters against each other. *A*, in this theoretical example, group A has parameter $x$ sampled from the uniform distribution $U(1,11)$, and parameter $y$ is the same as $x$ plus Gaussian noise: $y = x + N(\mu = 0, \sigma = 0.2)$. Group B has $x = U(1,11)$, $y = 0.9*x + N(\mu = 0, \sigma = 0.2)$. Groups A and B are obviously different when both parameters are plotted together. PC1 and PC2 are the two principal components, PC1 = (1,1), PC2 = (−1,1). *B*, However, Groups *A* and *B* are very overlapping when considering parameters $x$ and $y$ on their own. *C*, the difference becomes clear and statistically significant when plotting $y/x$, which captures the fact that Group A has $y = x$ and Group B has $y = 0.9*x$. *D*, the difference is also clear when projecting the data onto the second principal component (*P*-values for *B–D*, Mann–Whitney test). *E*, cells with higher input resistance tended to have stronger NaChBac current density ($r = 0.34$). Magenta upward-pointing triangles, 2 day expression; blue downward-pointing triangles, 4 d expression. *F*, the few cells that spiked with NaChBac expression tended to have smaller NaChBac currents (correlation, spike rate *vs.* NaChBac current $r = -0.21$). *G*, applying the approach from panel (*C*): normalizing NaChBac current density to input resistance. *P* = 0.69, Mann–Whitney test. *H*, scores for each principal component (PC), for the parameters from (*D,E*). Principal components analysis was performed after normalizing the data to have mean 0 and standard deviation 1 for each parameter. The PC weights were, in the order (input resistance, NaChBac current density, spike rate): PC1 (−0.70, 0.71, 0.02); PC2, (0.36, 0.33, 0.87); PC3, (0.61, 0.62, −0.49). *P* = 0.0721, 0.463, 0.281, Mann–Whitney test. Graphs show means ± 95% confidence intervals. See Table A3 for details of statistical tests.

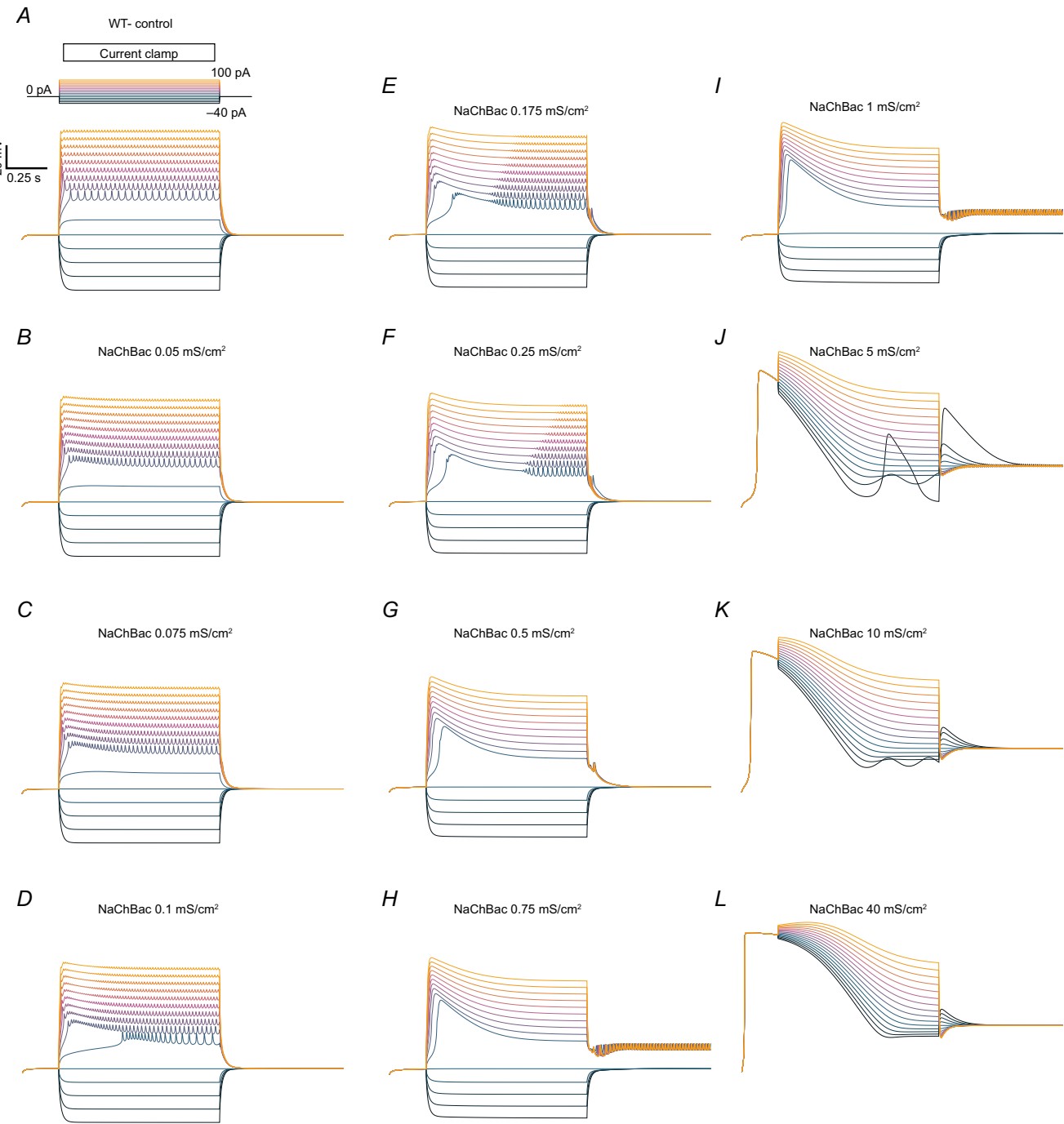

**Figure A8. Systematic expression of NaChBac alone does not recreate physiological response of NaChBac-expressing Kenyon cells**

*A*, current-clamp simulation, recording somatic responses to varying current injections using optimal parameter set of WT model from Fig. 9B. *B–L*, current-clamp simulations as in (*A*) but with increasing expression of NaChBac throughout all compartments of the model, without decreasing Para. Initial resting membrane potential for all models is –60 mV.

**Table A1. List of fly genotypes.**

| Figure | Shorthand name/purpose | Full genotype |
|---|---|---|
| 1A, A2 | Kenyon cell (KC)>NaChBac-GFP, mCherry | +; R13F02-GAL4, UAS-mCherry/UAS-NaChBac-GFP |
| 1B-I,7,8,A1,A3 | NaChBac and GFP in $\gamma$ KCs | Control: lexAop-GAL80/R44E04-LexA; mb247-GAL4, UAS-mCD8::GFP/+ |
| | | Experimental: lexAop-GAL80/R44E04-LexA; mb247-GAL4, UAS-mCD8::GFP/UAS-NaChBac |
| 1J | NaChBac and GCaMP6f in $\gamma$ KCs | Control: lexAop-GAL80, UAS-GCaMP6f/R44E04-LexA; mb247-GAL4/+ |
| | | Experimental: lexAop-GAL80, UAS-GCaMP6f/R44E04-LexA; mb247-GAL4/UAS-NaChBac |
| A4 | OK107>GFP | Control: UAS-Stinger/+; OK107-GAL4/+ |
| | R13F02>GFP | Experimental: UAS-Stinger/+; R13F02-GAL4/+ |
| A5 | No GCaMP6f (negative control) | +; R13F02-GAL4 |
| | R13F02>GCaMP6f (positive control) | UAS-GCaMP6f/+; R13F02-GAL4/+ |
| | R13F02>GCaMP6f, GAL80$^{ts}$ | UAS-GCaMP6f/+; R13F02-GAL4/tubP-GAL80$^{ts}$ |
| | R13F02>GCaMP6f, AID-GAL80 | UAS-GCaMP6f/+; R13F02-GAL4/AtTIR1-T2A-AID-GAL80-AID |
| 2,4,A6 | KC>NaChBac, GCaMP6f constitutive | Control: UAS-GCaMP6f/+; R13F02-GAL4/+ |
| | | Experimental: UAS-GCaMP6f/+; R13F02-GAL4/UAS-NaChBac |
| 2,4,A6 | KC>NaChBac, GCaMP6f with GAL80$^{ts}$ | Control: UAS-GCaMP6f/+; R13F02-GAL4/tubP-GAL80$^{ts}$ |
| | | Experimental: UAS-GCaMP6f/+; R13F02-GAL4, UAS-NaChBac/tubP-GAL80$^{ts}$ |
| 2,4,A6 | KC>NaChBac, GCaMP6f AGES (AID-GAL80 for auxin) | Control: UAS-GCaMP6f/+; R13F02-GAL4/AtTIR1-T2A-AID-GAL80-AID |
| | | Experimental: UAS-GCaMP6f/+; R13F02-GAL4, UAS-NaChBac/AtTIR1-T2A-AID-GAL80-AID |
| 3,7,8,A1,A3 | KC>NaChBac, GFP AGES (AID-GAL80 for auxin) | Control: UAS-mCD8::GFP/+; R13F02-GAL4/AtTIR1-T2A-AID-GAL80-AID |
| | | Experimental: UAS-mCD8::GFP/+; R13F02-GAL4, UAS-NaChBac/AtTIR1-T2A-AID-GAL80-AID |
| 5 | KC>NaChBac | Control: UAS-NaChBac/+ R13F02-GAL4/+ |
| | | Experimental: R13F02-GAL4, UAS-NaChBac/+ |
| 5 | KC>NaChBac AGES | Control: empty-GAL4(attP2), UAS-NaChBac/AtTIR1-T2A-AID-GAL80-AID |
| | | Experimental: R13F02-GAL4, UAS-NaChBac/AtTIR1-T2A-AID-GAL80-AID |
| 6 | paraFlpTag Constitutive NaChBac | Control: paraFlpTag/+; lexAop-FLP, mb247-dsRed/+; mb247-LexA/UAS-NaChBac |
| | | Experimental: paraFlpTag/+; lexAop-FLP, mb247-dsRed/+; mb247-LexA, R13F02-GAL4/UAS-NaChBac |
| 6 | paraFlpTag NaChBac with AGES | Control: paraFlpTag/+; lexAop-FLP, mb247-dsRed/+; mb247-LexA/UAS-NaChBac, AtTIR1-T2A-AID-GAL80-AID |
| | | Experimental: paraFlpTag/+; lexAop-FLP, mb247-dsRed/+; mb247-LexA, R13F02-GAL4/UAS-NaChBac,AtTIR1-T2A-AID-GAL80-AID |

**Table A2. Sources of transgenes**

| Transgene | Reference | Source |
|---|---|---|
| R13F02-GAL4 | Jenett et al., 2012 | BDSC 48571 |
| UAS-NaChBac-GFP | Nitabach et al., 2006 | BDSC 9466 |
| UAS-NaChBac | Nitabach et al., 2006 | BDSC 9468 |
| lexAop-GAL80 | | BDSC 32216 |
| R44E04-LexA | Bielopolski et al., 2019; Jenett et al., 2012 | BDSC 52736 |
| OK107-GAL4 | Connolly et al., 1996 | |
| UAS-Stinger-GFP | | Gift from Carolina Rezaval BDSC 84277 |
| UAS-GCaMP6f (attP40) | Chen et al., 2013 | BDSC 42747 |
| tubP-GAL80$^{ts}$ on III | McGuire et al., 2003 | |
| AtTIR1-T2A-AID-GAL80-AID | McClure et al., 2022 | Gift from Mirre Simons BDSC 92470 |
| empty-GAL4 (attP2) | | BDSC 68384 |
| mb247-LexA | Lin et al., 2014; Pitman et al., 2011 | |
| para[FlpTag] | Fendl et al., 2020 | Gift from Axel Borst |
| mb247-dsRed | Riemensperger et al., 2005 | |
| lexAop-FLP | Pitman et al., 2011 | |

**Table A3. Details of statistical tests**

| Figure | Condition | Breakdown | Statistical test | P-value | Significance |
|---|---|---|---|---|---|
| 1E | Max. spike rate | Control *vs*. NaChBac | Mann–Whitney test | 0.000583 | *** |
| 1G | Input resistance | Control *vs*. NaChBac | | 0.209 | ns |
| 1I | Peak current density | Control *vs*. NaChBac | | 0.0727 | ns |
| 1J | Odour Responses in $\gamma$KCs to isoamyl acetate | Calyx, control *vs*. NaChBac | | 0.000155 | *** |
| | | Lobes, control *vs*. NaChBac | | 0.00314 | ** |
| 2A | Constitutive expression, R13F02-GAL4> NaChBac, mean odour responses to isoamyl acetate | Calyx, control *vs*. NaChBac | Mixed-effects | 0.000358 | *** |
| | | Alpha', control *vs*. NaChBac | analysis of | 0.351 | ns |
| | | Beta', control *vs*. NaChBac | matched data – | 0.607 | ns |
| | | Alpha, control *vs*. NaChBac | Geisser–Greenhouse | 0.001403 | ** |
| | | Beta, control *vs*. NaChBac | correction – Sidak's | 0.731 | ns |
| | | Gamma, control *vs*. NaChBac | multiple | 0.00504 | ** |
| | | Main effect of lobes, control *vs*. NaChBac | comparisons test | 0.0367 | * |
| | | Main effect of genotype, control *vs*. NaChBac | | <0.0001 | **** |
| | | Interaction lobes × genotype | | 0.509 | ns |
| 2D | 2 days acute expression, R13F02-GAL4> NaChBac,Gal80ts, mean odour responses to isoamyl acetate | Calyx, control *vs*. NaChBac | Mixed-effects | 0.176 | ns |
| | | Alpha', control *vs*. NaChBac | analysis of | 0.999 | ns |
| | | Beta', control *vs*. NaChBac | matched data – | 0.000407 | *** |
| | | Alpha, Control *vs*. NaChBac | Geisser–Greenhouse | 0.228 | ns |
| | | Beta, Control *vs*. NaChBac | correction – Sidak's | 0.000554 | *** |
| | | Gamma, Control *vs*. NaChBac | multiple | 0.0134 | * |
| | | Main effect of lobes, Control *vs*. NaChBac | comparisons test | 0.00673 | ** |
| | | Main effect of genotype, Control *vs*. NaChBac | | <0.0001 | **** |
| | | Interaction lobes × genotype | | 0.0803 | ns |
| | 4 days acute expression, R13F02-GAL4> NaChBac,Gal80ts, mean odour responses to isoamyl acetate | Calyx, control *vs*. NaChBac | Mixed-effects | >0.999 | ns |
| | | Alpha', control *vs*. NaChBac | analysis of | 0.529 | ns |
| | | Beta', control *vs*. NaChBac | matched data – | 0.249 | ns |
| | | Alpha, control *vs*. NaChBac | Geisser–Greenhouse | >0.999 | ns |
| | | Beta, control *vs*. NaChBac | correction – Sidak's | 0.999 | ns |
| | | Gamma, control *vs*. NaChBac | multiple | 0.998 | ns |
| | | Main effect of lobes, control *vs*. NaChBac | comparison test | 0.000226 | *** |
| | | Main effect of genotype, control *vs*. NaChBac | | 0.167 | ns |
| | | Interaction lobes × genotype | | 0.0189 | * |
| | 2 days acute expression, R13F02-GAL4> NaChBac, AID-Tagged Gal80, mean odour responses to isoamyl acetate | Calyx, control *vs*. NaChBac | Mixed-effects | 0.00991 | ** |
| | | Alpha', control *vs*. NaChBac | analysis of | 0.657 | ns |
| | | Beta', control *vs*. NaChBac | matched data – | >0.999 | ns |
| | | Alpha, control *vs*. NaChBac | Geisser–Greenhouse | 0.0119 | * |
| | | Beta, control *vs*. NaChBac | correction – Sidak's | >0.999 | ns |
| | | Gamma, control *vs*. NaChBac | multiple | 0.0207 | * |
| | | Main effect of lobes, control *vs*. NaChBac | comparison test | 0.00197 | ** |
| | | Main effect of genotype, control *vs*. NaChBac | | 0.00263 | ** |
| | | Interaction lobes × genotype | | 0.000486 | *** |
| | 4 days acute expression, R13F02-GAL4> NaChBac, AID-Tagged Gal80, mean odour responses to isoamyl acetate | Calyx, control *vs*. NaChBac | Mixed-effects | 0.993 | ns |
| | | Alpha', control *vs*. NaChBac | analysis of | 0.137 | ns |
| | | Beta', control *vs*. NaChBac | matched data – | 0.786 | ns |
| | | Alpha, control *vs*. NaChBac | Geisser–Greenhouse | >0.999 | ns |
| | | Beta, control *vs*. NaChBac | correction – Sidak's | 0.981 | ns |
| | | Gamma, control *vs*. NaChBac | multiple | 0.999 | ns |
| | | Main effect of lobes, control *vs*. NaChBac | comparison test | 0.0115 | * |
| | | Main effect of genotype, control *vs*. NaChBac | | 0.3601 | ns |
| | | Interaction lobes × genotype | | 0.0919 | ns |

*(Continued)*

**Table A3. (Continued)**

| Figure | Condition | Breakdown | Statistical test | P-value | Significance |
|---|---|---|---|---|---|
| for Figure 2E see end | | | | | |
| 3B | Max. spike rate | Comparing all conditions | Kruskal–Wallis test | <0.0001 | **** |
| | | WT 2 d *vs.* NaChBac 2 d | Dunn's multiple | <0.0001 | **** |
| | | WT 4 d *vs.* NaChBac 4 d | comparisons test | 0.00451 | ** |
| | | WT 2 d *vs.* WT 4 d | | >0.999 | ns |
| | | NaChBac 2 d *vs.* NaChBac 4 d | | >0.999 | ns |
| 3D | Input resistance | Comparing all conditions | Kruskal–Wallis test | 0.138 | ns |
| 3F | Peak current density | Comparing all conditions | Kruskal–Wallis test | <0.0001 | **** |
| | | WT 2 d *vs.* NaChBac 2 d | Dunn's multiple | <0.0001 | **** |
| | | WT 4 d *vs.* NaChBac 4 d | comparisons test | 0.00908 | ** |
| | | WT 2 d *vs.* WT 4 d | | >0.9999 | ns |
| | | NaChBac 2 d *vs.* NaChBac 4 d | | >0.9999 | ns |
| 3G | Current to induce NaChBac potential | Comparing all conditions | Kruskal–Wallis test | 0.269 | ns |
| 3H | Mean depolarization | Comparing all conditions | Kruskal–Wallis test | 0.231 | ns |
| 3J | NaChBac current decay time constant | Comparing all conditions | Kruskal–Wallis test | 0.519 | ns |
| 4D | Correlation | Constitutive NaChBac expression, control *vs.* NaChBac | Mann–Whitney test | 0.000128 | *** |
| | | 2 Days acute NaChBac expression – Gal80ts, control *vs.* NaChBac | | <0.0001 | **** |
| | | 4 Days acute NaChBac expression – Gal80ts, Control *vs.* NaChBac | | 0.000167 | *** |
| | | 2 Days acute NaChBac expression – AuxinGal80, control *vs.* NaChBac | | 0.0049 | ** |
| | | 4 Days acute NaChBac expression – AuxinGal80, control *vs.* NaChBac | | 0.0956 | ns |
| 4E | Sparseness | Constitutive NaChBac expression, control *vs.* NaChBac | Mann–Whitney test | 0.00188 | ** |
| | | 2 Days acute NaChBac expression – Gal80ts, control *vs.* NaChBac | | <0.0001 | **** |
| | | 4 Days acute NaChBac expression – Gal80ts, control *vs.* NaChBac | | 0.0141 | * |
| | | 2 Days acute NaChBac Expression – AuxinGal80, Control *vs.* NaChBac | | 0.00448 | ** |
| | | 4 Days acute NaChBac expression – AuxinGal80, control *vs.* NaChBac | | 0.0321 | * |
| 4F | Average odour response | Constitutive NaChBac expression, control *vs.* NaChBac | Mann–Whitney test | 0.00191 | ** |
| | | 2 Days acute NaChBac expression – Gal80ts, control *vs.* NaChBac | | <0.0001 | **** |
| | | 4 Days acute NaChBac expression – Gal80ts, control *vs.* NaChBac | | 0.00248 | ** |
| | | 2 Days acute NaChBac expression – AuxinGal80, control *vs.* NaChBac | | 0.0202 | * |
| | | 4 Days acute NaChBac expression – AuxinGal80, control *vs.* NaChBac | | 0.00799 | ** |

(*Continued*)

**Table A3. (Continued)**

| Figure | Condition | Breakdown | Statistical test | *P*-value | Significance |
|---|---|---|---|---|---|
| 5B | Decision changes | Constitutive NaChBac expression *vs.* R13F02-GAL4 | Kruskal–Wallis test, Dunn's multiple | 0.000445 | *** |
| | | Constitutive NaChBac expression *vs.* NaChBac alone | comparisons test | <0.0001 | **** |
| | | R13F02-GAL4 | One sample | <0.0001 | **** |
| | | UAS-NaChBac | Wilcoxon's test | <0.0001 | **** |
| | | R13F02-GAL4 > UAS-NaChBac | with Holm–Bonferroni correction (different from 0) | 0.907 | ns |
| 5C | Behaviour – constitutive – decision before (%) | R13F02-GAL4 *vs.* R13F02-GAL4 > UAS-NaChBac | Kruskal–Wallis test, Dunn's multiple | >0.999 | ns |
| | | UAS-NaChBac *vs.* R13F02-GAL4 > NaChBac | comparisons test | 0.243 | ns |
| | | R13F02-GAL4 | One sample | 0.02604 | ** |
| | | UAS-NaChBac | Wilcoxon's test | <0.0001 | **** |
| | | R13F02-GAL4, UAS-NaChBac | with Holm–Bonferroni correction (different from 50) | <0.0001 | **** |
| 5E | Decision changes | 2 Days NaChBac expression | Kruskal–Wallis test, Dunn's multiple | 0.002102 | ** |
| | | 4 Days NaChBac expression | comparisons test | 0.004605 | ** |
| | | 2 days ctrl | One sample Wilcoxon's test | <0.0001 | **** |
| | | 2 days NaChBac | with Holm–Bonferroni | 0.00128 | **** |
| | | 4 days ctrl | correction | <0.0001 | **** |
| | | 4 days NaChBac | (different from 0) | <0.0001 | **** |
| 5F | Behaviour – AGES – decision before (%) | Day 2 Ctrl *vs.* day 2 Exp | Kruskal–Wallis test, Dunn's multiple | 0.752 | ns |
| | | Day 4 Ctrl *vs.* day 4 Exp | comparisons test | 0.0697 | ns |
| | | Empty-GAL4, UAS-NaChBac, AID-GAL80 2 days | One sample Wilcoxon's test | <0.0001 | **** |
| | | R13F02-GAL4, UAS-NaChBac, AID-GAL80 2 days | with Holm–Bonferroni | <0.0001 | **** |
| | | Empty-GAL4, UAS-NaChBac, AID-GAL80 4 days | correction (different from 50) | <0.0001 | **** |
| | | R13F02-GAL4, UAS-NaChBac, AID-GAL80 4 days | | <0.0001 | **** |
| 6D | ParaFlpTag, constitutive expression of NaChBac, normalized to dsRed | Horizontal control *vs.* horizontal NaChBac | Mixed-effects analysis | <0.0001 | **** |
| | | Vertical control *vs.* vertical NaChBac | | <0.0001 | **** |
| 6E | ParaFlpTag, constitutive expression of NaChBac, peduncle | GFP fluorescence normalized to dsRed of the peduncle | Mann–Whitney's test | <0.0001 | **** |
| 6F | ParaFlpTag, constitutive expression of NaChBac, peduncle | Control GFP fluorescence *vs.* NaChBac-GFP fluorescence | Kruskal–Wallis test, Dunn's multiple | 0.001 | *** |
| | | Control dsRed fluorescence *vs.* NaChBac dsRed fluorescence | comparisons test | 0.7021 | ns |

*(Continued)*

**Table A3. (Continued)**

| Figure | Condition | Breakdown | Statistical test | P-value | Significance |
|---|---|---|---|---|---|
| 6G | ParaFlpTag, 2 and 4 days acute expression of NaChBac, peduncle | 2 Days: Control GFP fluorescence *vs.* NaChBac-GFP fluorescence | Mann–Whitney's test | 0.00414 | ** |
| | | 4 Days: Control GFP fluorescence *vs.* NaChBac-GFP fluorescence | | 0.00613 | ** |
| 6H | ParaFlpTag, 2 and 4 days acute expression of NaChBac, peduncle | 2 Days: Control GFP fluorescence *vs.* NaChBac-GFP fluorescence | Kruskal–Wallis test, Dunn's multiple | 0.3824 | ns |
| | | 2 Days: Control dsRed fluorescence *vs.* NaChBac dsRed fluorescence | comparisons test | >0.999 | ns |
| | | 4 Days: Control GFP fluorescence *vs.* NaChBac-GFP fluorescence | Kruskal–Wallis test, Dunn's multiple | 0.040003 | * |
| | | 4 Days: Control dsRed fluorescence *vs.* NaChBac dsRed fluorescence | comparisons test | >0.999 | ns |
| 7B | Minimum somatic voltage at spike onset | Main effect of genotype | 2-way ANOVA with Sidak's multiple comparisons test | <0.0001 | **** |
| | | Main effect of auxin *vs.* constitutive | | 0.00331 | **** |
| | | Interaction | | 0.007403 | *** |
| | | Auxin, control *vs.* NaChBac | | 0.0428 | * |
| | | Constitutive, control *vs.* NaChBac | | <0.0001 | **** |
| 7E | Spike height | Main effect of genotype | 2-way ANOVA with Sidak's multiple comparisons test | 0.00508 | ** |
| | | Main effect of auxin *vs.* constitutive | | 0.199 | ns |
| | | Interaction | | 0.213 | ns |
| | | Auxin, control *vs.* NaChBac | | 0.00407 | ** |
| | | Constitutive, control *vs.* NaChBac | | 0.473 | ns |
| 7F | Peak d$V$/d$t$ | Main effect of genotype | 2-way ANOVA with Sidak's multiple comparisons test | 0.00555 | ** |
| | | Main effect of auxin *vs.* constitutive | | 0.299 | ns |
| | | Interaction | | 0.149 | ns |
| | | Auxin, control *vs.* NaChBac | | 0.00235 | ** |
| | | Constitutive, control *vs.* NaChBac | | 0.579 | ns |
| 7G | Spike height across voltages at spike onset | Control *vs.* NaChBac | Mixed-effects model | 0.00134 | ** |
| | | Effect of voltage | | <0.0001 | **** |
| 7H | Peak d$V$/d$t$ across voltages at spike onset | Control *vs.* NaChBac | Mixed-effects model | 0.00827 | ** |
| | | Effect of voltage | | <0.0001 | **** |
| 7K | IV curves of fast inward currents | Auxin, control (95% CI of slope: 0.0796 to 0.2318) | Linear regression, slope significantly non-zero? | <0.0001 | **** |
| | | Auxin, NaChBac (95% CI of slope: −0.00436 to 0.1443) | | 0.0635 | ns |
| | | Constitutive, control (95% CI of slope: 0.00768 to 0.1757) | | 0.0339 | * |
| | | Constitutive, NaChBac (95% CI of slope: −0.0650 to 0.2421) | | 0.238 | ns |
| 7L | Fast inward current amplitude | Comparing all conditions | Kruskal–Wallis test | 0.00543 | ** |
| | | Auxin, control *vs.* NachBac | Dunn's multiple | 0.0129 | * |
| | | Constitutive, control *vs.* NaChBac | comparisons test | 0.3203 | ns |
| 7M | | Spike height *vs.* fast inward current amplitude | Pearson's correlation coefficient | $r = 0.801$ | $P < 0.0001$ |
| 7N | | Spike peak d$V$/d$t$ *vs.* fast inward current amplitude | Pearson's correlation coefficient | $r = 0.831$ | $P < 0.0001$ |
| 7O | | Total spikes +10 to +100 pA *vs.* fast inward current amplitude | Pearson's correlation coefficient | $r = 0.618$ | $P < 0.0001$ |

*(Continued)*

**Table A3.  (Continued)**

| Figure | Condition | Breakdown | Statistical test | *P*-value | Significance |
|---|---|---|---|---|---|
| 7P | | Residual NaChBac current in last 100 ms *vs.* fast inward current amplitude (NaChBac cells only) | Pearson's correlation coefficient | $r = -0.178$ | $P = 0.4802$ |
| 8B | Steady-state outward current minus leak and NaChBac at +30 mV | Comparing all conditions | Kruskal–Wallis test | 0.212 | ns |
| 9H | Transient sodium conductance in best-fit models | WT *vs.* NaChBac | Unpaired *t*-test | 0.0245 | * |
| A1A | | NaChBac current *vs.* input resistance (NaChBac cells only) | Pearson's correlation coefficient | $r = -0.606$ | $P = 0.000113$ |
| A1B | | NaChBac current density *vs.* input resistance (NaChBac cells only) | Pearson's correlation coefficient | $r = -0.557$ | $P = 0.000516$ |
| A1D | | NaChBac current *vs.* series resistance (NaChBac cells only) | Pearson's correlation coefficient | $r = 0.257$ | $P = 0.136$ |
| A1G | | Input resistance *vs.* series resistance | Pearson's correlation coefficient | $r = -0.249$ | $P = 0.0408$ |
| A2 | Coefficient of variation from normalized fluorescence | NaChBac-GFP vs mCherry | Mann–Whitney's test | 0.5737 | ns |
| A3-Fig.1E | Max. spike rate | Control *vs.* NaChBac | Mann–Whitney's test | <0.0001 | **** |
| A3-Fig.1G | Input resistance | Control *vs.* NaChBac | Mann–Whitney's test | 0.931 | ns |
| A3-Fig.1H | Peak current density | Control *vs.* NaChBac | Mann–Whitney's test | 0.0419 | * |
| A3-Fig.3B | Max. spike rate | Comparing all conditions | Kruskal–Wallis test | <0.0001 | **** |
| | | wt 2 d *vs.* NaChBac 2 d | Dunn's multiple comparisons test | <0.0001 | **** |
| | | wt 4 d *vs.* NaChBac 4 d | | 0.00432 | ** |
| | | wt 2 d *vs.* wt 4 d | | >0.999 | ns |
| | | NaChBac 2 d *vs.* NaChBac 4 d | | 0.429 | ns |
| A3-Fig.3D | Input resistance | Comparing all conditions | Kruskal–Wallis test | 0.877 | ns |
| A3-Fig.3F | Peak current density | Comparing all conditions | Kruskal–Wallis test | <0.0001 | **** |
| | | wt 2 d *vs.* NaChBac 2 d | Dunn's multiple comparisons test | <0.0001 | **** |
| | | wt 4 d *vs.* NaChBac 4 d | | 0.00433 | ** |
| | | wt 2 d *vs.* wt 4 d | | >0.9999 | ns |
| | | NaChBac 2 d *vs.* NaChBac 4 d | | >0.9999 | ns |
| A3-Fig.3G | Current to induce NaChBac potential | Comparing all conditions | Kruskal–Wallis test | 0.269 | ns |
| A3-Fig.3H | Mean depolarization | Comparing all conditions | Kruskal–Wallis test | 0.231 | ns |
| A3-Fig.3J | NaChBac current decay time constant | Comparing all conditions | Kruskal–Wallis test | 0.519 | ns |
| A4A | Kenyon cell count | R13F02-GAL4 and OK107 Kenyon Cell Count | Mann–Whitney's Test | 0.818 | ns |
| A5 | Levels of GCaMP fluorescence | Day 0 *vs.* Day 2 auxin exposure | Kruskal–Wallis test, Dunn's multiple comparisons test | 0.00536 | ** |
| | | Day 0 *vs.* Day 4 auxin exposure | | <0.0001 | **** |
| | | Day 2 *vs.* Day 4 auxin exposure | | >0.999 | ns |
| | | Day 0 *vs.* Day 2 31°C exposure | | 0.00176 | ** |
| | | Day 0 *vs.* Day 4 31°C exposure | | <0.0001 | **** |
| | | Day 2 *vs.* Day 4 31°C exposure | | >0.999 | ns |
| | | PC2 | Mann–Whitney's test | 0.463 | ns |
| | | PC3 | Mann–Whitney's test | 0.281 | ns |

*(Continued)*

**Table A3. (Continued)**

| Figure | Condition | Breakdown | Statistical test | *P*-value | Significance |
|---|---|---|---|---|---|
| A6 | 8 Days<br>R13F02-GAL4 ><br>NaChBac,Gal80[ts] | Calyx, Control *vs*. NaChBac<br>Alpha', Control *vs*. NaChBac<br>Beta', Control *vs*. NaChBac<br>Alpha, Control *vs*. NaChBac<br>Beta, Control *vs*. NaChBac<br>Gamma, Control *vs*. NaChBac<br>Main effect of lobes, Control *vs*. NaChBac<br>Main effect of genotype, Control *vs*. NaChBac<br>Interaction lobes × genotype | Mixed-effects<br>analysis of<br>matched data –<br>Geisser–Greenhouse<br>correction – Sidak's<br>multiple<br>comparison test | 0.326<br>0.993<br>>0.999<br>0.981<br>0.954<br>0.4804<br>0.00114<br>0.156<br>0.105 | ns<br>ns<br>ns<br>ns<br>ns<br>ns<br>**<br>ns<br>ns |
| | 4 Days temp control<br>Days<br>R13F02-GAL4 ><br>NaChBac | Calyx, Control *vs*. NaChBac<br>Alpha', Control *vs*. NaChBac<br>Beta', Control *vs*. NaChBac<br>Alpha, Control *vs*. NaChBac<br>Beta, Control *vs*. NaChBac<br>Gamma, Control *vs*. NaChBac<br>Main effect of lobes, Control *vs*. NaChBac<br>Main effect of genotype, Control *vs*. NaChBac<br>Interaction lobes × genotype | Mixed-effects<br>analysis of<br>matched data –<br>Geisser–Greenhouse<br>correction – Sidak's<br>multiple<br>comparisons test | 0.0695<br>0.4705<br>0.374<br>0.0669<br>0.0522<br>0.264<br>0.0966<br>0.000147<br>0.249 | ns<br>ns<br>ns<br>ns<br>ns<br>ns<br>ns<br>***<br>ns |
| | 2 Days 18–2 days 31 –<br>age control<br>R13F02-GAL4 ><br>NaChBac, Gal80[ts] | Calyx, Control *vs*. NaChBac<br>Alpha', Control *vs*. NaChBac<br>Beta', Control *vs*. NaChBac<br>Alpha, Control *vs*. NaChBac<br>Beta, Control *vs*. NaChBac<br>Gamma, Control *vs*. NaChBac<br>Main effect of lobes, Control *vs*. NaChBac<br>Main effect of genotype, Control *vs*. NaChBac<br>Interaction lobes × genotype | Mixed-effects<br>analysis of<br>matched data –<br>Geisser–Greenhouse<br>correction – Sidak's<br>multiple<br>comparison test | 0.0134<br>0.623<br>0.06401<br>0.369<br>0.324<br>0.208<br>0.0369<br>0.000201<br>0.0858 | *<br>ns<br>ns<br>ns<br>ns<br>ns<br>*<br>***<br>ns |
| A7G | NaChBac current<br>divided by input<br>resistance | 2 d *vs*. 4 d auxin | Mann–Whitney's test | 0.694 | ns |
| A7H | Scores along<br>principal<br>components | PC1<br>PC2<br>PC3 | Mann–Whitney's test<br>Mann–Whitney's test<br>Mann–Whitney's test | 0.0721<br>0.463<br>0.281 | ns<br>ns<br>ns |

| Figure | Condition | Predicted mean difference | Upper 95% CI limit | Lower<br>95% CI<br>limit | Significance |
|---|---|---|---|---|---|
| 2E | Constitutive | −0.192 | −0.2651 | −0.119 | **** |
| | 2 Days (Gal80[ts]) | −0.2222 | −0.1221 | −0.322 | **** |
| | 4 Days (Gal80[ts]) | 0.06166 | 0.1504 | −0.0271 | ns |
| | 2 Days (AID-tagged<br>Gal80) | −0.1344 | −0.05025 | −0.219 | ** |
| | 4 Days (AID-tagged<br>Gal80) | −0.06417 | 0.0871 | −0.209 | ns |
| | Constitutive (4 Days,<br>31°C) | −0.2394 | −0.3373 | −0.142 | *** |
| | 2 Days 31°C (4 Days<br>Adult, Gal80[ts]) | −0.1796 | −0.2633 | −0.0959 | *** |

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

## Additional information

### Data availability statement

Custom software is available at the links given in the Methods. The raw data reported in the figures are given in Dataset S1.

### Competing interests

The authors declare no competing interests.

### Author contributions

Conceptualization: K.G.-W., J.A., M.P, A.C.L. Formal analysis, K.G.-W., E.R., A.M.-S., M.W.T., K.K., A.C.L. Funding acquisition, J.A., M.P., A.C.L. Investigation, K.G.-W., E.R., A.M.-S., M.W.T. Methodology, K.G.-W., A.M.-S., K.K., A.C.L. Software, A.M.-S., K.K., A.C.L. Supervision, J.A., M.P., A.C.L. Visualization, K.G.-W., E.R., A.M.-S., M.W.T., A.C.L. Writing – original draft, K.G.-W., A.M.-S., A.C.L. Writing – review & editing. K.G.-W., E.R., A.M.-S., M.W.T., K.K., J.A., M.P., A.C.L. Imaging and behaviour experiments were performed in A.C.L.'s laboratory. Electrophysiology experiments were performed in M.P.'s laboratory and analysed in M.P.'s and A.C.L.'s laboratories. Compartmental modelling was performed in J.A.'s laboratory. All authors approved the final version of the manuscript and agree to be accountable for all aspects of the work in ensuring that questions related to the accuracy or integrity of any part of the work are appropriately investigated and resolved. All persons designated as authors qualify for authorship, and all those who qualify for authorship are listed.

### Funding

This work was supported by the European Research Council (ERC; 639489 to AL, 101085605 to MP), Biotechnology and Biological Sciences Research Council (BB/S016031/1, BB/X000273/1, BB/X014568/1 to AL), a BBSRC White Rose PhD studentship (BB/M011151/1 to KGW), a Wellcome Trust Discovery Award (225814/Z/22/Z to AL), the Israel Science Foundation (ISF; 404/23 to MP), the United States–Israel Binational Science Foundation (BSF; 2020636 to MP) and the NIH/NINDS (R01NS118562 to JA).

### Acknowledgements

The authors thank members of the Lin and Juusola labs for useful discussions; Axel Borst, Carolina Rezaval, Mirre Simons, and the Bloomington Drosophila Stock Centre for fly stocks; the Wolfson Light Microscopy Facility for support with confocal imaging, and Kath Whitley, Cherry O'Keefe, and Aaron Kirkby-Nowogorsky for technical assistance. For the purpose of open access, the authors have applied a Creative Commons

## Keywords

drosophila, homeostatic plasticity, ion channel, Kenyon cell, mushroom body, NaChBac, sodium channel

## Supporting information

Additional supporting information can be found online in the Supporting Information section at the end of the HTML view of the article. Supporting information files available:

**Peer Review History**
Dataset S1: Raw data underlying all graphs

