## [Peer Review History · The Journal of Physiology]

Ectopic sodium channel expression decreases excitability of *Drosophila* Kenyon cells

Katie Greenin-Whitehead, Eyal Rozenfeld, Anthony Moreno-Sanchez, Melissa W Tan, Kurtulus Kullu, Jessica Ausborn, Moshe Parnas, and Andrew C Lin

DOI: 10.1113/JP288790

Corresponding author(s): Andrew Lin (andrew.lin@sheffield.ac.uk)

The following individual(s) involved in review of this submission have agreed to reveal their identity: Eve Marder (Referee #2)

Review Timeline:

Submission Date:	22-Feb-2025
Editorial Decision:	25-Mar-2025
Revision Received:	17-Jun-2025
Accepted:	16-Jul-2025

Senior Editor: Nathan Schoppa

Reviewing Editor: Nathan Schoppa

Transaction Report:

Dear Dr Lin,

Re: JP-RP-2025-288790 "**Ectopic sodium channel expression paradoxically decreases excitability of *Drosophila* Kenyon cells**" by Katie Greenin-Whitehead, Eyal Rozenfeld, Anthony Moreno-Sanchez, Melissa W Tan, Kurtulus Kullu, Jessica Ausborn, Moshe Parnas, and Andrew C Lin

Thank you for submitting your manuscript to The Journal of Physiology. It has been assessed by a Reviewing Editor and by 2 expert referees and we are pleased to tell you that it is potentially acceptable for publication following satisfactory major revision.

LANGUAGE EDITING AND SUPPORT FOR PUBLICATION: If you would like help with English language editing, or other article preparation support, Wiley Editing Services offers expert help, including English Language Editing, as well as translation, manuscript formatting, and figure formatting at www.wileyauthors.com/eoo/preparation. You can also find resources for Preparing Your Article for general guidance about writing and preparing your manuscript at www.wileyauthors.com/eoo/prepresources.

REVISION CHECKLIST:

We look forward to receiving your revised submission.

Yours sincerely,

Nathan Schoppa
Senior Editor
The Journal of Physiology

REQUIRED ITEMS

- Author photo and profile. First or joint first authors are asked to provide a short biography (no more than 100 words for one author or 150 words in total for joint first authors) and a portrait photograph. These should be uploaded and clearly labelled together in a Word document with the revised version of the manuscript. See Information for Authors for further details.

- You must start the Methods section with a paragraph headed Ethical approval (https://jp.msubmit.net/cgi-bin/main.plex?form_type=display_requirements#methods).

Research must comply with The Journal's policies regarding animal experiments (<https://physoc.onlinelibrary.wiley.com/hub/animal-experiments>) and adherence to these policies must be stated in the manuscript.

Authors should confirm in their Methods section that their experiments were carried out according to the guidelines laid down by their institution's animal welfare committee, including an ethics approval reference number. The Methods section must contain a statement about access to food, water and housing, details of the anaesthetic regime: anaesthetic used, dose and route of administration, and method of killing the experimental animals.

- Your manuscript must include a complete Additional Information section, including competing interests; funding; author contributions and acknowledgements.

- Please upload separate high-quality figure files via the submission form.

- You must upload original, uncropped western blot/gel images (including controls) if they are not included in the manuscript. This is to confirm that no inappropriate, unethical or misleading image manipulation has occurred. These should be uploaded as 'Supporting information for review process only'. Please label/highlight the original gels so that we can clearly see which sections/lanes have been used in the manuscript figures. For more information, see: <https://physoc.onlinelibrary.wiley.com/hub/journal-policies#imagmanip>.

- Please ensure that any tables are editable and in Word format, and wherever possible, embedded in the article file itself.

- Please ensure that the Article File you upload is a Word file.

- Papers must comply with the Statistics Policy: https://jp.msubmit.net/cgi-bin/main.plex?form_type=display_requirements#statistics.

In summary:

- If n {less than or equal to} 30, all data points must be plotted in the figure in a way that reveals their range and distribution. A bar graph with data points overlaid, a box and whisker plot or a violin plot (preferably with data points included) are acceptable formats.
- If $n > 30$, then the entire raw dataset must be made available either as supporting information, or hosted on a not-for-profit repository, e.g. FigShare, with access details provided in the manuscript.
- 'n' clearly defined (e.g. x cells from y slices in z animals) in the Methods. Authors should be mindful of pseudoreplication.
- All relevant 'n' values must be clearly stated in the main text, figures and tables.
- The most appropriate summary statistic (e.g. mean or median and standard deviation) must be used. Standard Error of the Mean (SEM) alone is not permitted.
- Exact p values must be stated. Authors must not use 'greater than' or 'less than'. Exact p values must be stated to three significant figures even when 'no statistical significance' is claimed.

- Please include an Abstract Figure file, as well as the Figure Legend text within the main article file. The Abstract Figure is a piece of artwork designed to give readers an immediate understanding of the research and should summarise the main conclusions. If possible, the image should be easily 'readable' from left to right or top to bottom. It should show the physiological relevance of the manuscript so readers can assess the importance and content of its findings. Abstract Figures should not merely recapitulate other figures in the manuscript. Please try to keep the diagram as simple as possible and without superfluous information that may distract from the main conclusion(s). Abstract Figures must be provided by authors no later than the revised manuscript stage and should be uploaded as a separate file during online submission labelled as File Type 'Abstract Figure'. Please also ensure that you include the figure legend in the main article file. All Abstract Figures should be created using BioRender. Authors should use The Journal's premium BioRender account to export high-resolution images. Details on how to use and access the premium account are included as part of this email.

EDITOR COMMENTS

Your study of homeostatic regulation of neuronal excitability in Kenyon cells in *Drosophila* has been reviewed by two expert reviewers. They were generally quite positive about the manuscript, citing its potential relevance for scientists using ectopic channel expression to alter neuronal excitability, as well as the general field of homeostatic plasticity in neurons. Many of the results were considered to be of high quality. The reviewers did however raise a number of concerns, all of which would need to be addressed in a revised manuscript. The most important points included:

1. As requested by both reviewers, additional information is needed about the electrophysiological recordings, including several recording parameters as well as electrophysiological variables examined.
2. As suggested by Reviewer 1, the authors should explore whether some of the heterogeneity in results such as spike rate can be explained by correlating them with other electrophysiological parameters.
3. The authors should examine other potential explanations for the reduced excitability following ectopic sodium channel expression, including changes in outward potassium currents and depolarization-induced block of sodium currents.
4. The Discussion section would be improved with some additional discussion of the relevance of this study in flies for mammals. As pointed out by Reviewer 2, the fly system has been shown to be somewhat of an outlier in studies of synaptic homeostasis. Also, comments from both reviewers about the use of terms such as "paradoxical" should be heeded.

5. The Methods section should be clear that most results, except the calcium traces, are depicted with error bars that reflect SD. Also, the authors should report exact p-values, except if $p < 0.001$. All journal policies regarding statistics and presentation of data should be followed.

REFeree COMMENTS

Referee #1:

Employing a combination of patch-clamp electrophysiology, in vivo calcium imaging, olfactory learning, immunohistochemistry, genetics, and compartmental modeling, Greenin-Whitehead and colleagues investigated how overexpression of the bacterial voltage-gated sodium channel NaChBac impacts neural excitability and olfactory responses of Kenyon cells, as well as aversive olfactory learning in adult *Drosophila*. Constitutive, or NaChBac expression for 2 and 4 days in adult Kenyon cells decreased action potential (AP) number and sparsened odor-induced calcium responses in Kenyon cells. Moreover, NaChBac expression impaired aversive olfactory conditioning. Odor-induced calcium responses in Kenyon cells were reduced after constitutive, or NaChBac expression for 2 days, but not for 4 days. Furthermore, constitutive NaChBac expression decreased the levels of the sodium channel paralytic (para), and compartmental modeling reproduced the excitability phenotype induced by NaChBac expression when considering a decrease in a transient para-like sodium conductance. Modeling also suggests that low NaChBac expression levels increase, while high expression levels decrease AP number. Together, they propose that (strong) ectopic overexpression of a sodium channel leads to a decrease, rather than an increase in excitability and odor-evoked responses in Kenyon cells, as well as impaired olfactory learning.

NaChBac expression is a commonly used tool to enhance neuronal excitability. Hence, the results of this study impact the interpretation of previous studies, as well as the design of future investigations. Moreover, a major finding of the work is that ectopic overexpression of a sodium channel reduces the levels of an endogenous sodium channel. The study is based on a lot of data acquired with a broad spectrum of approaches. While the calcium imaging and behavioral data are of high quality, robust, and generally support the major conclusions, the electrophysiology data are rather heterogenous, so that alternative interpretations cannot be fully ruled out yet. The following points should be addressed:

Major points

1. I have several questions regarding the patch-clamp recordings:

- a. APs were detected with Findpeaks in Matlab. Many APs look very small. What were the criteria for AP detection?
- b. Could other AP parameters, such as rheobase, peak, kinetics/width, or AHP, be analyzed? These parameters are expected to be influenced by loss of para.
- c. Given the scatter of some of the data (e.g., spikes/s; Fig. 1C), could they add the individual cells in addition to the average? Furthermore, correlating some of the parameters for individual cells may be informative (see below).
- d. What were the (average) series resistance (R_s) values of the respective groups, and was R_s compensated? IVs suggest that larger currents could not be clamped appropriately. Please provide the respective information.
- e. Please provide (average) absolute current amplitudes and membrane capacitance values to assess the R_s error. In addition, it would be helpful to add the membrane capacitance values to the representative data (e.g., Fig. 1G) to facilitate the comparison between the experimental groups.
- f. Were the currents leak subtracted? Please provide the corresponding information in the methods section.
- g. The decrease in AP number may be - in part - due to an increase in outward currents. Could they also quantify the steady-state outward currents and plot an IV? Since the membrane capacitance values are not reported, the representative traces shown in Fig. 1G suggest a marked difference in the steady-state outward current.

2. As acknowledged by the authors, some of the electrophysiology data are rather heterogeneous, with only about half of the NaChBac expressing cells displaying NaChBac-like currents. The authors argue that space-clamp errors are a major contributor to this heterogeneity. Given this heterogeneity, could they correlate some of the major parameters with one another for individual cells, such as AP number an inward current amplitude? Based on parameters that are affected by space clamp errors, such as current kinetics and reversal potential, could they correlate the data with these parameters, and roughly estimate the contribution of the space clamp error? A correlation with membrane capacitance may be also

informative.

3. Could the heterogeneity arise from different NaChBac expression levels? If available, quantifying NaChBac-GFP fluorescence in different cells may help assessing whether the heterogeneity in physiology may- in part - arise from different NaChBac expression levels.

4. The inward currents of the control group are very small, and the IVs do not suggest sodium currents (Fig. 1H, 3E). Could they cite previous work showing that these small inward currents are indeed sodium currents? In the NaChBac group, the inward currents are dominated by the NaChBac-like current. To assess whether there are changes in excitability and currents after NaChBac expression, it would be informative to probe excitability and IVs in the presence of NaChBac blockers, such as JZTx-27/JZTx-14, Tang et al., 2017; Zhang et al., 2018, or κ -LhTx-1; doi: 10.3389/fphar.2022.924661. This would reveal if para-like currents are decreased after NaChBac expression.

5. Although NaChBac expression reduces para levels, the modeling suggests that NaChBac expression alone reduces AP number (Fig. 7). A major question that remains unanswered is why NaChBac expression reduces AP number independent of para. Could the decrease in AP frequency be caused by a NaChBac-induced depolarization block of sodium channels? This possibility should be at least discussed. Alternatively, this hypothesis could be tested using the model and/or experiments: How do different levels of the NaChBac conductance affect other sodium conductances in the model? Alternatively, one could directly test this hypothesis in current clamp and voltage clamp by applying short and sustained depolarizations. In this regard, voltage-clamp protocols assessing sodium current inactivation, and recovery from inactivation would be very informative.

6. While odor-evoked responses were decreased after NaChBac expression for 2 days (Fig. 2), AP number was reduced after 2 and 4 days of NaChBac expression (Fig. 3). The peak inward current (Fig. 3E, F), and the reduction in AP number (Fig. 3B) seem slightly larger after 2 vs. 4 days of NaChBac expression. Similarly, the current to induce the max. NaChBac current seems slightly smaller after 2 days compared with 4 days of NaChBac expression. Together, these data imply that there may be small differences between the two time points. Considering the heterogeneity of the data, could they correlate these parameters (e.g., AP number vs. peak inward current; see above) for individual cells to assess whether there are (small) differences between the two time points? Given the importance of NaChBac expression levels (Fig. 7), could the different driver systems used in the calcium imaging and electrophysiology experiments contribute to the difference?

7. The evidence for compensatory plasticity is indirect. The observation of no changes in axonal/dendritic calcium transients after NaChBac expression for 4 days, along with a decrease in sparseness, as well as the decrease in para levels after constitutive NaChBac expression does not provide direct evidence for compensation. Statements regarding compensation should be toned down, or backed up by further discussion and/or experiments. For instance, statements like "While in Kenyon cells ectopic NaChBac expression causes a compensatory decrease in Para expression (Fig. 6)" (l. 446) should be revised. What is the evidence for compensation? Could NaChBac simply "outcompete" para?

Minor points

1. Terms like "paradoxically" should be avoided. For instance, a depolarization block may be actually expected after sodium channel overexpression (assuming no compensatory changes in other conductances).

2. What is the evidence that neuromuscular junctions stabilize synaptic strength itself? The following statement should be revised: "(...) whereas neuromuscular junctions stabilize synaptic strength itself to faithfully transmit motor commands, rather than maintaining average postsynaptic activity (Frank et al. 2020)." (l. 90)

3. There is a typo in Figure legend 2. Odor-evoked calcium influx is reduced by constitutive or 2-d, but not 4-d, NaChBac expression.

4. Figure 6: Please add the duration of NaChBac expression to the legend (and the figure).

Referee #2:

Overview: This is potentially a fascinating paper that addresses a fundamental problem in neuroscience, how neurons achieve long-term regulation of their excitability in the face of channel perturbations and environmental challenge. As such it builds on a long history of both experimental and computational work. Not surprisingly, it is most influenced by the earlier work in *Drosophila*, but this may or may not be framing the present work in the most general of terms, as the work on fly synaptic homeostasis is a fascinating outlier in our understanding of these very fundamental problems.

The authors study the effects of an ectopic bacterial voltage-gated sodium channel, NaChBac, on the excitability of Kenyon neurons in the fly. This is a fascinating perturbation as it specifically alters Na channel function and then allows the effects of this on excitability and other currents to be assayed. The authors appear to be surprised by the apparently counter-intuitive

effects they obtained, although the precedent for complex modifications of conductances that result from perturbations such as this are also available. For example, since the kinetics of the NaCHBac would tend to promote long-lasting plateaus, it is easy to understand why neurons might either be more or less excitable, as the dynamics of this current interact with other expressed voltage-dependent currents, and extending the duration of a plateau can slow firing...

The manuscript attempts to do a lot of different things, and as such is more difficult to understand than it might be, should it be less ambitious.

Specific Comments:

1. Lines 91-93 there is a fallacious general statement here that differences between central and peripheral nervous systems are necessarily accompanied by different strategies...the general statement more correctly would be that homeostatic mechanisms might be implemented differently at different locations in circuits, according to what is needed behaviorally.
2. Line 106...fallacy that the "central brain" is a simple and consistent structure.
3. Figure 1 Panels A and B are way too small. This figure has to be considerably bigger or broken down so that A,B, G, specifically are large enough to see. Likewise Figure 2A is way too small. Figure 4B is not clear (maybe needs to be bigger) Figure 6 C is too small. Figure 7...panels are missing important information like voltages, and may are too small.
4. What is plotted in Fig 1E? Clearly not the potential early in the pulse?
5. Would it be possible to get a more quantitative direct measure of para expression that doesn't depend on GFP fluorescence? PCR numbers?
6. What temperatures were the measured currents used for model fits? Were all modeled currents measured at the same temperature? Could temperature differences contribute to the slight discrepancies?

END OF COMMENTS

Response to reviewers, manuscript JP-RP-2025-288790R1

We thank the editor and reviewers for the constructive comments which have greatly improved the manuscript. Line numbers below are given as (line number in plain manuscript / line number in tracked changes version).

EDITOR COMMENTS

Your study of homeostatic regulation of neuronal excitability in Kenyon cells in Drosophila has been reviewed by two expert reviewers. They were generally quite positive about the manuscript, citing its potential relevance for scientists using ectopic channel expression to alter neuronal excitability, as well as the general field of homeostatic plasticity in neurons. Many of the results were considered to be of high quality. The reviewers did however raise a number of concerns, all of which would need to be addressed in a revised manuscript. The most important points included:

1. As requested by both reviewers, additional information is needed about the electrophysiological recordings, including several recording parameters as well as electrophysiological variables examined.

We have added new figures showing/analysing:

- Estimated uncompensated series resistance (**Fig. A1**)
- NaChBac absolute current amplitude (as opposed to current density) (**Fig. A1**)
- Predicted error from series resistance (**Fig. A1**)
- Correlations between various electrophysiological parameters (**Fig. A1, A3**)
- Height and slope of action potentials (**new Fig. 7**)
- Amplitude and IV curves of fast (non-NaChBac) inward currents (**new Fig. 7**)
- Steady-state voltage-gated outward currents (for point 3) (**new Fig. 8**)

2. As suggested by Reviewer 1, the authors should explore whether some of the heterogeneity in results such as spike rate can be explained by correlating them with other electrophysiological parameters.

Much of the heterogeneity in the measured NaChBac current likely arises from heterogeneity in input resistance (possibly from imperfect pipette seals), as the two parameters are correlated. This is illustrated in the new Appendix **Fig. A1**. We have now excluded cells with input resistance < 0.8 GΩ.

3. The authors should examine other potential explanations for the reduced excitability following ectopic sodium channel expression, including changes in outward potassium currents and depolarization-induced block of sodium currents.

In the **new Fig. 8**, we find no change in steady-state voltage-gated outward currents. Depolarization-induced block of sodium currents is now discussed in a new analysis of fast inward currents (**new Fig. 7**), analysed in the model in **Fig. 9** (formerly **Fig. 7**) and addressed in the Discussion.

4. The Discussion section would be improved with some additional discussion of the relevance of this study in flies for mammals. As pointed out by Reviewer 2, the fly system has been shown to be somewhat of an outlier in studies of synaptic homeostasis. Also, comments from both reviewers about the use of terms such as "paradoxical" should be heeded.

We added a comparison of effects of NaChBac in flies vs. mammals in the final paragraph (line 1096 / 1125 in tracked changes), and a reminder that homeostatic regulation of ion channels is conserved across species from crabs to humans, in the last paragraph of the Para section of the Discussion (lines 1034 / 1062 in tracked changes).

We have removed the word “paradoxically” throughout. We kept “surprisingly” because while the result may not be surprising when reasoning from first principles, it is surprising in the context of previous work that used NaChBac to overactivate, not silence, neurons.

5. The Methods section should be clear that most results, except the calcium traces, are depicted with error bars that reflect SD. Also, the authors should report exact p-values, except if $p < 0.001$. All journal policies regarding statistics and presentation of data should be followed.

We checked to ensure our error bars are described in every figure legend. To clarify, most of our error bars show the 95% confidence interval (CI) of the mean, rather than SD. **We feel that mean \pm 95% CI is the most appropriate summary statistic** because:

- The 95% CI gives an interval where the true mean is likely to fall, with 95% confidence (more precisely: if we repeatedly sample from the same underlying population, and calculate the 95% CI for each sample, then the true mean would fall inside that interval in ~95% of samples). We feel this is a more meaningful metric to show than SD when comparing the means of different groups, because it means the overlap in error bars provides a reasonable (though obviously not foolproof) visual impression of whether differences between groups are likely to be statistically significant.
- Since we show individual data points, the full spread of variability in data is already visible, so showing the SD adds less information than showing the 95% CI.
- We understand that people often choose to show SEM over SD because it makes the error bars smaller; however, 95% CI is actually bigger than the SD when $n \leq 6$.

We do use SEM for the calcium traces, but their corresponding data summaries show the individual data points with 95% CI, and the editor’s comment does not appear to object to the use of SEM for calcium traces. **We do not use the SEM alone for any data summaries.** We added this to the Methods: “Error bars in graphs are as described in the legends.”

We report exact p-values in **Table A3**, in accordance with the journal policy stating “Where there are many comparisons, a table of p values is requested.” As requested, we have moved it from the supporting information into the Appendix, but we are happy to consider an alternative format if this table is considered too long to be printed (233 rows) (e.g. it might be more space-efficient, though perhaps more cluttered, to put all the p-values in the figure legends instead).

In accordance with The Journal of Physiology’s policy about supporting information, we have moved former supporting figures into the main figures or the Appendix:

Formerly	Now
Fig. S1 (R13F02-GAL4 KC counts)	Fig. A4
Fig. S2 (R13F02-GAL4 with GAL80ts or AGES)	Fig. A5
Fig. S3 (Extra data for Fig. 2)	Fig. A6
Fig. S4 (pre-training odour preferences)	Fig. 5C, 5F

The reviewers may notice that some of the data points in the revised manuscript have changed. This is because:

1. We have now excluded some electrophysiological recordings with low input resistance ($R_{in} < 0.8 \text{ G}\Omega$)
2. We replaced the input resistance measurements generated from pCLAMP with input resistance measurements estimated directly from the current clamp recordings
3. We now quantify the NaChBac current from current traces smoothed by a 50 ms moving median filter (taking only the first 100 ms of the voltage step) to remove the repetitive fast inward currents representing unclamped spikes (see Reviewer 1 point 4 below)
4. Our current clamp recordings went up to +150 pA injections, but we only plotted the spike rate up to +100 pA because in many control cells, the spike rate started to drop above +100 pA, likely due to the inactivation of sodium channels by depolarization block. However, in the original manuscript, quantification of maximum spike rate did include the data from >+100 pA injections. We have since decided it is more appropriate to only count the maximum spike rate up to +100 pA for a more fair comparison between the control cells and NaChBac-expressing cells. In practice, this change had only minimal effects on the data.

Finally, we have added new data showing that Para expression is decreased not only after constitutive NaChBac expression but also after 2 or 4 days expression (**Fig. 6G,H**) (relevant to Reviewer 1 minor point 4).

Referee #1:

Employing a combination of patch-clamp electrophysiology, in vivo calcium imaging, olfactory learning, immunohistochemistry, genetics, and compartmental modeling, Greenin-Whitehead and colleagues investigated how overexpression of the bacterial voltage-gated sodium channel NaChBac impacts neural excitability and olfactory responses of Kenyon cells, as well as aversive olfactory learning in adult Drosophila. Constitutive, or NaChBac expression for 2 and 4 days in adult Kenyon cells decreased action potential (AP) number and sparsened odor-induced calcium responses in Kenyon cells. Moreover, NaChBac expression impaired aversive olfactory conditioning. Odor-induced calcium responses in Kenyon cells were reduced after constitutive, or NaChBac expression for 2 days, but not for 4 days. Furthermore, constitutive NaChBac expression decreased the levels of the sodium channel paralytic (para), and compartmental modeling reproduced the excitability phenotype induced by NaChBac expression when considering a decrease in a transient para-like sodium conductance. Modeling also suggests that low NaChBac expression levels increase, while high expression levels decrease AP number. Together, they propose that (strong) ectopic overexpression of a sodium channel leads to a decrease, rather than an increase in excitability and odor-evoked responses in Kenyon cells, as well as impaired olfactory learning.

NaChBac expression is a commonly used tool to enhance neuronal excitability. Hence, the results of this study impact the interpretation of previous studies, as well as the design of future investigations. Moreover, a major finding of the work is that ectopic overexpression of a sodium channel reduces the levels of an endogenous sodium channel. The study is based on a lot of data acquired with a broad spectrum of approaches. While the calcium imaging and behavioral data are of high quality, robust, and generally support the major conclusions, the electrophysiology data are

rather heterogenous, so that alternative interpretations cannot be fully ruled out yet. The following points should be addressed:

Major points

1. I have several questions regarding the patch-clamp recordings:

a. APs were detected with Findpeaks in Matlab. Many APs look very small. What were the criteria for AP detection?

We have added the following text in the Methods (lines 250-256 / 263-270 in tracked changes):

Since the measured KC action potentials' amplitudes were small and varied in size throughout time, findpeaks was only able to detect large, uniform action potentials. In cases where spike detection failed, spikes were detected using visual inspection. A spike was included based on a sharp depolarization and a pronounced narrow peak followed by repolarization. For characterisation of spike waveforms [refers to the reviewer's point 1b], spikes were only used if they were from cells that fired a total of at least 10 spikes over the range of +10 to +100 pA current injection.

b. Could other AP parameters, such as rheobase, peak, kinetics/width, or AHP, be analyzed? These parameters are expected to be influenced by loss of para.

We thank the reviewer for this suggestion. We have analysed the action potential waveforms in **new Fig. 7** (results text lines 789-869 / 811-891 in tracked changes). As expected, in NaChBac-expressing cells, the minimum somatic voltage at time of spike onset (an estimate of the spiking threshold) was higher, the spike amplitude (difference between peak voltage and voltage at time of spike onset) was lower, and the upstroke slope (peak dV/dt) was lower (though the latter two were only statistically significant in post-hoc tests with adult-only NaChBac expression, likely due to low sample size in the constitutive condition). We also analysed the amplitude of the fast inward currents recorded in voltage clamp, which likely represent unclamped spikes (see point 4 below), and these were also smaller with NaChBac expression.

We decided not to analyse after-hyperpolarization (AHP) in the revised manuscript. In the spike waveforms, it does appear that NaChBac-expressing cells have shallower AHPs. However, because the somatic voltage at spike onset was so much higher in NaChBac-expressing cells than control cells, this isn't necessarily a fair comparison: at higher voltages, the driving force for sodium is lower and sodium channels might be somewhat inactivated, making the spike shorter and potentially activating fewer potassium channels. For spike height and slope, we could control for the somatic voltage at spike onset (**Fig. 7G,H**), but this was more difficult for AHP, because at the voltages where NaChBac-expressing cells spike, control cells spike at such high frequencies that the AHP was absent. Thus, we couldn't do the analysis in **Fig. 7G,H** for AHPs. However, in voltage-clamp, many control cells showed a slow outward current after the fast inward currents, likely representing the AHP in unclamped spikes, and this outward current was smaller in NaChBac-expressing cells, though we didn't analyse this statistically. This would be expected if the spike was shallower, as the reduced depolarization would open fewer voltage-gated potassium channels.

c. Given the scatter of some of the data (e.g., spikes/s; Fig. 1C), could they add the individual cells in addition to the average? Furthermore, correlating some of the parameters for individual cells may be informative (see below).

We have added individual data points to the following graphs: **Fig. 1D** (formerly 1C), **1F** (formerly 1E), **1H**, **3A**, **3C**, **3E**, **5B**, **5C** (formerly S4A), **5E** (formerly 5D), **5F** (formerly S4B). We address the correlations in point 1d,e and point 2 below.

d. What were the (average) series resistance (R_s) values of the respective groups, and was R_s compensated? IVs suggest that larger currents could not be clamped appropriately. Please provide the respective information.

e. Please provide (average) absolute current amplitudes and membrane capacitance values to assess the R_s error.

As points d and e are related, we address them together.

For point e, we have added the following text into the figure legends

Fig. 1F (was 1E):

Absolute peak current amplitudes were (mean \pm s.d.): control, -7.0 ± 5.0 pA; NaChBac, -124.0 ± 162.6 pA. Membrane capacitances were (mean \pm s.d.): control, 14.7 ± 4.4 pF; NaChBac, 16.2 ± 3.6 pF.

Fig. 3F:

Absolute peak current amplitudes were (mean \pm s.d.): control 2 d, -7.3 ± 6.6 pA; NaChBac 2 d, -294.6 ± 180.0 pA; control 4 d, -7.3 ± 9.0 pA; NaChBac 4 d, -221.0 ± 123.3 pA; membrane capacitances were (mean \pm s.d.): control 2 d, 19.3 ± 9.7 pF; NaChBac 2 d, 13.6 ± 5.4 pF; control 4 d, 13.0 ± 5.1 pF; NaChBac 4 d, 13.9 ± 3.4 pF.

We also show absolute current amplitudes in **new Fig. A1A**.

Series resistance was automatically compensated by the amplifier software (pCLAMP) and the series resistance value was not recorded. However, we estimated the uncompensated series resistance by fitting an exponential decay to the capacitive transient at the start of the voltage step. The decay time constant equals the series resistance times the membrane capacitance: $\tau = R_s * C_m$ (Molleman, 2003). The membrane capacitance was estimated from the time constants in current clamp as explained in the original manuscript. Most of the recorded cells had a series resistance around 10-20 M Ω but there were a few outliers with high estimated series resistance. These may reflect cells with incomplete break-in.

Most cells with abnormally high estimated series resistance are now excluded by our decision to exclude cells with low input resistance (see point 2 below). There do remain 3 constitutive-NaChBac cells with somewhat high estimated series resistance (80, 83, 320 M Ω) but we decided not to exclude them because:

1. Thanks to their high input resistance, their ratio of series resistance to input resistance (R_s/R_{in}) was still fairly low (0.04, 0.08, 0.21, respectively), indicating that the vast majority of injected current exerted its effect across the cell membrane rather than the series resistance.
2. These cells show clear NaChBac currents (unlike most cells with low input resistance, which were excluded).
3. All the NaChBac cells have a peak NaChBac current at command voltages around -70 to -30 mV no matter what the R_s is, suggesting that series resistance did not strongly affect the NaChBac current's IV curve.

4. The estimated R_s error is low. The error in the leak current (i.e. the current required to step down to -100 mV in the voltage clamp experiments) can be estimated as $V_{\text{error}} = I_{\text{leak}} * R_s = (\Delta V / R_{\text{in}}) * R_s$. Thus, in the worst case of $R_s / R_{\text{in}} = 0.21$, the R_s error is 21%, so in stepping down from -60 mV to -100 mV, we would have reached -92 instead of -100 mV, which should still be sufficient to open the NaChBac channel. For the NaChBac current, the 3 cells with high R_s happened to have relatively small (though clear) NaChBac currents. Using $V_{\text{error}} = I_{\text{NaChBac}} * R_s$ the estimated R_s error was no higher than 10 mV.
5. In any case, the precise shape of the IV curve of the NaChBac current is not crucial to the main conclusions of our electrophysiological data, which are simply that (1) NaChBac expression produces a slow, voltage-gated inward current consistent with the reported properties of the NaChBac channel, and (2) NaChBac-expressing cells spike much less than control cells.

In all IV curves, we now label the x-axis “Command voltage (mV)” instead of “Voltage (mV)” to remind the reader that the true voltage experienced by the channel may differ from the command voltage.

Given the heterogeneity in series resistance, we plot the individual values rather than average values in the new Appendix **Fig. A1** and describe the analysis above in the Appendix. The raw data are presented in **Dataset S1**.

In addition, it would be helpful to add the membrane capacitance values to the representative data (e.g., Fig. 1G) to facilitate the comparison between the experimental groups.

We have added the membrane capacitance and input resistance of the two example cells shown in **Fig. 1B,C** (formerly 1B,G) to the figure legend.

f. Were the currents leak subtracted? Please provide the corresponding information in the methods section.

To calculate the NaChBac currents, we did not explicitly subtract the leak, but as we wrote in the original Methods, we did subtract the steady-state current in the last 100 ms of the voltage step, which includes the leak current. We have added the following text in the Methods:

Note that the current in the last 100 ms includes the leak current and any long-lasting voltage-gated currents, including NaChBac current in cases where it decays so slowly that there remains some NaChBac current in the last 100 ms; in these cases, this method likely underestimates the NaChBac current.

In the traces in **Fig. 1C** (formerly 1G), we did not subtract the leak current, both to present the data with minimal processing, and to show the currents cleanly separated so that the reader can more clearly see the fast inward currents that we interpret as unclamped action potentials. This has been clarified in the figure legend.

We did explicitly subtract the leak in response to the reviewer’s point (1g), that is, to estimate the component of this steady-state current which is a voltage-gated outward current, as opposed to the leak current. This information is now in the Methods.

g. The decrease in AP number may be - in part - due to an increase in outward currents. Could they also quantify the steady-state outward currents and plot an IV? Since the membrane capacitance values are not reported, the representative traces shown in Fig. 1G suggest a marked difference in the steady-state outward current.

We did not observe a significant difference in voltage-gated outward currents between control and NaChBac-expressing cells and have added this analysis in **new Fig. 8** (see explanation below; lines 871-891 / 893-913 in tracked changes). The difference in the outward currents in the representative traces in Fig. 1G (now **Fig. 1C**) most likely reflects a difference in the leak currents (the control cell had a higher input resistance than the NaChBac cell: 1.36 G Ω v. 0.99 G Ω , which is now noted in the figure legend).

To calculate voltage-gated steady-state outward currents, we first subtracted the baseline current at -100 mV before the voltage step from the median current in the last 100 ms of the voltage step. Next, we assumed that no significant voltage-gated currents were opened between -100 and -70 mV, so that the leak current could be extrapolated from a linear fit to the currents between -100 and -70 mV. This assumption seemed reasonable because the slopes of these linear fits (which should be 1/R, according to Ohm's law) matched the reciprocals of the input resistances calculated from the current clamp recordings:

The measured steady-state currents tended to diverge outward from the estimated linear leak current at higher voltages. For example, in the representative traces in **Fig. 1C** (formerly 1G), the steady-state outward current increases more per voltage step at higher voltages than would be expected by extrapolating a linear trend from the lower voltages. This divergence suggested the presence of a sustained voltage-gated outward current, such as that produced by the Shab channel (Smith et al., 2019), which is strongly expressed in Kenyon cells (Aso et al., 2019, Amin et al., 2020). To estimate the voltage-gated outward current, we subtracted the linear leak current. This method suggested that in control cells, the voltage-gated outward current started to open around -40 to -30 mV (**new Fig. 8A**). This can be seen on these IV plots which subtract only the leak current:

The dip in the IV curve in NaChBac-expressing cells is because some NaChBac-expressing cells had a NaChBac current that had not yet fully decayed by the end of the 1 s voltage step (note the decay time constants in **Fig. 3J**). Therefore, we subtracted the estimated residual NaChBac current in the last 100 ms as follows. We assumed that the measured current I is made of $I_{obs} = I_{leak} + I_K + I_{NaChBac}$, where I_{leak} and I_K are constant over the voltage step while $I_{NaChBac}$ has a peak (I_{Np}) early on and possibly a residual inward current (I_{Nr}) at the end. Note that because the peak NaChBac current was calculated by subtracting the current in the last 100 ms (i.e., I_{obs}), this observed peak NaChBac current $I_{Np,obs}$ actually represents $I_{Np} - I_{Nr}$.

Because the voltage-gated outward current in control cells was largely absent between -60 to -40 mV (i.e., $I_K = 0$), we took the trace with the biggest NaChBac current in that range, and used $I_{NaChBac} = I_{obs} - I_{leak}$ to estimate I_{Nr} . To estimate I_{Nr} at other voltages, we assumed that the kinetics of the NaChBac current are not dramatically altered by voltage (see e.g. Fig 1 in (Ren et al., 2001)). We calculated the ratio $I_{Nr}/I_{Np,obs}$ (the ratio of residual to peak NaChBac current), and applied this across the other voltage steps: e.g., $I_{Nr,-30mV} = I_{Np,obs,-30mV} * (I_{Nr,-60mV}/I_{Np,obs,-60mV})$. For the purposes of calculating I_{Nr} , only inward values of $I_{Np,obs}$ were used. Finally, we estimated $I_K = I_{obs} - I_{leak} - I_{Nr}$. Whether NaChBac was expressed for 2 d, for 4 d, or constitutively, it did not significantly change the estimated voltage-gated outward current (**new Fig. 8B,C**).

2. As acknowledged by the authors, some of the electrophysiology data are rather heterogeneous, with only about half of the NaChBac expressing cells displaying NaChBac-like currents. The authors argue that space-clamp errors are a major contributor to this heterogeneity. Given this heterogeneity, could they correlate some of the major parameters with one another for individual cells, such as AP number an inward current amplitude? Based on parameters that are affected by space clamp errors, such as current kinetics and reversal potential, could they correlate the data with these parameters, and roughly estimate the contribution of the space clamp error? A correlation with membrane capacitance may be also informative.

We thank the reviewer for this suggestion. After plotting the correlations suggested by the reviewer, we no longer believe that heterogeneity in space clamp errors is a major contributor to heterogeneity in measured NaChBac currents. In particular, we examined the current kinetics and reversal potentials of the fast inward currents (presumably from Para channels) analysed in point 4 below (**new Fig. 7**), supposing that in cells with worse space clamp, these Para currents would be slower (from spatial filtering) and have a shallower IV curve (from poor voltage control of the axon initial segment). Cells with smaller measured NaChBac currents did **not** tend to have slower Para currents or a shallower Para IV curve. Therefore, we have removed our speculation that heterogeneity in NaChBac currents is caused by heterogeneity in space clamp errors.

Instead, the measured NaChBac current was most strongly correlated with measured input resistance: cells with small/no measured NaChBac currents tended to have low input resistance (though the correlation isn't perfect – remaining variation may stem from heterogeneity in NaChBac expression levels as in point 3 below) (new Appendix figure **A1A,B**). It may be that recordings with a low input resistance simply had an unhealthy cell or an imperfect seal between the pipette and the membrane.

Based on this correlation, we decided to exclude cells with input resistance $< 0.8 \text{ G}\Omega$ from the main figures. The number of excluded cells is (number of excluded voltage clamp recordings in parentheses): control 2 d, 4 (1); NaChBac 2 d, 5 (4); control 4 d, 1 (1), NaChBac 4 d, 5 (5); control constitutive, 3 (2); NaChBac constitutive, 7 (4). After removing these cells, almost all NaChBac-expressing cells displayed NaChBac-like currents (5/7 constitutive, 8/8 2 d, 7/7 4 d).

Importantly, NaChBac-expressing cells spike less than control cells across the whole range of input resistances measured, even in the excluded cells (new Appendix figure **A1C**). However, in the interest of full reporting, we reproduce the graphs of **Fig. 1,3** in new **Fig. A3** with the excluded cells included, marked by "x"s. Unsurprisingly, the decrease in spiking caused by NaChBac expression is statistically significant whether or not the low- R_{in} cells are excluded.

3. Could the heterogeneity arise from different NaChBac expression levels? If available, quantifying NaChBac-GFP fluorescence in different cells may help assessing whether the heterogeneity in physiology may- in part - arise from different NaChBac expression levels.

We thank the reviewer for this suggestion. We have quantified NaChBac-GFP fluorescence in different brains and different cells within each brain and there is indeed significant heterogeneity in fluorescence. Similar heterogeneity is seen when expressing mCherry using the same GAL4 driver (coefficient of variation across flies of mean fluorescence across cells, NaChBacGFP = 0.295, mCherry = 0.304; within-fly coefficient of variation, NaChBac = 0.301 ± 0.0459 , mCherry = 0.339 ± 0.191 ; mean \pm s.d.). Thus, heterogeneity is likely an inherent feature of GAL4-driven expression. These data are now included in **Fig. A2**.

In the course of carrying out this experiment, we also obtained higher-resolution images of NaChBac-GFP expression, which are now shown in the new **Fig. 1A**.

4. The inward currents of the control group are very small, and the IVs do not suggest sodium currents (Fig. 1H, 3E). Could they cite previous work showing that these small inward currents are indeed sodium currents? In the NaChBac group, the inward currents are dominated by the NaChBac-like current. To assess whether there are changes in excitability and currents after NaChBac expression, it would be informative to probe excitability and IVs in the presence of NaChBac blockers, such JZTx-27/JZTx-14, Tang et al., 2017; Zhang et al., 2018, or κ -LhTx-1; doi: 10.3389/fphar.2022.924661. This would reveal if para-like currents are decreased after NaChBac expression.

The native voltage-gated sodium currents are not expected to follow the normal IV curve for sodium currents because of poor space clamp. Unlike NaChBac channels, which are expressed throughout the Kenyon cells, including the soma (**Fig. 1A**), Para channels are excluded from the soma and dendrites (**Fig. 6**) and are therefore electrotonically distant from the patch pipette. Thus, we have poor voltage control over Para currents. This can be noted by the fact that both control and NaChBac-expressing cells show repetitive fast inward currents even when we clamp the cell at +30 mV (**Fig. 1C**, formerly 1G) - if we had good voltage control over the Para currents, there would

be only one fast inward current and then no others as the channel would be inactivated by the depolarization block, as in e.g., (O'Dowd & Aldrich, 1988). (Note that O'Dowd and Aldrich 1988 was carried out on 24-h cultured dissociated neurons, whose processes are expected to be less extensive than that of a Kenyon cell *in vivo*, so space clamp would be less of a problem. They also explicitly excluded cells with poor space clamp, i.e. cells whose Para currents had flat IV curves or an extrapolated reversal potential $> +70$ mV.)

The reviewer is correct that the peak inward current as quantified in the original **Fig. 1H, 3E** is dominated by the NaChBac-like current. However, the NaChBac-like current is temporary and frequently Para currents start to appear after the NaChBac current is finished (e.g. **Fig. 1C**, formerly 1G), although they are less frequent than in control cells, consistent with the reduced action potentials. We appreciate that this was not readily apparent from the quantification of inward currents in our original figures. We have now updated the analysis of inward currents to separate out the slow inward current (i.e., NaChBac) from the repetitive fast inward currents (unclamped spikes, i.e. Para currents). To quantify the **slow** inward current, we now smooth the current trace using a moving median over a 50 ms time window. This smooths away the fast inward currents but preserves the slow inward current, and we take the trough of the smoothed trace in the first 100 ms of the voltage step as the amplitude of the slow current. Thus, the control group now doesn't show any inward currents on **Fig. 1H, 3E**.

We now quantify the **fast** inward currents in **new Fig. 7** to go along with the analysis of spike waveforms in response to the reviewer's point 1b. We have added the detection of fast inward current to the Methods:

To measure fast inward currents in voltage-clamp, the final 400 ms of each current trace (which largely excludes the NaChBac current) was smoothed by a moving average of 0.8-2.2 ms (the optimal smoothing window for picking up real events and rejecting noise was determined empirically for each cell). The first derivative dl/dt was taken and smoothed by the same moving average window. We used the Matlab findPeaks function to find events where (1) the derivative dl/dt showed a trough < -2.5 pA/ms followed by a peak > 2.5 pA/ms within 3 ms (each with $|prominence| > 5$ pA/ms, to remove double peaks) and (2) the current showed a trough with prominence > 2 pA between the peak and trough of dl/dt . The average current waveform at each voltage step was taken by averaging 3 ms before and 6 ms after the trough of the current (without smoothing) across all events.

Despite our inability to clamp away the fast Para currents, the fast Para currents did become somewhat smaller as the command voltage became more depolarised. Although we cannot truly clamp the axon initial segment to the command voltage, it most likely is at least depolarised more (i.e., closer to the reversal potential of sodium) when the command voltage is higher, thus the Para currents become smaller when the command voltage is higher (**new Fig. 7K**).

Further supporting the idea that these fast currents represent unclamped spikes, the amplitude of the fast inward currents is strongly correlated with spike height, spike upstroke slope, and total spike count (**Fig. 7M-O**).

It should be noted that rather few of our NaChBac cells showed clear Para currents that could be picked up with this analysis (1/8 voltage clamp recordings in 2-d cells, 3/7 in 4-d cells, and 4/7 in constitutive cells), consistent with our finding that NaChBac-expressing cells spike much less than control cells. We also now show that these currents are smaller in NaChBac-expressing cells (**Fig. 7I-P**), suggesting that Para conductances are indeed reduced, supporting our Para-FlpTag results and our current-clamp spiking data.

We agree that it would be interesting to use JZTx-27/JZTx-14 or κ -LhTx-1 to block NaChBac pharmacologically. However, we are not aware of these peptides being available commercially, or of any studies on whether they affect insect (as opposed to mammalian) sodium channels, and we considered it outside the scope of this manuscript to synthesize and purify these interesting toxins and test their effects on NaChBac and Para currents. We believe the analysis above addresses the reviewer's query about NaChBac's effects on Para currents such that blocking NaChBac pharmacologically is not necessary.

5. Although NaChBac expression reduces para levels, the modeling suggests that NaChBac expression alone reduces AP number (Fig. 7). A major question that remains unanswered is why NaChBac expression reduces AP number independent of para. Could the decrease in AP frequency be caused by a NaChBac-induced depolarization block of sodium channels? This possibility should be at least discussed. Alternatively, this hypothesis could be tested using the model and/or experiments: How do different levels of the NaChBac conductance affect other sodium conductances in the model? Alternatively, one could directly test this hypothesis in current clamp and voltage clamp by applying short and sustained depolarizations. In this regard, voltage-clamp protocols assessing sodium current inactivation, and recovery from inactivation would be very informative.

We appreciate the reviewer's suggestion to examine the potential for NaChBac-induced depolarization block of transient sodium channels, as it aligns with our original interpretation of the mechanism. To more clearly illustrate this point, we investigated changes in gating dynamics of the transient sodium current.

Our modeling shows that the inclusion of NaChBac in our models leads to sustained depolarization, which drives the membrane potential into a range where the transient sodium channels remain mostly inactivated. This occurs both in models with reduced para conductance and in those without para reduction, provided NaChBac is present at sufficient conductance levels (**new Fig. 9I-J**). NaChBac-induced depolarization of the membrane potential prevents recovery from inactivation of the transient sodium channels, thereby reducing their availability for the generation of repetitive spiking. At low NaChBac conductance, this effect is partial, allowing action potentials to persist (**Fig. A8B-F**). However, as NaChBac conductance increases, the prolonged depolarization leads to more complete and persistent inactivation of the transient sodium current, ultimately suppressing firing (**Fig. A8H-L**).

These simulations support the view that NaChBac-mediated depolarization imposes a voltage-dependent inactivation block of transient sodium channels even when Para expression levels are not reduced. However, the depolarization block and a decrease in Para conductance levels had a cumulative effect (compare **Fig. 9I** and **9J**), and our model fit the experimental data better when Para levels were reduced (**Fig. 9D-G,L**).

We have updated the main text to make this mechanism more explicit.

6. While odor-evoked responses were decreased after NaChBac expression for 2 days (Fig. 2), AP number was reduced after 2 and 4 days of NaChBac expression (Fig. 3). The peak inward current (Fig. 3E, F), and the reduction in AP number (Fig. 3B) seem slightly larger after 2 vs. 4 days of NaChBac expression. Similarly, the current to induce the max. NaChBac current seems slightly smaller after 2 days compared with 4 days of NaChBac expression. Together, these data imply that there may be small differences between the two time points. Considering the heterogeneity of the

data, could they correlate these parameters (e.g., AP number vs. peak inward current; see above) for individual cells to assess whether there are (small) differences between the two time points?

We thank the reviewer for this suggestion. We interpret the reviewer as suggesting that if two populations differ between each other along 2 parameters that are correlated, this difference might only show up when plotting the 2 parameters against each other rather than considering only 1 parameter at a time, for example if the two populations have different correlations:

Or two populations might be very overlapping on individual parameters, but more clearly separated when plotted in two dimensions:

(A) In this theoretical example, group A has parameter x sampled from the uniform distribution $U(1,11)$, and parameter y is the same as x plus Gaussian noise: $y = x + N(\mu=0, \sigma=0.2)$. Group B has $x = U(1,11)$, $y = 0.9 \cdot x + N(\mu=0, \sigma=0.2)$. Groups A and B are obviously different when both parameters are plotted together. PC1 and PC2 are the two principal components, PC1 = (1,1), PC2 = (-1,1). **(B)** However, Groups A and B are very overlapping when considering parameters x and y on their own. **(C)** The difference becomes clear and statistically significant when plotting y/x , which captures the fact that Group A has $y=x$ and Group B has $y=0.9 \cdot x$. **(D)** The difference is also clear when projecting the data onto the second principal component. (p-values for B-D: Mann-Whitney test)

To test whether either of these scenarios occurs in our data, in the **new Fig. A7**, we plotted spike count vs. peak slow inward current, and input resistance vs. inward current, as these were somewhat correlated (described in point 2). In these graphs, the 2 d and 4 d data did not show

obviously different correlations or any obvious visual separation between the two groups. To test this more rigorously:

1. We normalized NaChBac current to input resistance (analogous to panel C above). There was still no significant difference between 2 d and 4 d.
2. We ran principal components analysis (PCA) using the 3 features (input resistance, peak slow inward current, spike count) to see if the 2 d and 4 d data differed significantly along any of the 3 principal components. They did not.

We have added a new paragraph to the Results:

Although the NaChBac current and NaChBac's effect on spike rate appeared slightly smaller at 4 d than at 2 d, these differences were not statistically significant. Because cells with stronger NaChBac current density somewhat tended to have higher input resistance ($r=0.34$) and lower peak spike rate ($r=-0.21$), we asked whether a more subtle difference between 2 d and 4 d might appear when taking into account the covariance of these three parameters. No differences between 2 d and 4 d were apparent when plotting these variables against each other or when normalising NaChBac current to input resistance (**Fig. A7**). In addition, principal components analysis using these three parameters (NaChBac current, input resistance, spike rate) did not reveal significant differences between 2 d and 4 d along any of the principal components (**Fig. A7**). While we do not exclude the possibility that a larger sample size might reveal a significant but small difference between 2-d vs. 4-d NaChBac expression, our results show that both manipulations strongly reduce action potentials.

We decided to put this analysis in **Fig. A7** in the Appendix because we felt that adding normalization and PCA into the main figures would be a needless distraction.

Given the importance of NaChBac expression levels (Fig. 7), could the different driver systems used in the calcium imaging and electrophysiology experiments contribute to the difference?

We actually used the same driver systems in the calcium imaging and electrophysiology experiments in each condition (**Fig. 1B-I** matches **Fig. 1J**, while **Fig. 3** matches the AID-GAL80 data in **Fig. 2**). In addition, we reproduced the calcium imaging data with constitutive NaChBac expression with both driver systems (mb247-GAL4+R44E04-LexA>GAL80 in **Fig. 1J**, and R13F02-GAL4 in **Fig. 2A**).

7. The evidence for compensatory plasticity is indirect. The observation of no changes in axonal/dendritic calcium transients after NaChBac expression for 4 days, along with a decrease in sparseness, as well as the decrease in para levels after constitutive NaChBac expression does not provide direct evidence for compensation. Statements regarding compensation should be toned down, or backed up by further discussion and/or experiments. For instance, statements like "While in Kenyon cells ectopic NaChBac expression causes a compensatory decrease in Para expression (Fig. 6)" (l. 446) should be revised. What is the evidence for compensation? Could NaChBac simply "outcompete" para?

We thank the reviewer for this point. We have toned down statements of compensation throughout. For example, we changed the cited example to "While in Kenyon cells ectopic NaChBac expression **decreases** Para expression (Fig. 6)". In other places, we have, e.g., added "apparent" (lines in plain manuscript: 150, 1044 / **lines in track changes version: 156, 1072**), added "under this interpretation" (line 1004 / **1032**), replaced "this compensatory increase" with "this increase" (line 734 / **756**), replaced "show compensatory downregulation of" with "downregulate" (line 1099 /

1128), replaced “compensate downward” with “decrease” (line 1000 / 1027), replaced “is” with “would be” (line 1069 / 1098), or replaced “compensation” with “difference” (line 1059-60 / 1087-88), “change” (1061-65 / 1089-94) or “plasticity” (line 1038 / 1066). We have left our original phrasing intact where “compensation” refers to other studies, refers to a hypothesis or prediction, or already appears with qualifiers like “suggests that”.

In addition, we have added the following passages to the Discussion.

Lines 1012-32 / 1040-49 in tracked changes:

We do not exclude the alternative interpretation that decreased Para reflects a passive side effect of NaChBac expression, rather than homeostatic compensation driven by the cell sensing excess depolarization or sodium influx. A generic mechanism (e.g., NaChBac outcompetes other membrane proteins for trafficking or chaperone machinery) seems unlikely, since steady-state voltage-gated outward currents were unaffected by NaChBac expression (**Fig. 8**), and these currents are presumably carried by Shaw/Shab potassium channels that likely rely on the same trafficking or chaperone machinery and are expressed at similar levels to Para (Amin et al. 2020; Aso et al. 2019). However, it could be that Para expression is especially sensitive to, e.g., cellular stress triggered by excess depolarization

Lines 1055-1058 / 1083-86 in tracked changes:

As with Para expression, we do not exclude the alternative interpretation that the increase in odour-evoked axonal/dendritic calcium influx between 2 and 4 d reflects a passive side effect of NaChBac expression, rather than active compensation targeting a set point of calcium influx.

Minor points

1. Terms like “paradoxically” should be avoided. For instance, a depolarization block may be actually expected after sodium channel overexpression (assuming no compensatory changes in other conductances).

We have removed the word “paradoxically” throughout. We kept “surprisingly” because while the result may not be surprising when reasoning from first principles, it is surprising in the context of previous work that used NaChBac to overactivate, not silence, neurons.

2. What is the evidence that neuromuscular junctions stabilize synaptic strength itself? The following statement should be revised: “(...) whereas neuromuscular junctions stabilize synaptic strength itself to faithfully transmit motor commands, rather than maintaining average postsynaptic activity (Frank et al. 2020).” (l. 90)

We were thinking of, e.g., (Ouanounou et al., 2016). However, as Reviewer 2 also raised a concern with this passage and we did not think it was a critical part of the motivation for our study, we have removed this sentence and the preceding one (lines 95-100 in tracked changes).

3. There is a typo in Figure legend 2. Odor-evoked calcium influx is reduced by constitutive or 2-d, but not 4-d, NaChBac expression.

Thanks - we added the missing “is”.

4. *Figure 6: Please add the duration of NaChBac expression to the legend (and the figure).*

The data in the original manuscript was for constitutive expression; while the manuscript was in review, we started additional experiments to also test 2 and 4 day expression. We have added the new data as **Fig. 6G,H** and labelled the duration clearly. We note that while the difference in Para-GFP alone between control vs. NaChBac is not always statistically significant, thanks to the co-variation of GFP and dsRed, the normalized ratio of Para-GFP/dsRed is significantly different in all conditions (see Reviewer 1's point 6).

Referee #2:

Overview: This is potentially a fascinating paper that addresses a fundamental problem in neuroscience, how neurons achieve long-term regulation of their excitability in the face of channel perturbations and environmental challenge. As such it builds on a long history of both experimental and computational work. Not surprisingly, it is most influenced by the earlier work in Drosophila, but this may or may not be framing the present work in the most general of terms, as the work on fly synaptic homeostasis is a fascinating outlier in our understanding of these very fundamental problems.

The authors study the effects of an ectopic bacterial voltage-gated sodium channel, NaChBac, on the excitability of Kenyon neurons in the fly. This is a fascinating perturbation as it specifically alters Na channel function and then allows the effects of this on excitability and other currents to be assayed. The authors appear to be surprised by the apparently counter-intuitive effects they obtained, although the precedent for complex modifications of conductances that result from perturbations such as this are also available. For example, since the kinetics of the NaChBac would tend to promote long-lasting plateaus, it is easy to understand why neurons might either be more or less excitable, as the dynamics of this current interact with other expressed voltage-dependent currents, and extending the duration of a plateau can slow firing...

The manuscript attempts to do a lot of different things, and as such is more difficult to understand than it might be, should it be less ambitious.

Specific Comments:

1. Lines 91-93 there is a fallacious general statement here that differences between central and peripheral nervous systems are necessarily accompanied by different strategies...the general statement more correctly would be that homeostatic mechanisms might be implemented differently at different locations in circuits, according to what is needed behaviorally.

We take the reviewer's point and have removed this passage.

2. Line 106...fallacy that the "central brain" is a simple and consistent structure.

We are not sure exactly which phrase the reviewer would like us to change, as we do not believe that we have stated anywhere that the central brain is a simple and consistent structure. The

sentence at line 106 merely says that compensation in ion channels has not been studied in the central brain *in vivo*.

3. Figure 1 Panels A and B are way too small. This figure has to be considerably bigger or broken down so that A,B, G, specifically are large enough to see. Likewise Figure 2A is way too small. Figure 4B is not clear (maybe needs to be bigger) Figure 6 C is too small. Figure 7...panels are missing important information like voltages, and may are too small.

Figure 1: We replaced Panel A with new, higher-resolution images obtained in the course of quantifying NaChBac-GFP fluorescence as suggested by Reviewer 1. We enlarged and re-located panel B and G as panel B and C.

Figure 2A, 4B, 6C: The images have been enlarged

Figure 7. The traces have been enlarged, and the baseline membrane potentials for panels B-G have been added in the figure legend.

4. What is plotted in Fig 1E? Clearly not the potential early in the pulse?

We apologise for omitting this detail and have changed the figure legend for **Fig. 1F** (formerly **1E**) and **3C** to “Median membrane potential during the current pulse at each level of current injected (excluding the first and last 50 ms).”

5. Would it be possible to get a more quantitative direct measure of para expression that doesn't depend on GFP fluorescence? PCR numbers?

Unfortunately, PCR would be impractical - if we dissect out brains, even if we cut off the optic lobes, the Kenyon cells are only ~8-10% of the number of neurons in the midbrain (~2,000 Kenyon cells per hemisphere vs. 50,000 neurons in the midbrain, which doesn't include glia). Thus any changes in Para expression in Kenyon cells would be diluted out by other neurons, which are presumably unaffected (or at least less affected) by NaChBac expression specifically in Kenyon cells. To analyse Para expression specifically in Kenyon cells would require something like single-cell RNA-seq, which seems like overkill for this question, or cell-type-specific isolation of mRNAs, which we tried unsuccessfully in the past. We tried expressing NaChBac in all neurons using elav-GAL4 but, unsurprisingly, this manipulation was lethal (i.e., we crossed elav-GAL4 and UAS-NaChBac flies, and no adults carrying both transgenes hatched from the cross).

We argue that our GFP based measure is actually superior to PCR, as it reveals protein expression, not mRNA, and it also reveals the localization of the protein in addition to the level. With PCR, we would not have been able to see that Para is particularly reduced in the spike initiation zone (posterior peduncle).

6. What temperatures were the measured currents used for model fits? Were all modeled currents measured at the same temperature? Could temperature differences contribute to the slight discrepancies?

The recordings in Fig. 1,3 were done at 25 °C and we have added this information in the Methods. We fit our NaChBac model to data from (Strege et al., 2023) since it most closely reproduced the time course of our own recordings. While Strege et al. (2023) don't specify temperature and we can therefore assume recordings were done at room temperature, we compared their data to other

recordings from NaChBac channels that were performed at 28 °C (Kuzmenkin et al., 2004) and yielded very similar kinetics. The para and potassium channels were modeled after recordings at room temperature as well (Günay et al., 2015). Temperature differences would indeed introduce different kinetics; however, our aim was to qualitatively parallel the experimental recordings in our models and therefore fit the data at hand with the minimum complement of channels (NaChBac, para, K) necessary to investigate the underlying mechanisms.

References

- Kuzmenkin A, Bezanilla F & Correa AM (2004). Gating of the Bacterial Sodium Channel, NaChBac. *J Gen Physiol* 124, 349–356.
- Molleman A (2003). Patch Clamping: An Introductory Guide to Patch Clamp Electrophysiology.
- O'Dowd D & Aldrich R (1988). Voltage-clamp analysis of sodium channels in wild-type and mutant *Drosophila* neurons. *J Neurosci* 8, 3633–3643.
- Ouanounou G, Baux G & Bal T (2016). A novel synaptic plasticity rule explains homeostasis of neuromuscular transmission. *Elife*; DOI: 10.7554/elife.12190.005.
- Ren D, Navarro B, Xu H, Yue L, Shi Q & Clapham DE (2001). A prokaryotic voltage-gated sodium channel. *Science* 294, 2372–2375.
- Strege PR, Cowan LM, Alcaïno C, Mazzone A, Ahern CA, Milesco LS, Farrugia G & Beyder A (2023). Mechanosensitive pore opening of a prokaryotic voltage-gated sodium channel. *eLife* 12, e79271.

Dear Dr Lin,

Re: JP-RP-2025-288790R1 "**Ectopic sodium channel expression decreases excitability of *Drosophila* Kenyon cells**" by Katie Greenin-Whitehead, Eyal Rozenfeld, Anthony Moreno-Sanchez, Melissa W Tan, Kurtulus Kullu, Jessica Ausborn, Moshe Parnas, and Andrew C Lin

We are pleased to tell you that your paper has been accepted for publication in The Journal of Physiology.

Yours sincerely,

Nathan Schoppa
Senior Editor
The Journal of Physiology

If you would like to receive our 'Research Roundup', a monthly newsletter highlighting the cutting-edge research published in The Physiological Society's family of journals (The Journal of Physiology, Experimental Physiology, Physiological Reports, The Journal of Nutritional Physiology and The Journal of Precision Medicine: Health and Disease), please click this link, fill in your name and email address and select 'Research Roundup':
<https://www.physoc.org/journals-and-media/membernews>

- **TRANSPARENT PEER REVIEW POLICY:** To improve the transparency of its peer review process, The Journal of Physiology publishes online as supporting information the peer review history of all articles accepted for publication. Readers will have access to decision letters, including Editors' comments and referee reports, for each version of the manuscript as well as any author responses to peer review comments. Referees can decide whether or not they wish to be named on the peer review history document.
- You can help your research get the attention it deserves! Check out Wiley's free Promotion Guide for best-practice recommendations for promoting your work at: www.wileyauthors.com/eo/guide. You can learn more about Wiley Editing Services which offers professional video, design, and writing services to create shareable video abstracts, infographics, conference posters, lay summaries, and research news stories for your research at: www.wileyauthors.com/eo/promotion.
- **IMPORTANT NOTICE ABOUT OPEN ACCESS:** To assist authors whose funding agencies mandate public access to published research findings sooner than 12 months after publication, The Journal of Physiology allows authors to pay an Open Access (OA) fee to have their papers made freely available immediately on publication.

EDITOR COMMENTS

Senior Editor:

Thank you for submitting the revised manuscript. All prior concerns that were brought up by the referees and the editorial staff have been addressed, and the work is considered to be quite significant. Congratulations on the acceptance of your paper!

REFEREE COMMENTS

Referee #1:

The authors have addressed all of my points adequately.

Referee #2:

The authors have systematically and carefully considered the reviewers' and editor's comments, and made a number of revisions that improve the manuscript. I feel that this work addresses an important problem, and adds to our understanding of how excitability is regulated in neurons.